# Transfer Learning for Benign Overfitting in High-Dimensional Linear Regression

**Yeichan Kim**
Yonsei University
kimyeic98@yonsei.ac.kr

**Ilmun Kim**[*]
KAIST
ilmunk@kaist.ac.kr

**Seyoung Park**[*]
Yonsei University
ishspsy@yonsei.ac.kr

## Abstract

Transfer learning is a key component of modern machine learning, enhancing the performance of target tasks by leveraging diverse data sources. Simultaneously, overparameterized models such as the minimum-$\ell_2$-norm interpolator (MNI) in high-dimensional linear regression have garnered significant attention for their remarkable generalization capabilities, a property known as *benign overfitting*. Despite their individual importance, the intersection of transfer learning and MNI remains largely unexplored. Our research bridges this gap by proposing a novel two-step Transfer MNI approach and analyzing its trade-offs. We characterize its non-asymptotic excess risk and identify conditions under which it outperforms the target-only MNI. Our analysis reveals *free-lunch* covariate shift regimes, where leveraging heterogeneous data yields the benefit of knowledge transfer at limited cost. To operationalize our findings, we develop a data-driven procedure to detect *informative* sources and introduce an ensemble method incorporating multiple informative Transfer MNIs. Finite-sample experiments demonstrate the robustness of our methods to model and data heterogeneity, confirming their advantage.

## 1 Introduction

Transfer learning [49, 67, 72, 21] is a scheme that aims to improve the performance of a learning task of interest, namely the *target task*, by leveraging knowledge acquired from related, but possibly different, *source tasks*. As modern datasets grow in size and heterogeneity, the seamless integration of diverse sources of information has become increasingly critical, positioning transfer learning as a valuable approach. Its success is particularly evident in high-dimensional regression, as there has been a recent surge of research demonstrating its advantage in tasks including, but not limited to, LASSO [61] and its variants [6, 34, 20, 55], generalized linear models [60, 35], and nonparametric regression [40, 64, 38, 9].

While these transfer learning methods rely on explicit regularization [53, 17, 18] to address poor generalization arising from overfitting in high-dimensional regimes, contemporary deep learning methods have challenged the conventional wisdom of bias-variance trade-off framework. In particular, certain *interpolators* that achieve zero training error have been found to generalize remarkably well to unseen data, despite the absence of any explicit regularization. This surprising phenomenon of *benign overfitting* [7, 8, 5, 4, 44, 54] upends the traditional notion of model capacity and generalization. Given the prevalence of overparameterized models in modern machine learning, this stark contrast raises an intriguing question:

> *Can transfer learning further enhance the impressive out-of-sample generalization capabilities of such interpolators in high-dimensional linear regression?*

---

[*]Co-corresponding authors.

39th Conference on Neural Information Processing Systems (NeurIPS 2025).

**Related work.** Motivated by the empirical success of overparameterized deep learning models [70, 23, 46], a substantial body of research has sought to understand why benign overfitting occurs, focusing on linear regression as a tractable setting for theoretical investigation. The theoretical foundation of our research, along with that of subsequent studies, has been established by Bartlett et al. [5], who studied the non-asymptotic excess risk of the **minimum-$\ell_2$-norm interpolator (MNI)** with $n < p$ (i.e., dimension exceeds sample size). They demonstrated that if the covariance eigenvalues decay rapidly up to some "intrinsic" dimension $k^* < n$ (Assumption 2) and yet, the remaining $p - k^*$ components have *effective ranks* (Definition 1) greater than $n$, then the MNI trained on noisy data can achieve vanishing excess risk by capturing signal from the leading $k^*$ high-spectrum components and dispersing noise over the many low-spectrum tail components, as the tails span an "effective space" larger than the noise dimension $n$. In this situation, the MNI behaves similarly to ridge regression, exhibiting the effect known as *implicit regularization* [28]. Hastie et al. [19] explored the asymptotic excess risk of MNI as $p/n \to \gamma \in (0, \infty)$ based on random matrix theory [2], and Tsigler and Bartlett [62] extended this *ridgeless* interpolator to benign overfitting in ridge regression. Zhou et al. [71] and Koehler et al. [29] interpreted the benign overfitting of MNI under the lens of uniform convergence. Additionally, Muthukumar et al. [45] popularized the notion of *harmless interpolation*, demonstrating that under certain conditions, the MNI trained on noisy data does not severely degrade generalization, as estimation variance vanishes with increasing $p/n$.

In parallel, distribution shift in transfer learning for regression has been characterized by the two main pillars: (1) **model shift**, where the conditional distribution of response given covariates differs between the target and source tasks; (2) **covariate shift**, where the marginal distribution of covariates varies. Wang and Schneider [66] established a generalization bound under model shift, provided that the shift is smooth conditional on covariates. Recently, Tahir et al. [57] quantified model shift in high-dimensional regression by the cosine similarity between target and source model coefficients, showing that transfer performance depends critically on this alignment. Covariate shift has been studied along several lines, such as minimax risk formulations [32] and importance weighting [12], among others. Recent analyses of linear models under model and covariate shifts have examined optimal ridge regularization parameter, which can be negative [50].

Despite the potential significance of the aforementioned question, its exploration within transfer learning for MNIs remains severely limited. Mallinar et al. [42] studied an **out-of-distribution (OOD)** [39] setting[1], where the MNI is trained exclusively on source (in-distribution, ID) data and tested on the target distribution. They provided conditions where a beneficial covariate shift yields lower OOD risk than ID risk, focusing on shifts in the eigenvalues of spiked covariances [24] and degree of overparameterization. However, their approach does not use target samples during training, even though at least limited target data are often available. In contrast, Wu et al. [69] proposed an online **stochastic gradient descent (SGD)**[2] algorithm pre-trained with the source and then fine-tuned on the target, encompassing OOD settings by allowing no fine-tuning. They showed that pre-training with $O(n^2)$ can be comparable to $n$-target-supervised learning under certain covariate shifts. Despite using target data, their work, like Mallinar et al. [42], did not examine model shift, and its experiments are primarily focused on underparameterized regimes. Notably, Song et al. [56] developed the **pooled-MNI**, explicitly accounting for model shift. They derived its non-asymptotic excess risk and proposed a data-driven estimate for the optimal target sample size under model shift. However, pooling multiple sources can be susceptible to distribution shifts, as shown in our numerical evaluations (Section 5). Additional literature review is deferred to Appendix A, which discusses minimum-norm interpolators beyond the Euclidean norm.

**Summary of contributions.** Our study bridges the gap by investigating how transfer learning can enhance the generalization of target-only MNI through a novel knowledge transfer scheme. The main contributions are as follows.

- We propose a novel two-step **Transfer MNI**: pre-train a source-only MNI and then fine-tune it by interpolating target data while "staying close" to the pre-trained model. Under model shift with isotropic covariates, we identify when Transfer MNI outperforms target-only MNI, specify the optimal transfer size, and quantify the maximal improvement in excess risk. We provide non-asymptotic excess risk bounds under both model and covariate shifts when each single-task

---

[1]Concurrently with our work, Tang et al. [59] analyze OOD generalization of ridge regression and MNI and show that principal component regression can attain a faster convergence rate under certain conditions.

[2]SGD initialized at zero with suitable learning rates is *implicitly biased* toward the MNI [68].

MNI is benignly overfitted. We further uncover *free-lunch* covariate shifts that alleviate the cost of knowledge transfer while preserving its generalization gain for Transfer MNI.

- Based on a data-driven procedure for detecting *informative* sources inducing positive transfer, we propose an ensemble method aggregating multiple informative Transfer MNIs with data-adaptive weights determined by source informativeness.

- Finite-sample experiments demonstrate robustness to distribution shifts and superior performance over transfer baselines, including the pooled-MNI [56] and SGD-based transfer [69], thereby confirming the empirical advantage of our approach under overparameterization.

**Notation.** Bold upper- (e.g., $\mathbf{M}$ and $\mathbf{\Lambda}$) and lower-case (e.g., $\mathbf{v}$ and $\boldsymbol{\beta}$) letters denote matrices and vectors, respectively, with $\|\mathbf{M}\|$ and $\|\mathbf{v}\|$ denoting the operator and $\ell_2$-norms. For $Q \in \mathbb{N}$, define the sets $[Q] := \{1, 2, \ldots, Q\}$ and $[Q]_0 := \{0, 1, 2, \ldots, Q\}$. Write $a \vee b := \max(a, b)$ and $a \wedge b := \min(a, b)$. For $x \geq 0$, let $\lfloor x \rfloor$ denote the floor function of $x$. Denote by $\mathbb{1}(A)$ the indicator function of an event $A$. For sequences $\{a_n\}_{n \in \mathbb{N}}$ and $\{b_n\}_{n \in \mathbb{N}}$, write $a_n = O(b_n)$, $a_n \lesssim b_n$, or $b_n \gtrsim a_n$ if $|a_n/b_n| \leq c$ for some constant $c > 0$ and all sufficiently large $n$; write $a_n \asymp b_n$ if $a_n = O(b_n)$ and $b_n = O(a_n)$; write $a_n = o(b_n)$, $a_n \ll b_n$, or $b_n \gg a_n$ if $|a_n/b_n| \to 0$ as $n \to \infty$.

## 2 Preliminaries

**Task setup.** Consider an overparameterized linear regression setting involving one target task and $Q$ source tasks. We have access to the training datasets $\{(\mathbf{X}^{(q)}, \mathbf{y}^{(q)})\}_{q=0}^Q$, each with $n_q$ samples of $p$-dimensional covariate generated from the following well-specified linear model, with $q = 0$ corresponding to the target:

$$\mathbf{y}^{(q)} = \mathbf{X}^{(q)}\boldsymbol{\beta}^{(q)} + \boldsymbol{\epsilon}^{(q)}, \quad q \in [Q]_0. \tag{1}$$

Here, $\mathbf{y}^{(q)} \in \mathbb{R}^{n_q}$ is the response vector; $\mathbf{X}^{(q)} \in \mathbb{R}^{n_q \times p}$ is the design matrix with $i$-th row (sample) $\mathbf{x}_i^{(q)} \in \mathbb{R}^p$; $\boldsymbol{\beta}^{(q)} \in \mathbb{R}^p$ is the fixed model coefficient; $\boldsymbol{\epsilon}^{(q)} \in \mathbb{R}^{n_q}$ is the random noise. Provided $n_q \leq p$, the MNI trained on each dataset is uniquely given by

$$\hat{\boldsymbol{\beta}}_{\mathrm{M}}^{(q)} := \arg\min_{\boldsymbol{\beta} \in \mathbb{R}^p} \{\|\boldsymbol{\beta}\| : \mathbf{X}^{(q)}\boldsymbol{\beta} = \mathbf{y}^{(q)}\} = \mathbf{X}^{(q)\top}\big(\mathbf{X}^{(q)}\mathbf{X}^{(q)\top}\big)^\dagger \mathbf{y}^{(q)}, \quad q \in [Q]_0, \tag{2}$$

where $\mathbf{M}^\dagger \in \mathbb{R}^{d \times m}$ denotes the Moore–Penrose inverse [51] of $\mathbf{M} \in \mathbb{R}^{m \times d}$; Appendix C.1 provides a proof of its uniqueness and minimality in $\ell_2$-norm. The MNIs $\hat{\boldsymbol{\beta}}_{\mathrm{M}}^{(0)}$ and $\hat{\boldsymbol{\beta}}_{\mathrm{M}}^{(q)}$ correspond to our target and ($q$-th) source tasks, respectively.

**Distribution shift.** For well-specified linear models, **model shift** is attributed to the model *contrast* [34] defined as

$$\boldsymbol{\delta}^{(q)} := \boldsymbol{\beta}^{(q)} - \boldsymbol{\beta}^{(0)}, \quad q \in [Q].$$

On the other hand, **covariate shift** is characterized by discrepancies in the covariance structure $\mathbf{\Sigma} = \mathbb{E}\mathbf{x}\mathbf{x}^\top$. While Kausik et al. [27], LeJeune et al. [33], Mallinar et al. [42] assumed simultaneous diagonalizability, which gives spectral decompositions $\mathbf{\Sigma}^{(0)} = \mathbf{V}\mathbf{\Lambda}^{(0)}\mathbf{V}^\top$ and $\mathbf{\Sigma}^{(q)} = \mathbf{V}\mathbf{\Lambda}^{(q)}\mathbf{V}^\top$ for some common orthogonal matrix $\mathbf{V} \in \mathbb{R}^{p \times p}$, Assumption 1 and our numerical studies (Section 5) do not necessarily impose such a shared eigenbasis.

In Assumptions 1 and 2 and Definitions 1 and 2, the index $q$ applies for all $q \in [Q]_0$.

**Assumption 1.** *Let the design matrices $\mathcal{X} := \{\mathbf{X}^{(q)}\}_{q=0}^Q$ and random noises $\mathcal{E} := \{\boldsymbol{\epsilon}^{(q)}\}_{q=0}^Q$ be mutually independent, and the following holds.*

- *Each design is of the form $\mathbf{X}^{(q)} = \mathbf{Z}^{(q)}(\mathbf{\Sigma}^{(q)})^{1/2}$, where the rows of $\mathbf{Z}^{(q)} \in \mathbb{R}^{n_q \times p}$ are i.i.d. $\nu_x$-sub-Gaussian vectors[3] with i.i.d. mean-zero, unit-variance components, and $\mathbf{\Sigma}^{(q)} \in \mathbb{R}^{p \times p}$ is deterministic, symmetric positive definite with eigenvalues $\lambda_1^{(q)} \geq \lambda_2^{(q)} \geq \ldots \geq \lambda_p^{(q)} > 0$.*

- *Each $\|\mathbf{\Sigma}^{(0)}(\mathbf{\Sigma}^{(q)})^{-1}\|$ is bounded above by a universal constant $C_{\mathbf{\Sigma}^{(q)}} > 0$ with $C_{\mathbf{\Sigma}^{(0)}} = 1$.*

- *Each noise $\boldsymbol{\epsilon}^{(q)}$ has i.i.d. mean-zero components with finite variance $\sigma_q^2 > 0$.*

---

[3] A random vector $\mathbf{z} \in \mathbb{R}^p$ is $\nu$-sub-Gaussian if for any fixed unit vector $\mathbf{u} \in \mathbb{R}^p$, the random variable $\mathbf{u}^\top \mathbf{z}$ is $\nu$-sub-Gaussian (with variance proxy $\nu^2$), i.e., $\mathbb{E}\left[\exp\left(t\mathbf{u}^\top(\mathbf{z} - \mathbb{E}\mathbf{z})\right)\right] \leq \exp\left(t^2\nu^2/2\right)$ for all $t \in \mathbb{R}$.

The sub-Gaussianity of covariates is standard in recent transfer learning research [32, 12, 42, 20]. When $\mathbf{\Sigma}^{(0)}$ and $\mathbf{\Sigma}^{(q)}$ are simultaneously diagonalizable, the condition $\|\mathbf{\Sigma}^{(0)}(\mathbf{\Sigma}^{(q)})^{-1}\| = O(1)$ reduces to upper-bounded spectral ratios, i.e., $\max_{j\in[p]} \lambda_j^{(0)}/\lambda_j^{(q)} = O(1)$, as in the multiplicative spectral shift in Mallinar et al. [42]. Lastly, a finite second moment of the noise suffices as the formulation of mean squared loss in (5) only requires that each $\mathbb{E}\boldsymbol{\epsilon}^{(q)}\boldsymbol{\epsilon}^{(q)\top}$ is well-defined.

**Effective ranks.** We additionally introduce the key condition underpinning the benign overfitting of MNI, characterized by the *effective ranks* [30, 5] of $\mathbf{\Sigma}^{(q)}$.

**Definition 1.** *If $\lambda_{k+1}^{(q)} > 0$ for $k \geq 0$, the effective ranks of $\mathbf{\Sigma}^{(q)}$ are defined as*

$$r_k(\mathbf{\Sigma}^{(q)}) := \frac{\sum_{j>k} \lambda_j^{(q)}}{\lambda_{k+1}^{(q)}}, \qquad R_k(\mathbf{\Sigma}^{(q)}) := \frac{\left(\sum_{j>k} \lambda_j^{(q)}\right)^2}{\sum_{j>k}\left(\lambda_j^{(q)}\right)^2}.$$

**Assumption 2.** *There exists universal constants $b_q, c_q \geq 1$ such that the minimal index*

$$k_q^* := \min\left\{k \geq 0 : r_k(\mathbf{\Sigma}^{(q)}) \geq b_q n_q\right\}$$

*is well-defined with $0 \leq k_q^* \leq n_q/c_q$.*

Under Assumption 2, Bartlett et al. [5] formalized the benign overfitting of MNI as follows.

**Definition 2.** *The single-task MNI $\hat{\boldsymbol{\beta}}_{\mathrm{M}}^{(q)}$ with covariance $\mathbf{\Sigma}^{(q)}$ and $n_q < p$ is benign if*

$$\lim_{n_q\to\infty} \frac{r_0(\mathbf{\Sigma}^{(q)})}{n_q} = \lim_{n_q\to\infty} \frac{k_q^*}{n_q} = \lim_{n_q\to\infty} \frac{n_q}{R_{k_q^*}(\mathbf{\Sigma}^{(q)})} = 0.$$

## 3 Single-Source Transfer Task and Out-of-Sample Generalization

We propose a single-source transfer method in the form of *late-fusion* [56], where the $q$-th source task is learned independently and then integrated with the target task. In the first step, we pre-train the $q$-th source-only MNI $\hat{\boldsymbol{\beta}}_{\mathrm{M}}^{(q)}$. The two-step **Transfer MNI (TM)** $\hat{\boldsymbol{\beta}}_{\mathrm{TM}}^{(q)}$ then fine-tunes by interpolating the target dataset while minimizing the Euclidean distance from the pre-trained $\hat{\boldsymbol{\beta}}_{\mathrm{M}}^{(q)}$; that is,

$$\hat{\boldsymbol{\beta}}_{\mathrm{TM}}^{(q)} := \arg\min_{\boldsymbol{\beta}\in\mathbb{R}^p}\left\{\|\boldsymbol{\beta} - \hat{\boldsymbol{\beta}}_{\mathrm{M}}^{(q)}\| : \mathbf{X}^{(0)}\boldsymbol{\beta} = \mathbf{y}^{(0)}\right\}, \quad q \in [Q].$$

The TM estimate admits the following interpretable decomposition structure that clarifies the knowledge transfer mechanism (see Appendix C.2 for the derivation):

$$\underbrace{\hat{\boldsymbol{\beta}}_{\mathrm{TM}}^{(q)}}_{\text{transfer task}} = \underbrace{\hat{\boldsymbol{\beta}}_{\mathrm{M}}^{(0)}}_{\text{target task}} + \underbrace{\left(\mathbf{I}_p - \mathbf{H}^{(0)}\right)\hat{\boldsymbol{\beta}}_{\mathrm{M}}^{(q)}}_{\text{late-fusion knowledge transfer}}. \tag{3}$$

Here, $\mathbf{H}^{(q)} := \mathbf{X}^{(q)\top}\left(\mathbf{X}^{(q)}\mathbf{X}^{(q)\top}\right)^{\dagger}\mathbf{X}^{(q)}$ denotes the orthogonal projection onto the design row space $\mathcal{S}_q := \mathrm{span}\{\mathbf{x}_i^{(q)}\}_{i=1}^{n_q}$ for $q \in [Q]_0$; conversely, $\mathbf{I}_p - \mathbf{H}^{(q)}$ is the orthogonal projection onto the null space $\mathcal{S}_q^{\perp}$ such that $\mathbb{R}^p = \mathcal{S}_q \oplus \mathcal{S}_q^{\perp}$. Denoting by $(\boldsymbol{v})_{\mathcal{S}_0} = \mathbf{H}^{(0)}\boldsymbol{v}$ and $(\boldsymbol{v})_{\mathcal{S}_0^{\perp}} = (\mathbf{I}_p - \mathbf{H}^{(0)})\boldsymbol{v}$ the projections of a vector $\boldsymbol{v} \in \mathbb{R}^p$ onto the target row and null spaces respectively, we obtain

$$\left(\hat{\boldsymbol{\beta}}_{\mathrm{TM}}^{(q)}\right)_{\mathcal{S}_0} = \hat{\boldsymbol{\beta}}_{\mathrm{M}}^{(0)}, \qquad \left(\hat{\boldsymbol{\beta}}_{\mathrm{TM}}^{(q)}\right)_{\mathcal{S}_0^{\perp}} = \left(\mathbf{I}_p - \mathbf{H}^{(0)}\right)\hat{\boldsymbol{\beta}}_{\mathrm{M}}^{(q)}. \tag{4}$$

That is, the fine-tuning step for TM retains target-learned signal in the span of $n_0$ target samples (i.e., $\mathcal{S}_0$) where benign overfitting ensures high prediction accuracy for the target-only MNI, while transferring source information only into the null space $\mathcal{S}_0^{\perp}$ where the target samples provide no information with $\left(\hat{\boldsymbol{\beta}}_{\mathrm{M}}^{(0)}\right)_{\mathcal{S}_0^{\perp}} = \mathbf{0}_p$.

Given this "retain-plus-transfer" mechanism exhibited by TM, we next analyze its generalization behaviors by comparing the excess risks of target and transfer tasks. Suppose we are given an out-of-sample target instance $\mathbf{x}_0 \in \mathbb{R}^p$. The excess risk of an estimate $\hat{\boldsymbol{\beta}} \equiv \hat{\boldsymbol{\beta}}(\mathcal{X}, \mathcal{E})$ measured on the target distribution is defined by the following conditional mean squared loss:

$$\mathcal{R}(\hat{\boldsymbol{\beta}}) := \mathbb{E}_{(\mathbf{x}_0, \mathcal{E})}\left[\left(\mathbf{x}_0^{\top}\hat{\boldsymbol{\beta}} - \mathbf{x}_0^{\top}\boldsymbol{\beta}^{(0)}\right)^2 \mid \mathcal{X}\right] = \mathbb{E}_{\mathcal{E}}\left[\left(\hat{\boldsymbol{\beta}} - \boldsymbol{\beta}^{(0)}\right)^{\top}\mathbf{\Sigma}^{(0)}\left(\hat{\boldsymbol{\beta}} - \boldsymbol{\beta}^{(0)}\right) \mid \mathcal{X}\right]. \tag{5}$$

The canonical bias-variance decomposition shows that the excess risk is the sum of the (squared) bias $\mathcal{B}$ and variance $\mathcal{V}$, i.e., $\mathcal{R}(\hat{\boldsymbol{\beta}}) = \mathcal{B}(\hat{\boldsymbol{\beta}}) + \mathcal{V}(\hat{\boldsymbol{\beta}})$. The bias $\mathcal{B}_{\mathrm{M}}^{(0)}$ and variance $\mathcal{V}_{\mathrm{M}}^{(0)}$ of the target-only MNI are obtained by Hastie et al. [19] as follows:

$$\mathcal{B}_{\mathrm{M}}^{(0)} := \boldsymbol{\beta}^{(0)\top}\boldsymbol{\Pi}^{(0)}\boldsymbol{\beta}^{(0)}, \qquad \mathcal{V}_{\mathrm{M}}^{(0)} := \frac{\sigma_0^2}{n_0}\mathrm{Tr}\big(\hat{\boldsymbol{\Sigma}}^{(0)\dagger}\boldsymbol{\Sigma}^{(0)}\big), \tag{6}$$

where we write $\boldsymbol{\Pi}^{(0)} := (\mathbf{I}_p - \mathbf{H}^{(0)})\boldsymbol{\Sigma}^{(0)}(\mathbf{I}_p - \mathbf{H}^{(0)})$ and $\hat{\boldsymbol{\Sigma}}^{(q)} := (1/n_q)\mathbf{X}^{(q)\top}\mathbf{X}^{(q)}$ for $q \in [Q]_0$. The following lemma extends the bias-variance decomposition to our proposed TM estimate.

**Lemma 1.** *Under the mutual independence and mean-zero condition of $(\mathcal{X}, \mathcal{E})$ in Assumption 1, the excess risk of the TM estimate is the sum of bias $\mathcal{B}_{\mathrm{TM}}^{(q)}$ and variance $\mathcal{V}_{\mathrm{TM}}^{(q)}$ such that*

$$\mathcal{B}_{\mathrm{TM}}^{(q)} := \boldsymbol{\beta}^{(0)\top}(\mathbf{I}_p - \mathbf{H}^{(q)})\boldsymbol{\Pi}^{(0)}(\mathbf{I}_p - \mathbf{H}^{(q)})\boldsymbol{\beta}^{(0)} + \boldsymbol{\delta}^{(q)\top}\mathbf{H}^{(q)}\boldsymbol{\Pi}^{(0)}\mathbf{H}^{(q)}\boldsymbol{\delta}^{(q)}$$
$$- 2\boldsymbol{\delta}^{(q)\top}\mathbf{H}^{(q)}\boldsymbol{\Pi}^{(0)}(\mathbf{I}_p - \mathbf{H}^{(q)})\boldsymbol{\beta}^{(0)},$$
$$\mathcal{V}_{\mathrm{TM}}^{(q)} := \underbrace{\frac{\sigma_q^2}{n_q}\mathrm{Tr}\big(\hat{\boldsymbol{\Sigma}}^{(q)\dagger}\boldsymbol{\Pi}^{(0)}\big)}_{=: \mathcal{V}_{\uparrow}^{(q)} \text{ (variance inflation)}} + \mathcal{V}_{\mathrm{M}}^{(0)}.$$

**Remark 1** (Bias reduction versus variance inflation). *The variance inflation $\mathcal{V}_{\uparrow}^{(q)}$ in Lemma 1 is positive almost surely, so the knowledge transfer always requires a higher variance as its cost. If $\hat{\boldsymbol{\beta}}_{\mathrm{TM}}^{(q)}$ reduces its estimation bias enough to outweigh the variance inflation, it attains a lower excess risk than the target-only MNI, referred to as positive transfer.*

### 3.1 Model Shift under Isotropic Covariates

We analyze the effect of model shift under the isotropic case where $\boldsymbol{\Sigma}^{(0)} = \boldsymbol{\Sigma}^{(q)} = \mathbf{I}_p$. Although not "benign," this case ensures at least a "harmless" [45] performance for the single-task MNI, making it a worthwhile subject of investigation. In addition, our analysis provides an important insight: the dynamics between bias reduction and variance inflation (Remark 1) is determined by the interplay among $p$, $n_q$, shift-to-signal ratio (SSR), and signal-to-noise ratio (SNR), where SSR and SNR are defined for each $q$-th source as

$$\mathrm{SSR}_q := \frac{\|\boldsymbol{\delta}^{(q)}\|^2}{\|\boldsymbol{\beta}^{(0)}\|^2} \geq 0, \qquad \mathrm{SNR}_q := \frac{\|\boldsymbol{\beta}^{(0)}\|^2}{\sigma_q^2} > 0, \quad q \in [Q],$$

provided $\boldsymbol{\beta}^{(0)} \neq \mathbf{0}_p$.

**Theorem 1.** *Under Assumption 1, and further assuming that $(\mathbf{Z}^{(0)}, \mathbf{Z}^{(q)})$ have i.i.d. standard Gaussian entries with $p > (n_0 + 1) \vee (n_q + 1)$ and $\boldsymbol{\Sigma}^{(0)} = \boldsymbol{\Sigma}^{(q)} = \mathbf{I}_p$, the expected bias and variance (expectation over $\mathcal{X}$) of the target-only MNI and TM estimate are as follows:*

$$\mathbb{E}_{\mathcal{X}}\mathcal{B}_{\mathrm{M}}^{(0)} = \frac{p - n_0}{p}\|\boldsymbol{\beta}^{(0)}\|^2, \qquad \mathbb{E}_{\mathcal{X}}\mathcal{B}_{\mathrm{TM}}^{(q)} = \frac{p - n_0}{p}\Big(\frac{p - n_q}{p}\|\boldsymbol{\beta}^{(0)}\|^2 + \frac{n_q}{p}\|\boldsymbol{\delta}^{(q)}\|^2\Big),$$

$$\mathbb{E}_{\mathcal{X}}\mathcal{V}_{\mathrm{M}}^{(0)} = \frac{\sigma_0^2 n_0}{p - (n_0 + 1)}, \qquad \mathbb{E}_{\mathcal{X}}\mathcal{V}_{\mathrm{TM}}^{(q)} = \underbrace{\Big(\frac{p - n_0}{p}\Big)\frac{\sigma_q^2 n_q}{p - (n_q + 1)}}_{= \mathbb{E}_{\mathcal{X}}\mathcal{V}_{\uparrow}^{(q)}} + \mathbb{E}_{\mathcal{X}}\mathcal{V}_{\mathrm{M}}^{(0)}.$$

From Theorem 1, we observe a trade-off between $p$ and $n_q$. If $\|\boldsymbol{\beta}^{(0)}\| > \|\boldsymbol{\delta}^{(q)}\|$, increasing $n_q$ further reduces the bias of TM. An increase in $p$ mitigates the variance inflation, but only at the expense of compromising this bias reduction effect. Based on Theorem 1, we formalize the regime where TM is expected to outperform the target-only MNI and specify the optimal transfer size maximizing the improvement in expected excess risk $\mathbb{E}_{\mathcal{X}}\mathcal{R}_{\mathrm{M}}^{(0)} - \mathbb{E}_{\mathcal{X}}\mathcal{R}_{\mathrm{TM}}^{(q)}$.

**Corollary 1.** *Under the setup in Theorem 1, the TM estimate satisfies*

$$\mathbb{E}_{\mathcal{X}}\mathcal{R}_{\mathrm{TM}}^{(q)} < \mathbb{E}_{\mathcal{X}}\mathcal{R}_{\mathrm{M}}^{(0)} \iff \mathrm{SSR}_q < 1 \ \text{ and } \ \mathrm{SNR}_q(1 - \mathrm{SSR}_q) > \frac{p}{p - (n_q + 1)}.$$

*The improvement in expected excess risk $\Delta(n_q) := \mathbb{E}_{\mathcal{X}} \mathcal{R}_{\mathrm{M}}^{(0)} - \mathbb{E}_{\mathcal{X}} \mathcal{R}_{\mathrm{TM}}^{(q)}$ as a function of $n_q$ is strictly concave on $n_q \in [1, p-1]$. If and only if $\mathrm{SSR}_q < 1$ and $\mathrm{SNR}_q(1 - \mathrm{SSR}_q) \geq \frac{p(p-1)}{(p-2)^2}$, the optimal transfer size $n_q^*$ maximizing $\Delta(n_q)$ exists and equals $n_q^* = p - 1 - \sqrt{\frac{p(p-1)}{\mathrm{SNR}_q(1-\mathrm{SSR}_q)}} \in [1, p-1)$, with $\Delta(n_q^*) = \left( \frac{p-n_0}{p} \right) \left( \frac{(n_q^*)^2(1-\mathrm{SSR}_q)}{p(p-1)} \right) \|\boldsymbol{\beta}^{(0)}\|^2 > 0$ being the maximal improvement.*

In Corollary 1, negative transfer occurs when model shift dominates the signal with $\mathrm{SSR}_q \geq 1$. The improvement $\Delta(n_q)$ strictly increases in $n_q$ up to the optimal threshold $n_q^*$ but strictly decreases for $n_q > n_q^*$; that is, transferring more source samples helps only up to $n_q^*$, and beyond $n_q^*$, it always degrades the efficacy of knowledge transfer. While Song et al. [56] also investigated the same isotropic Gaussian setting under model shift, only Corollary 1 provides necessary and sufficient conditions for positive transfer and identifies the maximal improvement $\Delta(n_q^*)$.

## 3.2 Convergence under Benign Covariates

Taking both model and covariate shifts into account, we now consider general sub-Gaussian covariates satisfying Assumption 2 and analyze non-asymptotic excess risk bounds and their implication.

**Theorem 2.** *Let Assumptions 1 and 2 hold. For each $q \in [Q]_0$ and any $t \geq \log(2)$, define*

$$\psi_q(t) := \sqrt{\frac{r_0(\boldsymbol{\Sigma}^{(q)}) + t}{n_q}} + \frac{r_0(\boldsymbol{\Sigma}^{(q)}) + t}{n_q}, \qquad \Upsilon_q := \frac{k_q^*}{n_q} + \frac{n_q}{R_{k_q^*}(\boldsymbol{\Sigma}^{(q)})}.$$

*With probability at least $1 - 2e^{-\delta}$ and $1 - 4e^{-\eta}$ respectively, the biases are bounded above by*

$$\mathcal{B}_{\mathrm{M}}^{(0)} \lesssim \psi_0(\delta) \|\boldsymbol{\Sigma}^{(0)}\| \|\boldsymbol{\beta}^{(0)}\|^2,$$

$$\mathcal{B}_{\mathrm{TM}}^{(q)} \lesssim \psi_0(\eta) \|\boldsymbol{\Sigma}^{(0)}\| \|\boldsymbol{\delta}^{(q)}\|^2 + \psi_q(\eta) C_{\boldsymbol{\Sigma}^{(q)}} \|\boldsymbol{\Sigma}^{(q)}\| \|\boldsymbol{\beta}^{(0)}\|^2.$$

*Furthermore, there exist universal constants $c_0, c_q \geq 1$ such that with probability at least $1 - 7e^{-n_q/c_q} - 2e^{-\xi}$, the variance inflation of the TM estimate is bounded above by*

$$\mathcal{V}_{\uparrow}^{(q)} \lesssim \sigma_q^2 \Upsilon_q \psi_0(\xi) \big( \lambda_p^{(q)} \big)^{-1} \|\boldsymbol{\Sigma}^{(0)}\|,$$

*and with probability at least $1 - 10e^{-n_0/c_0}$, the excess risk of the TM estimate is bounded below by*

$$\mathcal{R}_{\mathrm{TM}}^{(q)} \gtrsim \sigma_0^2 \Upsilon_0,$$

*where $\mathcal{V}_{\mathrm{M}}^{(0)} \asymp \sigma_0^2 \Upsilon_0$ on the same high-probability event lower-bounding $\mathcal{R}_{\mathrm{TM}}^{(q)}$.*

The bias dynamics in Theorem 2 mirrors that in the isotropic case. If $r_0(\boldsymbol{\Sigma}^{(0)}) \ll n_0$, which is sufficient for a vanishing target-only bias, and $r_0(\boldsymbol{\Sigma}^{(0)}) \asymp r_0(\boldsymbol{\Sigma}^{(q)})$, $\mathcal{B}_{\mathrm{TM}}^{(q)}$ also vanishes. Moreover, it can achieve a faster convergence if $\|\boldsymbol{\delta}^{(q)}\|^2 \ll \|\boldsymbol{\beta}^{(0)}\|^2$ and $n_0 < n_q$. As for variance, the target- and source-only variances are within a constant factor of $\Upsilon_0$ and $\Upsilon_q$ respectively (see Corollary 3 in Appendix), and hence the terms vanish when each single-task is benign. The upper bound on $\mathcal{V}_{\uparrow}^{(q)}$ is a product of the two "benign" terms $\Upsilon_q$ and $\psi_0$ and the reciprocal of eigenvalue that reflects the lack of simultaneous diagonalizability in Assumption 1. While the reciprocal may loosen the bound, our numerical experiments (Section 5) show rapid decay in variances as $p \gg n_0 \vee n_q$. Exploring a tractable covariance structure to refine this bound is left for future work. Finally, the lower bound on $\mathcal{R}_{\mathrm{TM}}^{(q)}$ follows from Lemma 1, where the TM variance is no smaller than the target-only variance; the bias reduction effect cannot further improve the lower bound.

## 3.3 Free-Lunch Covariate Shift

In light of the invariance of spectral decay rates to uniform upscaling, we propose a type of covariate shift that yields a lower TM excess risk than without any covariate shift. To proceed, define the following minimal index for $\boldsymbol{\Sigma}^{(0)}$ satisfying Assumption 1, which is always well-defined:

$$\tau^* := \min \left\{ k < p : \lambda_{k+1}^{(0)} \asymp \lambda_p^{(0)} \right\}. \tag{7}$$

We may expect $\tau^* \approx k_0^*$ for some benign covariance structures, where $k_0^*$ is as specified in Assumption 2 for $\boldsymbol{\Sigma}^{(0)}$; the beginning of Appendix C.7 illustrates a case where $\tau^* \ll p$ and $\tau^* \ll n_0$ indeed.

**Corollary 2** (*Free-lunch* covariate shift). *For each $q \in [Q]_0$, let the spectral decomposition of $\mathbf{\Sigma}^{(q)}$ be $\mathbf{\Sigma}^{(q)} = \mathbf{V}^{(q)}\mathbf{\Lambda}^{(q)}\mathbf{V}^{(q)\top}$, where $\mathbf{V}^{(q)} \in \mathbb{R}^{p \times p}$ is orthogonal and $\mathbf{\Lambda}^{(q)} = \mathrm{Diag}(\lambda_1^{(q)}, \ldots, \lambda_p^{(q)})$. Suppose $\mathbf{\Lambda}^{(q)} = \alpha\mathbf{\Lambda}^{(0)}$ for some $\alpha > 1$, and under (A), the leading $\tau^*$ eigenvector pairs in $(\mathbf{V}^{(q)}, \mathbf{V}^{(0)})$ align; under (B), eigenvector pairs fully align with $\mathbf{V}^{(q)} = \mathbf{V}^{(0)}$, i.e., $\mathbf{\Sigma}^{(q)} = \alpha\mathbf{\Sigma}^{(0)}$.*

*Compared to the homogeneous case $\mathbf{\Sigma}^{(q)} = \mathbf{\Sigma}^{(0)}$, the following holds for each covariate shift case:*

(A) *The upper bound on $\mathcal{B}_{\mathrm{TM}}^{(q)}$ remains identical up to a constant factor independent of $\alpha$, while the upper bound on $\mathcal{V}_{\uparrow}^{(q)}$ is multiplied by $\alpha^{-1}$ (i.e., reduced by a factor of $\alpha$).*

(B) *The exact bias remains identical, while the exact variance inflation is multiplied by $\alpha^{-1}$.*

The covariate shift (A) in Corollary 2 shows that as long as each $\lambda_j^{(q)}$ is uniformly upscaled from $\lambda_j^{(0)}$, preserving their decay rates, any misalignment in "noisy" eigendirections between $\mathbf{\Sigma}^{(q)}$ and $\mathbf{\Sigma}^{(0)}$ beyond the leading $\tau^*$ high-signal components does not affect the convergence rate of bias, while still promoting faster convergence in variance inflation. With all $p > \tau^*$ eigenvector pairs aligning between $\mathbf{\Sigma}^{(q)}$ and $\mathbf{\Sigma}^{(0)}$, covariate shift (B) offers a greater advantage than (A) by specifying the exact impact on the bias and variance inflation, rather than on their upper bounds.

**Remark 2** (Relaxed *free-lunch* condition). *The $\tau^*$-alignment condition for* (A) *in Corollary 2 can be relaxed: even if not all of leading $\tau^*$ source eigenvectors align with the target counterparts, we can still achieve the same free-lunch effect as in* (A) *whenever*

$$\|\mathbf{V}_{\tau^*}^{(0)} - \mathbf{V}_{\tau^*}^{(q)}\| \lesssim \lambda_{\tau^*+1}^{(0)}/\lambda_1^{(0)},$$

*where $\mathbf{V}_{\tau^*}^{(0)} \in \mathbb{R}^{p \times \tau^*}$ (resp. $\mathbf{V}_{\tau^*}^{(q)} \in \mathbb{R}^{p \times \tau^*}$) comprises the leading $\tau^*$ eigenvectors in $\mathbf{V}^{(0)}$ (resp. $\mathbf{V}^{(q)}$) as specified in Corollary 2.*

## 4 Informative Multi-Source Transfer Task

In this section, we propose a transfer task that incorporates multiple sources identified as *informative*, those that induce positive transfer. The index set of informative sources is given by

$$\mathcal{I} := \Big\{ q \in [Q] : \mathcal{R}_{\mathrm{TM}}^{(q)} - \mathcal{R}_{\mathrm{M}}^{(0)} < 0 \Big\}, \tag{8}$$

which, however, is unknown in practice. Hence, it is of crucial interest in transfer learning to develop a data-driven procedure for detecting informative sources. Inspired by Tian and Feng [60], we utilize the $K$-fold cross-validation (CV) [1] to detect source transferability by comparing a "proxy" of excess risks.

First, partition the target dataset into $K$ folds of equal size, each denoted by $(\mathbf{X}^{(0)[k]}, \mathbf{y}^{(0)[k]})$ for $k \in [K]$; a common choice suggests $K = 5$ [17]. At each training step, we use the left-out folds $(\mathbf{X}^{(0)[-k]}, \mathbf{y}^{(0)[-k]}) := \{(\mathbf{X}^{(0)[k]}, \mathbf{y}^{(0)[k]})\}_{k=1}^K \setminus (\mathbf{X}^{(0)[k]}, \mathbf{y}^{(0)[k]})$ to train an estimate $\hat{\boldsymbol{\beta}}^{[-k]}$ and then evaluate the squared loss on the $k$-th fold given by

$$\widehat{\mathcal{L}}^{[k]}\big(\hat{\boldsymbol{\beta}}^{[-k]}\big) := \frac{1}{n_0/K}\big\|\mathbf{y}^{(0)[k]} - \mathbf{X}^{(0)[k]}\hat{\boldsymbol{\beta}}^{[-k]}\big\|^2.$$

We repeat this across all $K$ folds to obtain the terminal CV loss $\widehat{\mathcal{L}}\big(\hat{\boldsymbol{\beta}}\big) := \sum_{k=1}^K \widehat{\mathcal{L}}^{[k]}\big(\hat{\boldsymbol{\beta}}^{[-k]}\big)/K$, which estimates the prediction risk $\mathcal{R}\big(\hat{\boldsymbol{\beta}}\big) + \sigma_0^2$. We then estimate the oracle set (8) by

$$\widehat{\mathcal{I}} := \Big\{ q \in [Q] : \widehat{\mathcal{L}}\big(\hat{\boldsymbol{\beta}}_{\mathrm{TM}}^{(q)}\big) - \widehat{\mathcal{L}}\big(\hat{\boldsymbol{\beta}}_{\mathrm{M}}^{(0)}\big) \leq D^{(0)} \Big\}, \tag{9}$$

with some detection threshold $D^{(0)} > 0$ to be specified later that depends on the target CV loss.

If $\widehat{\mathcal{I}}$ is non-empty, we train the TM estimate $\hat{\boldsymbol{\beta}}_{\mathrm{TM}}^{(i)}$ for each $i \in \widehat{\mathcal{I}}$ and form a weighted linear combination of $\hat{\boldsymbol{\beta}}_{\mathrm{TM}}^{(i)}$. Each weight $w_i$ is initialized by the inverse of the CV loss $\widehat{\mathcal{L}}(\hat{\boldsymbol{\beta}}_{\mathrm{TM}}^{(i)})$ and then normalized so that $\sum_{i \in \widehat{\mathcal{I}}} w_i = 1$, which serves as a data-adaptive measure of source informativeness. This allows us to leverage $|\widehat{\mathcal{I}}|$ sources for knowledge transfer, which we name **Informative-Weighted Transfer MNI (WTM)**. The entire procedure, from detecting informative sources to computing the WTM estimate, is outlined in Algorithm 1 (Appendix D), specifying $D^{(0)}$ in the set (9).

# 5 Numerical Experiments

We evaluate the finite-sample performance of our proposed TM and WTM estimates by comparing them to the target-only MNI as a baseline. For the WTM estimate, we set $K = 5$ and $\varepsilon_0 = 1/2$ in Algorithm 1. Additionally, we test the pooled-MNI (PM) proposed by Song et al. [56] as a benchmark transfer task, which has an analytical solution:

$$\hat{\boldsymbol{\beta}}_{\mathrm{PM}} := \arg\min_{\boldsymbol{\beta} \in \mathbb{R}^p} \big\{ \|\boldsymbol{\beta}\| : \mathbf{X}^{(q)}\boldsymbol{\beta} = \mathbf{y}^{(q)}, \ \forall q \in [Q]_0 \big\}.$$

To consider a fine-tuning-based benchmark, we also test $\mathrm{SGD}_q$ [69] each pre-trained with the $q$-th source, adjusting the initial learning rate from their sourcecode to ensure convergence.

Each setup is replicated over 50 independent simulations, and the average excess risk over the 50 simulations is plotted against each value of $p \in \{300, 400, \ldots, 1000\}$. The mutually independent noise has i.i.d. mean-zero Gaussian components with common variance $\sigma^2 = 1$. We adopt the parametric configurations described below to simulate distribution shifts. Since only $\{(\mathbf{Z}^{(q)}, \boldsymbol{\epsilon}^{(q)})\}_{q=0}^Q$ in Assumption 1 are subject to randomness in our setup, all other parameters are generated with a fixed seed across the 50 simulations to ensure their deterministic nature; see Appendix F.1 for details.

**Model shift.** Let $\boldsymbol{\beta}^{(0)} = S^{1/2}\mathbf{u}^{(0)}$ and $\boldsymbol{\delta}^{(q)} = (\mathrm{SSR}_q \cdot S)^{1/2}\mathbf{u}^{(q)}$ for $S > 0$ and $q \in [Q]$, where each $\mathbf{u}^{(q)}$ is randomly generated from the $p$-dimensional unit sphere $\mathbb{S}^{p-1}$ without any structural assumption on the coefficients, e.g., sparsity. We write $\mathbf{SSR} = (\mathrm{SSR}_1, \mathrm{SSR}_2, \ldots, \mathrm{SSR}_Q)$.

**Covariate shift.** Let $\mathbf{X}^{(q)} = \mathbf{Z}^{(q)}(\boldsymbol{\Sigma}^{(q)})^{1/2}$ where $\{\mathbf{Z}^{(q)}\}_{q=0}^Q$ are mutually independent and have i.i.d. standard Gaussian entries. In Section 5.1 with $Q = 3$ sources, the "benign" spectral matrices $\boldsymbol{\Lambda}^{(q)} = \mathrm{Diag}\big(\lambda_1^{(q)}, \ldots, \lambda_p^{(q)}\big)$ are as follows: $\lambda_j^{(0)} = 15j^{-1}\log^{-\beta}\left(\frac{(j+1)e}{2}\right)$; $\lambda_j^{(1)} = 15j^{-\gamma}$; $\lambda_j^{(2)} = 15j^{-(1+\log(n_2)/n_2)}$; $\lambda_j^{(3)} = 15\left[\mathbb{1}(j = 1) + \mathbb{1}(j > 1)\varepsilon\frac{1+\theta^2-2\theta\cos\left(j\pi/(p+1)\right)}{1+\theta^2-2\theta\cos\left(\pi/(p+1)\right)}\right]$, where we set [4] $\beta = 3/2$, $\gamma = \theta = 1/2$, and $\varepsilon = 10^{-5}$. While we always fix the target covariance diagonal so that $\boldsymbol{\Sigma}^{(0)} = \boldsymbol{\Lambda}^{(0)}$, under covariate shift, we assign for each source covariance the spectral decomposition $\boldsymbol{\Sigma}^{(q)} = \mathbf{V}^{(q)}\boldsymbol{\Lambda}^{(q)}\mathbf{V}^{(q)\top}$, where $\{\mathbf{V}^{(q)}\}_{q=1}^3$ are independently sampled from the orthogonal group $\mathcal{O}_p := \{\mathbf{Q} \in \mathbb{R}^{p\times p} : \mathbf{Q}^\top\mathbf{Q} = \mathbf{Q}\mathbf{Q}^\top = \mathbf{I}_p\}$. By doing so, we test the robustness of our methods to covariate shift without simultaneous diagonalizability, as discussed prior to Assumption 1. We evaluate the effect of free-lunch covariate shift (Corollary 2) by adjusting the source covariances by $\mathbf{V}^{(q)}(\alpha\boldsymbol{\Lambda}^{(q)})\mathbf{V}^{(q)\top}$ with $\alpha > 1$, so we do not alter any source eigendirection to align with the target. In Section 5.2, we set $Q = 2$ and $\boldsymbol{\Sigma}^{(0)} = \boldsymbol{\Sigma}^{(1)} = \boldsymbol{\Sigma}^{(2)} = \mathbf{I}_p$, and under free-lunch covariate shift, $\boldsymbol{\Sigma}^{(1)} = \boldsymbol{\Sigma}^{(2)} = \alpha\mathbf{I}_p$ with $\alpha > 1$.

## 5.1 Benign Overfitting Experiments

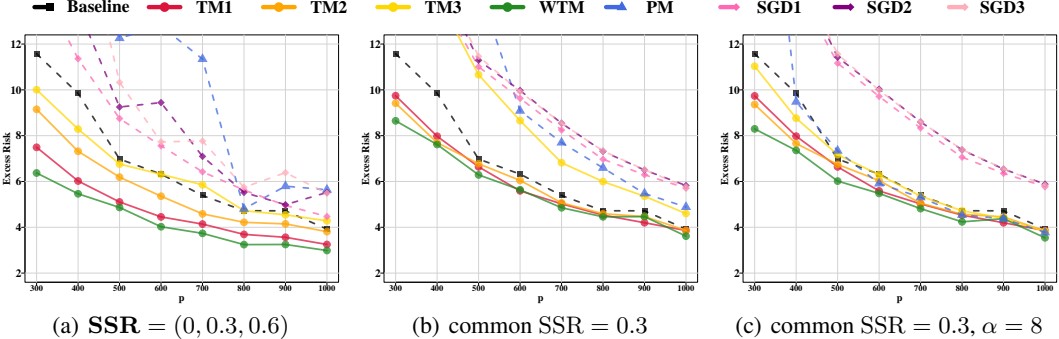

(a) $\mathbf{SSR} = (0, 0.3, 0.6)$  (b) common SSR $= 0.3$  (c) common SSR $= 0.3$, $\alpha = 8$

Figure 1: Let $n_0 = 25$ and $n_1 = n_2 = n_3 = 75$ for overfitting with $S = 500$. Figures (b) and (c) incorporate covariate shifts as detailed, with (c) additionally benefiting from the "free-lunch" effect.

---

[4]According to Bartlett et al. [5], $\boldsymbol{\Lambda}^{(0)}$ and $\boldsymbol{\Lambda}^{(1)}$ are benign if and only if $\beta > 1$ and $\gamma \in (0, 1)$, and $\boldsymbol{\Lambda}^{(3)}$ is benign if and only if $\theta < 1$, $p \gg n_3$, and $p\varepsilon \ll n_3$.

While $\hat{\boldsymbol{\beta}}_{\text{PM}}$ can outperform our estimates when it only learns source data without any distribution shift from the target (see Appendix E.1), the aggregate impact of distribution shifts for $\hat{\boldsymbol{\beta}}_{\text{PM}}$ is catastrophic throughout Fig. 1. On the contrary, in Fig. 1.(a), each $\hat{\boldsymbol{\beta}}_{\text{TM}}^{(q)}$ remains robust, outperforming the target-only MNI even under severe model shift (SSR $= 0.6$). Notably, the ensemble $\hat{\boldsymbol{\beta}}_{\text{WTM}}$ outperforms each individual $\hat{\boldsymbol{\beta}}_{\text{TM}}^{(q)}$, validating our data-adaptive weighting scheme based on CV losses.

Figures 1.(b) and 1.(c) additionally introduce covariate shift in the broadest sense, as not only the spectrum of each $\boldsymbol{\Sigma}^{(q)}$ differs from that of $\boldsymbol{\Sigma}^{(0)}$, but also all eigenvectors of $\boldsymbol{\Sigma}^{(q)}$ arbitrarily misalign with those of $\boldsymbol{\Sigma}^{(0)}$. As a result, $\hat{\boldsymbol{\beta}}_{\text{TM}}^{(3)}$ evidently suffers negative transfer in Fig. 1.(b); nevertheless, $\beta_{\text{WTM}}$ efficiently leverages only *informative* TM estimates by filtering out $\hat{\boldsymbol{\beta}}_{\text{TM}}^{(3)}$, consistently outperforming all competitors. When the *free-lunch* factor $\alpha > 1$ is additionally applied in Fig. 1.(c), $\hat{\boldsymbol{\beta}}_{\text{TM}}^{(3)}$ achieves significantly faster convergence and now performs comparably to the target-only MNI.

## 5.2 Harmless Interpolation Experiments

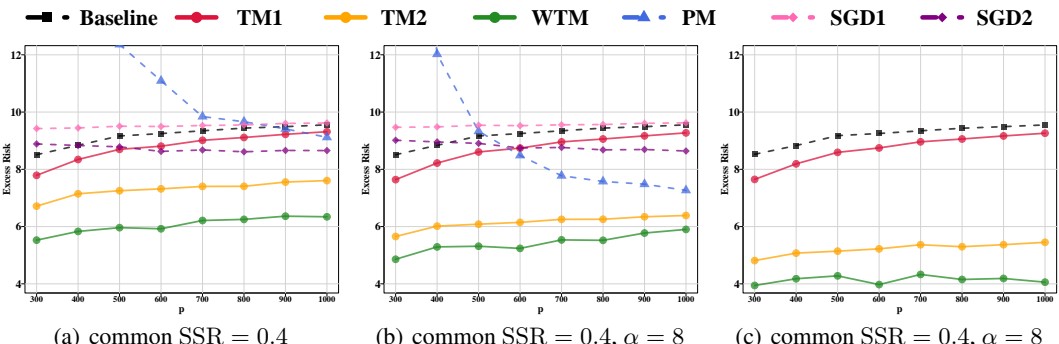

Figure 2: Let $n_0 = n_1 = 50$, and $n_2 = \lfloor n_2^* \rfloor$, the optimal transfer size for $\hat{\boldsymbol{\beta}}_{\text{TM}}^{(2)}$, with $S = 10$. Figure (c) adjusts $n_2^*$ in Corollary 1 via the modified signal-to-noise ratio $\text{SNR}_\alpha := \alpha \|\boldsymbol{\beta}^{(0)}\|^2 / \sigma^2$, which allows $\hat{\boldsymbol{\beta}}_{\text{TM}}^{(2)}$ to leverage even more source samples than the original $\lfloor n_2^* \rfloor$ in (a) and (b).

In Fig. 2, the target-only MNI with isotropic covariance demonstrates *harmless interpolation* [45], with its excess risk steadily converging to $\|\boldsymbol{\beta}^{(0)}\|^2 = 10$. Therefore, the primary evaluation is whether each transfer task can prevent, or at least delay, its excess risk from converging to 10. In Fig. 2.(a), despite considerable model shift (SSR $= 0.4$) and a sub-optimal transfer size, $\hat{\boldsymbol{\beta}}_{\text{TM}}^{(1)}$ slightly outperforms $\hat{\boldsymbol{\beta}}_{\text{M}}^{(0)}$. While $\hat{\boldsymbol{\beta}}_{\text{PM}}$ and SGD$_2$ eventually surpass the baseline as $p$ grows, they still lag behind $\hat{\boldsymbol{\beta}}_{\text{TM}}^{(2)}$ by a significant margin. Notably, even though $\hat{\boldsymbol{\beta}}_{\text{TM}}^{(1)}$ performs worse than $\hat{\boldsymbol{\beta}}_{\text{TM}}^{(2)}$ (while still inducing positive transfer), their ensemble $\hat{\boldsymbol{\beta}}_{\text{WTM}}$ substantially further reduces the excess risk.

Our methods benefit from the free-lunch covariate shift in Fig. 2.(b). However, in this scenario, the transfer size $\lfloor n_2^* \rfloor$ is no longer optimal because the variance inflation of TM is reduced by a factor of $\alpha$, essentially reducing the source noise level from $\sigma_2^2 = \sigma^2$ to $\sigma_2^2 = \sigma^2 / \alpha$. Thus, transferring even more than $\lfloor n_2^* \rfloor$ samples becomes advantageous. Taking this into account, we use $\text{SNR}_\alpha := \alpha \|\boldsymbol{\beta}^{(0)}\|^2 / \sigma^2$, which results in a new optimal transfer size that depends on $\text{SNR}_\alpha$ and is larger than $\lfloor n_2^* \rfloor$ used in Fig. 2.(a)-(b). By incorporating this new optimum, $\hat{\boldsymbol{\beta}}_{\text{TM}}^{(2)}$ and $\hat{\boldsymbol{\beta}}_{\text{WTM}}$ in Fig. 2.(c) further outperform their counterparts in Fig. 2.(b). We present in Appendix E.2 additional experiment results under model shift with isotropic covariates, where the target sample size $n_0$ is optimized for $\hat{\boldsymbol{\beta}}_{\text{PM}}$ as identified by Song et al. [56]. To enhance the clarity of plotted figures, we separately report in Appendix F.2 the empirical mean and standard deviation of excess risks computed over the 50 independent simulations.

## 6 Conclusion and Future Work

We propose a novel two-step Transfer MNI and a data-adaptive ensemble aggregating multiple *informative* Transfer MNIs. These methods address the posed question highlighting the underexplored nature of transfer learning for MNIs, supported by both theoretical characterizations and numerical

validations under various distribution shift conditions. We further identify *free-lunch* covariate shift regimes for Transfer MNI where the trade-off between bias reduction and variance inflation is neutralized, allowing us to "cherry-pick" the benefit of knowledge transfer at limited cost.

**Consistency of informative source detection.** Beyond the scope of this paper, we aim to provide a theoretical foundation for the empirical success of the WTM estimate prominently demonstrated throughout our experiments. Specifically, we aim to establish the consistency of informative source detection via cross-validation in Algorithm 1 by proving that the event $\mathcal{I} = \widehat{\mathcal{I}}$ holds with high probability, where $\mathcal{I}$ is the oracle set in (8) and $\widehat{\mathcal{I}}$ is its CV-driven estimate in (9). This agenda connects to the literature on source transferability detection, a pivotal aspect of transfer learning research. While classification settings are well studied [3, 47, 58, 22], less is known for regression. Recently, Nguyen et al. [48] introduced linear MSE as a regression transferability metric, and Tian and Feng [60] established cross-validation consistency for transfer generalized linear models. However, both analyses rely on explicit regularization, whereas our setting features benign overfitting with implicit regularization. Establishing consistency of transferability detection in this regime remains open and would meaningfully advance the relevant literature.

**Extension to minimum-RKHS-norm interpolator.** Another promising direction is to analyze the extension of Transfer MNI from finite-dimensional linear regression to nonlinear, infinite-dimensional regression in a reproducing kernel Hilbert space (RKHS) [53] via the minimum-RKHS-norm interpolator, including both fixed kernels and network-induced kernels such as the Neural Tangent Kernel (NTK) [23]. Extensive literature has explored benign overfitting in single-task cases; see Appendix A for a detailed review. However, to the best of our knowledge, transfer learning for the minimum-RKHS-norm interpolator beyond OOD settings still remains underexplored.

For the RKHS $\mathcal{H} \equiv \mathcal{H}_K$ associated with a positive semi-definite kernel $K : \mathbb{X} \times \mathbb{X} \to \mathbb{R}$ and norm $\|\cdot\|_{\mathcal{H}}$, let $\hat{f}_{\mathrm{M}}^{(0)}$ and $\hat{f}_{\mathrm{M}}^{(q)}$ be the minimum-RKHS-norm interpolators trained on the target and $q$-th source datasets, respectively; their analytical forms with empirical kernel (Gram) matrices $\mathbf{K}^{(q)} \in \mathbb{R}^{n_q \times n_q}$ for $q \in [Q]_0$ are well-known, where the $(i, j)$-th entry is $\mathbf{K}_{ij}^{(q)} = K(\mathbf{x}_i^{(q)}, \mathbf{x}_j^{(q)})$. This naturally motivates the RKHS analogue of Transfer MNI:

$$\hat{f}_{\mathrm{TM}}^{(q)} := \arg\min_{f \in \mathcal{H}} \left\{ \left\| f - \hat{f}_{\mathrm{M}}^{(q)} \right\|_{\mathcal{H}} : f(\mathbf{x}_i^{(0)}) = y_i^{(0)}, \ \forall i \in [n_0] \right\}, \quad q \in [Q],$$

i.e., $\hat{f}_{\mathrm{TM}}^{(q)}$ interpolates the target dataset $\{(\mathbf{x}_i^{(0)}, y_i^{(0)})\}_{i=1}^{n_0}$ while minimizing the RKHS-norm distance from the pre-trained $\hat{f}_{\mathrm{M}}^{(q)}$. With the target evaluation operator $E_0 : \mathcal{H} \to \mathbb{R}^{n_0}$ and its adjoint $E_0^* : \mathbb{R}^{n_0} \to \mathcal{H}$ such that $(E_0 f)_i = f(\mathbf{x}_i^{(0)})$, $E_0 E_0^* = \mathbf{K}^{(0)}$, and $\hat{f}_{\mathrm{M}}^{(0)} = E_0^* \mathbf{K}^{(0)\dagger} \mathbf{y}^{(0)}$, the projection operator $P_0$ onto the target-induced span $\mathcal{S}_{0,\mathcal{H}} := \mathrm{span}\{K(\cdot, \mathbf{x}_i^{(0)})\}_{i=1}^{n_0}$ is given by $P_0 := E_0^* \mathbf{K}^{(0)\dagger} E_0$. The operator acts as the RKHS counterpart to the projection matrix $\mathbf{H}^{(0)}$ for linear regression in that $P_0^* = P_0$ and $P_0^2 = P_0$ (i.e., self-adjoint and idempotent). Hence, the analytical form of Transfer MNI in Equation (3) systematically extends to provide the unique closed form of $\hat{f}_{\mathrm{TM}}^{(q)}$:

$$\hat{f}_{\mathrm{TM}}^{(q)} = \hat{f}_{\mathrm{M}}^{(0)} + (I_{\mathcal{H}} - P_0) \hat{f}_{\mathrm{M}}^{(q)},$$

where $I_{\mathcal{H}}$ is the identity operator. Indeed, the RKHS TM estimate $\hat{f}_{\mathrm{TM}}^{(q)}$ fully inherits the "retain-plus-transfer" mechanism in (4): it matches $\hat{f}_{\mathrm{M}}^{(0)}$ on $\mathcal{S}_{0,\mathcal{H}}$ and the fine-tuning step transfers knowledge learned by $\hat{f}_{\mathrm{M}}^{(q)}$ only into the complement space, i.e., $(I_{\mathcal{H}} - P_0) \hat{f}_{\mathrm{TM}}^{(q)} = (I_{\mathcal{H}} - P_0) \hat{f}_{\mathrm{M}}^{(q)}$.

Given the above formulation, it arises as an important direction to analyze the generalization gains of RKHS Transfer MNI with respect to factors that govern benign overfitting in an RKHS-such as the ground truth function class, kernel type, ambient input dimension, and sample size-as well as distribution shifts in the ground truth function and covariates between target and source tasks. We anticipate that our proposed scheme can serve as a versatile baseline for further studies on transfer learning with *implicitly regularized* minimum-norm interpolators that span a broad class of norms and, on a separate axis, extend beyond linear models, encompassing fixed RKHS kernels, NTK-type network-induced kernels, and data-adaptive kernels from deep representation learning.

## Acknowledgment

The authors are grateful to the anonymous reviewers for their constructive comments and suggestions, which helped improve the quality of this paper. Yeichan Kim and Ilmun Kim acknowledge support from the Basic Science Research Program through the National Research Foundation of Korea (NRF) funded by the Ministry of Education (2022R1A4A1033384), and the Korea government (MSIT) RS-2023-00211073. Seyoung Park's work was supported by the National Research Foundation of Korea (NRF) grant funded by the MSIT (No. RS-2025-00517793) and by the Yonsei University Research Fund of 2025-22-0071.

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

# A    Additional Literature Review

**Minimum-$\ell_p$-norm interpolators with $p \neq 2$.** For the minimum-$\ell_1$-norm interpolator, i.e., Basis Pursuit (BP) [11], its consistency hinges on the ambient input dimension $d$, sample size $n$, and model sparsity level $s$. Li and Wei [36] analyzed the asymptotic excess risk as $d/n \to \gamma \in (0, \infty)$ with isotropic design and linear sparsity, illustrating multiple descent phases with increasing degree of overparameterization. Wang et al. [65] demonstrated matching excess risk bounds of order $\sigma^2 / \log(d/n)$ for noisy BP with isotropic design and noise variance $\sigma^2$, provided $s \lesssim n / \log(d/n)^5$ and $n \lesssim d \lesssim \exp(n^{1/5})$. Complementary to these results, Donhauser et al. [15] showed an interpolation-specific bias-variance tension across $\ell_p$: interpolation achieves fast polynomial rates near $1/n$ for $p > 1$, but only logarithmic rates for $p = 1$. As an instructive counterpoint for BP, Chatterji and Long [10] proved lower bounds under Gaussian design, showing that the excess risk of BP can converge exponentially more slowly than that of the minimum-$\ell_2$-norm interpolator even when the ground truth is sparse.

A separate, algorithm-agnostic line controls the population error of *any* interpolator via uniform convergence: Koehler et al. [29] analyzed generalization in terms of Gaussian width and, instantiating the result on the $\ell_1$ simplex, obtained BP consistency. Beyond Hilbert-focused arguments, Chinot et al. [13] developed a robustness framework for minimum-norm interpolators with potentially adversarial errors, where their bounds track localized complexity, the norm of interpolated noise, and the subdifferential at the ground truth. Finally, Kur et al. [31] placed minimum-norm interpolators in general Banach spaces: under 2-uniform convexity, the bias is controlled by Gaussian complexity, while under cotype 2, they established a reverse Efron–Stein variance lower bound and sharpness of $\ell_p$ regression with $p \in [1, 2]$.

**Minimum-RKHS-norm interpolator.** Within a reproducing kernel Hilbert space (RKHS) [53], a line of recent works has examined when the minimum-RKHS-norm interpolator, often called kernel *ridgeless* regression (KRR)[5], generalizes well. Liang and Rakhlin [37] attributed implicit regularization in an RKHS to high input dimension, kernel curvature, and favorable spectral decays and provided out-of-sample guarantees under such conditions.

In fixed dimension, interpolation can fundamentally fail: Rakhlin and Zhai [52] showed that Laplace-kernel-based KRR is inconsistent for any bandwidth selection, making consistency a high-dimensional phenomenon. Refining this fixed-$d$ paradigm, Haas et al. [16] proved that for "smooth" kernels, benign overfitting is impossible in fixed $d$, yet "spiky-smooth" kernels (with large derivatives) can achieve rate-optimal benign overfitting; they further transferred the mechanism to wide networks by modifying activations. For the widely used Gaussian kernel, Medvedev et al. [43] showed that in fixed $d$, KRR is never consistent even when the bandwidth is tuned and, with sufficiently large noise, is often worse than the null predictor; with increasing $d$, they tracked transitions among benign, tempered, and catastrophic regimes, where a sub-polynomial-$d$ example typically exhibits benign overfitting.

In high-dimensional regimes, Donhauser et al. [14] identified a structural barrier for a broad class of rotationally invariant kernels (RBF, inner product, and fully connected NTKs): KRR is consistent only for low-degree polynomial targets, so spectral decay alone does not guarantee benign overfitting. Finally, Mallinar et al. [41] proposed a unifying taxonomy of *benign*, *tempered*, and *catastrophic* overfitting linked to the kernel eigenspectrum and ridge regularization parameter $\lambda \geq 0$: overfitting is benign with a positive $\lambda > 0$, whereas for KRR with $\lambda = 0$, it depends on spectral decay rates; Laplace kernels and ReLU NTKs typically lay in the tempered regime.

---

[5]While the acronym KRR often refers to kernel ridge regression with a regularization parameter $\lambda > 0$ in the literature, we use KRR to denote the ridgeless minimum-norm interpolator with $\lambda = 0$.

# B  Preliminary Lemmas

## B.1  Matrix Properties

**Lemma 2** (Expectation of quadratic forms). *Let $\hat{\boldsymbol{\theta}} \in \mathbb{R}^p$ be a random vector with mean $\mathbb{E}\big[\hat{\boldsymbol{\theta}}\big]$ and covariance $\mathrm{Cov}\big(\hat{\boldsymbol{\theta}}\big)$, and $\mathbf{M} \in \mathbb{R}^{p \times p}$ be a deterministic (or conditioned) symmetric matrix. The expectation of the quadratic form $\hat{\boldsymbol{\theta}}^\top \mathbf{M} \hat{\boldsymbol{\theta}}$ is given by*

$$\mathbb{E}\big[\hat{\boldsymbol{\theta}}^\top \mathbf{M} \hat{\boldsymbol{\theta}}\big] = \mathbb{E}\big[\hat{\boldsymbol{\theta}}\big]^\top \mathbf{M} \mathbb{E}\big[\hat{\boldsymbol{\theta}}\big] + \mathrm{Tr}\Big(\mathbf{M}\mathrm{Cov}\big(\hat{\boldsymbol{\theta}}\big)\Big).$$

*Proof.* Since trace is invariant to cyclic permutation, we have

$$
\begin{aligned}
\mathbb{E}\big[\hat{\boldsymbol{\theta}}^\top \mathbf{M} \hat{\boldsymbol{\theta}}\big] = \mathbb{E}\Big[\mathrm{Tr}(\hat{\boldsymbol{\theta}}^\top \mathbf{M} \hat{\boldsymbol{\theta}})\Big] &= \mathrm{Tr}\Big(\mathbf{M}\mathbb{E}[\hat{\boldsymbol{\theta}}\hat{\boldsymbol{\theta}}^\top]\Big) \\
&= \mathrm{Tr}\left(\mathbf{M}\Big(\mathrm{Cov}(\hat{\boldsymbol{\theta}}) + \mathbb{E}[\hat{\boldsymbol{\theta}}]\mathbb{E}[\hat{\boldsymbol{\theta}}]^\top\Big)\right) \\
&= \mathrm{Tr}\left(\mathbf{M}\mathrm{Cov}(\hat{\boldsymbol{\theta}})\right) + \mathrm{Tr}\left(\mathbf{M}\mathbb{E}[\hat{\boldsymbol{\theta}}]\mathbb{E}[\hat{\boldsymbol{\theta}}]^\top\right) \\
&= \mathrm{Tr}\left(\mathbf{M}\mathrm{Cov}(\hat{\boldsymbol{\theta}})\right) + \mathbb{E}[\hat{\boldsymbol{\theta}}]^\top \mathbf{M}\mathbb{E}[\hat{\boldsymbol{\theta}}].
\end{aligned}
$$

$\square$

The following lemma is an adjustment of Lemma 3 in Ju et al. [26] and Proposition 3 and Lemma 16 in Ju et al. [25], tailored to align with our setting of well-specified linear models. The result follows from the rotational invariance of Gaussian distribution.

**Lemma 3** (Standard Gaussian orthogonal projection). *Let $\mathbf{X}^{(q)} \in \mathbb{R}^{n_q \times p}$ have i.i.d. standard Gaussian entries with $p > n_q + 1$. Since the matrix has full row rank almost surely, the orthogonal projection onto the $n_q$-dimensional row space of $\mathbf{X}^{(q)}$ is given by*

$$\mathbf{H}^{(q)} = \mathbf{X}^{(q)\top}\big(\mathbf{X}^{(q)}\mathbf{X}^{(q)\top}\big)^{-1}\mathbf{X}^{(q)},$$

*and $\mathbf{I}_p - \mathbf{H}^{(q)}$ is the orthogonal projection onto the $(p-n_q)$-dimensional null space, with expectations*

$$\mathbb{E}\mathbf{H}^{(q)} = \frac{n_q}{p}\mathbf{I}_p, \qquad \mathbb{E}\big[\mathbf{I}_p - \mathbf{H}^{(q)}\big] = \Big(1 - \frac{n_q}{p}\Big)\mathbf{I}_p.$$

*Thus, for any fixed vector $\mathbf{a} \in \mathbb{R}^p$, we have*

$$\mathbb{E}\big\|\mathbf{H}^{(q)}\mathbf{a}\big\|^2 = \frac{n_q}{p}\|\mathbf{a}\|^2, \qquad \mathbb{E}\big\|\big(\mathbf{I}_p - \mathbf{H}^{(q)}\big)\mathbf{a}\big\|^2 = \Big(1 - \frac{n_q}{p}\Big)\|\mathbf{a}\|^2.$$

The following lemma can be generalized to a broad class of norms, with our focus on the norms of a real matrix.

**Lemma 4** (Hölder's inequality for Schatten $p$-norm). *Given two real matrices $\mathbf{A}$ and $\mathbf{B}$, Hölder's inequality yields*

$$\left|\mathrm{Tr}\big(\mathbf{A}^\top \mathbf{B}\big)\right| \le \|\mathbf{A}\|_p \|\mathbf{B}\|_q,$$

*where $\frac{1}{p} + \frac{1}{q} = 1$ and $\|\mathbf{A}\|_p$ is the Schatten p-norm of $\mathbf{A}$ defined as*

$$\|\mathbf{A}\|_p := \Big(\mathrm{Tr}\big(|\mathbf{A}|^p\big)\Big)^{1/p} : \qquad |\mathbf{A}| := \big(\mathbf{A}^\top \mathbf{A}\big)^{1/2}.$$

*The above inequality also holds when $p = 1$ and $q = \infty$, where the infinity norm corresponds to the operator norm, i.e,*

$$\|\mathbf{B}\|_\infty = \|\mathbf{B}\|.$$

*In particular, if both $\mathbf{A}$ and $\mathbf{B}$ are symmetric and positive semi-definite, we have*

$$\mathrm{Tr}(\mathbf{A}\mathbf{B}) \le \mathrm{Tr}(\mathbf{A})\|\mathbf{B}\|.$$

**Lemma 5** (Sherman-Morrison-Woodbury formula). *Let $\mathbf{A} \in \mathbb{R}^{n \times n}$ be an invertible matrix and $\mathbf{Z} \in \mathbb{R}^{n \times k}$ be a matrix with $k < n$ such that $\mathbf{Z}\mathbf{Z}^\top + \mathbf{A}$ is invertible. By the Sherman-Morrison-Woodbury formula, it follows that*

$$\mathbf{Z}^\top\left(\mathbf{Z}\mathbf{Z}^\top + \mathbf{A}\right)^{-1}\mathbf{Z} = \mathbf{Z}^\top\left(\mathbf{A}^{-1} - \mathbf{A}^{-1}\mathbf{Z}(\mathbf{I}_k + \mathbf{Z}^\top\mathbf{A}^{-1}\mathbf{Z})^{-1}\mathbf{Z}^\top\mathbf{A}^{-1}\right)\mathbf{Z}$$

$$= \mathbf{Z}^\top\mathbf{A}^{-1}\mathbf{Z}\left(\mathbf{I}_k + \mathbf{Z}^\top\mathbf{A}^{-1}\mathbf{Z}\right)^{-1}$$

*and*

$$\mathbf{Z}^\top\left(\mathbf{Z}\mathbf{Z}^\top + \mathbf{A}\right)^{-2}\mathbf{Z} = \left(\mathbf{I}_k + \mathbf{Z}^\top\mathbf{A}^{-1}\mathbf{Z}\right)^{-1}\mathbf{Z}^\top\mathbf{A}^{-2}\mathbf{Z}\left(\mathbf{I}_k + \mathbf{Z}^\top\mathbf{A}^{-1}\mathbf{Z}\right)^{-1}.$$

## B.2 Concentration Inequalities

For simplicity, we present the subsequent lemmas using the target design matrix $\mathbf{X} \equiv \mathbf{X}^{(0)}$ and covariance $\mathbf{\Sigma} \equiv \mathbf{\Sigma}^{(0)}$, whose eigenvalues are denoted by

$$\lambda_1 \geq \lambda_2 \geq \ldots \geq \lambda_p > 0,$$

and assume without loss of generality that $\mathbf{X}$ has full row rank so that $\mathbf{X}\mathbf{X}^\top$ is invertible. We further define the target sample covariance $\hat{\mathbf{\Sigma}} = (1/n_0)\mathbf{X}^\top\mathbf{X}$, and the eigenvalues of a matrix $\mathbf{M} \in \mathbb{R}^{d \times d}$ are denoted by

$$\mu_1(\mathbf{M}) \geq \mu_2(\mathbf{M}) \geq \ldots \geq \mu_d(\mathbf{M}).$$

All subsequent results readily extend to the source counterparts $\mathbf{X}^{(q)}$, $\mathbf{\Sigma}^{(q)}$, $\lambda_j^{(q)}$, and $\hat{\mathbf{\Sigma}}^{(q)}$ under Assumptions 1 and 2.

The following lemma provides a high-probability upper bound on the operator norm of a sub-Gaussian sample covariance, characterized by the effective ranks in Definition 1.

**Lemma 6** (Exercise 9.2.5, Vershynin [63]). *Let Assumption 1 hold. There exists a constant $u > 0$ that depends only on the sub-Gaussian parameter $\nu_x$ in Assumption 1 such that with probability at least $1 - 2e^{-t}$ for $t \geq \log(2)$,*

$$\|\hat{\mathbf{\Sigma}} - \mathbf{\Sigma}\| \leq u\left(\sqrt{\frac{r_0(\mathbf{\Sigma}) + t}{n_0}} + \frac{r_0(\mathbf{\Sigma}) + t}{n_0}\right)\|\mathbf{\Sigma}\|.$$

Next, we restates some key results from Bartlett et al. [5] and Mallinar et al. [42] for ease on the reader and completeness. The central focus of their analysis is the normalized variance of target-only MNI $\mathcal{V}_{\mathrm{M}}^{(0)}$ in Equation (6), where

$$\mathcal{V}_{\mathrm{M}}^{(0)}/\sigma_0^2 = \frac{1}{n_0}\mathrm{Tr}\left(\hat{\mathbf{\Sigma}}^\dagger\mathbf{\Sigma}\right) = \mathrm{Tr}\left(\mathbf{X}^\top(\mathbf{X}\mathbf{X}^\top)^{-2}\mathbf{X}\mathbf{\Sigma}\right). \tag{10}$$

Following Bartlett et al. [5], we fix $\mathbf{X} = \mathbf{Z}\mathbf{\Sigma}^{1/2}$ in the eigenbasis of $\mathbf{\Sigma}$, where $\mathbf{Z} \equiv \mathbf{Z}^{(0)} \in \mathbb{R}^{n_0 \times p}$ is the sub-Gaussian whitened design matrix as specified in Assumption 1. Then, the gram matrix and its reduced-rank variants can be expressed as

$$\mathbf{A} := \mathbf{X}\mathbf{X}^\top = \sum_{j=1}^p \lambda_j \mathbf{z}_j\mathbf{z}_j^\top, \qquad \mathbf{A}_{-j} := \sum_{i \neq j}\lambda_i \mathbf{z}_i\mathbf{z}_i^\top, \qquad \mathbf{A}_k := \sum_{j > k}\lambda_j \mathbf{z}_j\mathbf{z}_j^\top, \tag{11}$$

where $\{\mathbf{z}_j\}_{j=1}^p \subset \mathbb{R}^{n_0}$ are i.i.d. columns of $\mathbf{Z}$; each $j$-th column $\mathbf{z}_j$ has i.i.d. $\nu_x$-sub-Gaussian mean-zero, unit-variance components. Consequently, the trace term in Equation (10) can be written as

$$\mathrm{Tr}\left(\mathbf{X}^\top(\mathbf{X}\mathbf{X}^\top)^{-2}\mathbf{X}\mathbf{\Sigma}\right) = \sum_{j=1}^p \lambda_j^2 \mathbf{z}_j^\top\left(\sum_{i=1}^p \lambda_i \mathbf{z}_i\mathbf{z}_i^\top\right)^{-2}\mathbf{z}_j$$

$$= \sum_{j=1}^p \lambda_j^2 \mathbf{z}_j^\top\left(\lambda_j \mathbf{z}_j\mathbf{z}_j^\top + \mathbf{A}_{-j}\right)^{-2}\mathbf{z}_j$$

$$= \sum_{j=1}^p \frac{\lambda_j^2 \mathbf{z}_j^\top\mathbf{A}_{-j}^{-2}\mathbf{z}_j}{(1 + \lambda_j \mathbf{z}_j^\top\mathbf{A}_{-j}^{-1}\mathbf{z}_j)^2}, \tag{12}$$

where the last equality holds by Lemma 5. Opening the discussion, the following lemma gives concentration of the spectrum of the matrices in (11).

**Lemma 7** (Lemma 5, Bartlett et al. [5])**.** *Let Assumption 1 hold. There exist universal constants $b, c \geq 1$ such that for any $k \geq 0$, with probability at least $1 - 2e^{-n/c}$,*

1. *for all $j \geq 1$,*

$$\mu_{k+1}(\mathbf{A}_{-j}) \leq \mu_{k+1}(\mathbf{A}) \leq \mu_1(\mathbf{A}_k) \leq c\left(\sum_{i>k} \lambda_i + n_0 \lambda_{k+1}\right),$$

2. *for all $1 \leq j \leq k$,*

$$\mu_{n_0}(\mathbf{A}) \geq \mu_{n_0}(\mathbf{A}_{-j}) \geq \mu_{n_0}(\mathbf{A}_k) \geq \frac{1}{c}\sum_{i>k} \lambda_i - cn_0 \lambda_{k+1},$$

3. *if $r_k(\mathbf{\Sigma}) \geq bn_0$, then*

$$\frac{1}{c}\lambda_{k+1} r_k(\mathbf{\Sigma}) \leq \mu_{n_0}(\mathbf{A}_k) \leq \mu_1(\mathbf{A}_k) \leq c\lambda_{k+1} r_k(\mathbf{\Sigma}).$$

### B.2.1 Upper Bound on Single-Task Variance

The following lemma provides high-probability bounds on the norm of random sub-Gaussian vector with independent components. The result is used to provide an upper bound on the trace term in Equation (12).

**Lemma 8** (Corollary 1, Bartlett et al. [5] and Corollary B.5, Mallinar et al. [42])**.** *There exists a universal constant $c > 0$ such that for a random vector $\mathbf{z} \in \mathbb{R}^{n_0}$ with independent $\nu_x$-sub-Gaussian mean-zero, unit-variance components, any random subspace $\mathcal{G} \subset \mathbb{R}^{n_0}$ of co-dimension $k$ that is independent of $\mathbf{z}$, and any $t > 0$, with probability at least $1 - 3e^{-t}$,*

$$\|\mathbf{z}\|^2 \leq n_0 + c\nu_x^2(t + \sqrt{n_0 t}),$$
$$\|\mathbf{\Pi}_{\mathcal{G}}\mathbf{z}\|^2 \geq n_0 - c\nu_x^2(k + t + \sqrt{n_0 t}),$$

*where $\mathbf{\Pi}_{\mathcal{G}}$ is the orthogonal projection onto $\mathcal{G}$. Furthermore, let $t \in (0, n_0/c_0]$ and $k \in [0, n_0/c_1]$ for $c_1 > c_0$ with sufficiently large $c_0$. Then, with probability at least $1 - 3e^{-t}$,*

$$\|\mathbf{z}\|^2 \leq c_2 n_0,$$
$$\|\mathbf{\Pi}_{\mathcal{G}}\mathbf{z}\|^2 \geq n_0/c_3,$$

*where $c_2$ and $c_3$ only depend on $c$, $c_0$, and $\nu_x$. For each $\mathbf{z}_j \in \mathbb{R}^{n_0}$ and the corresponding subspace $\mathcal{G}_j$ defined analogously to $\mathbf{z}_j$ and $\mathcal{G}_j$, define the event*

$$E_j = \left\{\|\mathbf{z}_j\|^2 \leq c_2 n_0 \ \cap \ \|\mathbf{\Pi}_{\mathcal{G}_j}\mathbf{z}_j\|^2 \geq n_0/c_3\right\}, \quad j \in [\ell].$$

*Then, the union bound argument implies*

$$\mathbb{P}\left(\bigcup_{j=1}^{\ell}(E_j)^c\right) \leq \sum_{j=1}^{\ell} \mathbb{P}\left((E_j)^c\right)$$

$$\leq \sum_{j=1}^{\ell} 3e^{-t} = 3\ell e^{-t}$$

$$\implies \mathbb{P}\left(\bigcap_{j=1}^{\ell} E_j\right) \geq 1 - 3\ell e^{-t} = 1 - 3e^{-\left(t - \log(\ell)\right)},$$

*which necessitates $0 < t - \log(\ell) \leq n/c_0$ to complete the bound. Since each event $E_j$ is defined for $t \in (0, n/c_0]$, by taking $t = n/c_0$ pre-event and requiring that*

$$\log(\ell) \leq n/c_0 \iff \ell \leq e^{n/c_0},$$

*all events $\{E_j\}_{j=1}^{\ell}$ hold simultaneously with probability at least $1 - 3e^{n/c_0}$.*

We further introduce a lemma concerning the weighted sum of sub-exponential random variables.

**Lemma 9** (Lemma 7, Bartlett et al. [5]). *Suppose that $\{\lambda_j\}_{j\in\mathbb{N}}$ is a non-increasing sequence of non-negative numbers such that $\sum_{j\in\mathbb{N}} \lambda_j < \infty$ (as is the case of the eigenvalues of $\Sigma$) and that $\{\xi_j\}_{j\in\mathbb{N}}$ are independent centered $\nu_x^2$-sub-exponential random variables. Then, there exists a universal constant $a > 0$ such that for any $t > 0$, with probability at least $1 - 2e^{-t}$,*

$$\left|\sum_j \lambda_j \xi_j\right| \le a\nu_x^2 \max\left(t\lambda_1, \sqrt{t\sum_j \lambda_j^2}\right).$$

We are now ready to provide a high-probability upper bound on the variance of single-task MNI.

**Lemma 10** (Lemma 6, Bartlett et al. [5]). *Let Assumption 1 hold. There exist universal constants $b, c \ge 1$ such that if*

$$0 \le k \le n_0/c, \quad r_k(\Sigma) \ge bn_0, \quad \ell \le k,$$

*then with probability at least $1 - 7e^{-n_0/c}$,*

$$\frac{\sigma_0^2}{n_0}\mathrm{Tr}\left(\hat{\Sigma}^\dagger \Sigma\right) \le c\sigma_0^2\left(\frac{\ell}{n_0} + \frac{n_0\sum_{j>\ell}\lambda_j^2}{\left(\lambda_{k+1}r_k(\Sigma)\right)^2}\right).$$

*Proof.* It suffices to upper-bound the trace term in Equation (12), which is given by

$$\mathrm{Tr}\left(\mathbf{X}^\top(\mathbf{XX}^\top)^{-2}\mathbf{X}\Sigma\right) = \sum_{j=1}^{\ell} \frac{\lambda_j^2 \mathbf{z}_j^\top \mathbf{A}_{-j}^{-2} \mathbf{z}_j}{(1 + \lambda_j \mathbf{z}_j^\top \mathbf{A}_{-j}^{-1}\mathbf{z}_j)^2} + \sum_{j>\ell} \lambda_j^2 \mathbf{z}_j^\top \mathbf{A}^{-2}\mathbf{z}_j.$$

Fix $b, c_1 \ge 1$ as specified in Lemma 7. Then, with probability at least $1 - 2e^{-n/c_1}$, if $r_k(\Sigma) \ge bn_0$ then for all vectors in $\{\mathbf{z}_j\}_{j=1}^{\ell} \subset \mathbb{R}^{n_0}$, we have

$$\mathbf{z}_j^\top \mathbf{A}_{-j}^{-2}\mathbf{z}_j \le \mu_1(\mathbf{A}_{-j}^{-2})\|\mathbf{z}_j\|^2 \le \mu_{n_0}(\mathbf{A}_{-j})^{-2}\|\mathbf{z}_j\|^2 \le \frac{c_1^2\|\mathbf{z}_j\|^2}{\left(\lambda_{k+1}r_k(\Sigma)\right)^2},$$

and on the same high-probability event, we have simultaneously

$$\begin{aligned}
\mathbf{z}_j^\top \mathbf{A}_{-j}^{-1}\mathbf{z}_j &\ge (\mathbf{\Pi}_{\mathcal{G}_j}\mathbf{z}_j)^\top \mathbf{A}_{-j}^{-1}(\mathbf{\Pi}_{\mathcal{G}_j}\mathbf{z}_j) \\
&\ge \mu_{n_0}(\mathbf{A}_{-j}^{-1})\|\mathbf{\Pi}_{\mathcal{G}_j}\mathbf{z}_j\|^2 \\
&\ge \mu_{k+1}(\mathbf{A}_{-j})^{-1}\|\mathbf{\Pi}_{\mathcal{G}_j}\mathbf{z}_j\|^2 \\
&\ge \frac{\|\mathbf{\Pi}_{\mathcal{G}_j}\mathbf{z}_j\|^2}{c_1\lambda_{k+1}r_k(\Sigma)},
\end{aligned}$$

where $\mathbf{\Pi}_{\mathcal{G}_j}$ is the orthogonal projection onto the span of the bottom $(n_0 - k)$ eigenvectors of $\mathbf{A}_{-j}$, independent of $\mathbf{z}_j$. Therefore, for all $j \le \ell$, it follows that

$$\frac{\lambda_j^2 \mathbf{z}_j^\top \mathbf{A}_{-j}^{-2}\mathbf{z}_j}{(1 + \lambda_j \mathbf{z}_j^\top \mathbf{A}_{-j}^{-1}\mathbf{z}_j)^2} \le \frac{\mathbf{z}_j^\top \mathbf{A}_{-j}^{-2}\mathbf{z}_j}{(\mathbf{z}_j^\top \mathbf{A}_{-j}^{-1}\mathbf{z}_j)^2} \le c_1^4 \frac{\|\mathbf{z}_j\|^2}{\|\mathbf{\Pi}_{\mathcal{G}_j}\mathbf{z}_j\|^4}.$$

Here, we recall Lemma 8 with a union bound over $\ell$ events. Let $t \le n_0/c_0$ and $k \in [0, n_0/c]$ for sufficiently large $c_0 < c$, where $c$ is as specified in the lemma. Since $\ell \le k$, with probability at least $1 - 3e^{-n_0/c_0}$, the intersection of all $\ell$ events

$$\bigcap_{j=1}^{\ell}\left\{\|\mathbf{z}_j\|^2 \le c_2n_0 \ \cap \ \|\mathbf{\Pi}_{\mathcal{G}_j}\mathbf{z}_j\|^2 \ge n_0/c_3\right\}$$

hold for constants $c_2$ and $c_3$ that only depend on $\nu_x$, $c_0$, and $c$. Combining the results, with probability at least $1 - 5e^{-n_0/c_0}$ for some sufficiently large $c_0$, we have

$$\sum_{j=1}^{\ell} \frac{\lambda_j^2 \mathbf{z}_j^\top \mathbf{A}_{-j}^{-2}\mathbf{z}_j}{(1 + \lambda_j \mathbf{z}_j^\top \mathbf{A}_{-j}^{-1}\mathbf{z}_j)^2} \le c_4\frac{\ell}{n_0}.$$

Next, we consider the sum of the tail summands. On the same high-probability event in Lemma 7 we used, we have $\mu_{n_0}(\mathbf{A}_k) \geq \lambda_{k+1} r_k(\mathbf{\Sigma})/c_1$ if $r_k(\mathbf{\Sigma}) \geq bn_0$. Also, since $\mu_1(\mathbf{A}^{-2}) \leq \mu_{n_0}^{-2}(\mathbf{A})$, this implies that on the same event, if $r_k(\mathbf{\Sigma}) \geq bn_0$, then

$$\mathbf{z}_j^\top \mathbf{A}^{-2} \mathbf{z}_j \;\leq\; \mu_1(\mathbf{A}^{-2})\|\mathbf{z}_j\|^2 \;\leq\; \mu_{n_0}(\mathbf{A})^{-2}\|\mathbf{z}_j\|^2 \;\leq\; \mu_{n_0}(\mathbf{A}_k)^{-2}\|\mathbf{z}_j\|^2 \;\leq\; \frac{c_1^2}{(\lambda_{k+1} r_k(\mathbf{\Sigma}))^2}\|\mathbf{z}_j\|^2,$$

which implies

$$\sum_{j>\ell} \lambda_j^2 \mathbf{z}_j^\top \mathbf{A}^{-2} \mathbf{z}_j \;\leq\; \frac{c_1^2 \sum_{j>\ell} \lambda_j^2 \|\mathbf{z}_j\|^2}{(\lambda_{k+1} r_k(\mathbf{\Sigma}))^2}.$$

Notice that each $\|\mathbf{z}_j\|^2$ is a sum of $n_0$ independent $\nu_x^2$-sub-exponential random variables such that $\mathbb{E}\|\mathbf{z}_j\|^2 = n_0$. Therefore, by Lemma 9, there exists a universal constant $a > 0$ such that with probability at least $1 - 2e^{-t}$ for $t < n_0/c_0$,

$$\sum_{j>\ell} \lambda_j^2 \|\mathbf{z}_j\|^2 \leq n_0 \sum_{j>\ell} \lambda_j^2 + a\nu_x^2 \max\left(\lambda_{\ell+1}^2 t, \; \sqrt{tn_0 \sum_{j>\ell} \lambda_j^4}\right)$$

$$\leq n_0 \sum_{j>\ell} \lambda_j^2 + a\nu_x^2 \max\left(t \sum_{j>\ell} \lambda_j^2, \; \sqrt{tn_0} \sum_{j>\ell} \lambda_j^2\right)$$

$$\leq c_5 n_0 \sum_{j>\ell} \lambda_j^2$$

for some sufficiently large $c_5$. Combining the above results yields

$$\sum_{j>\ell} \lambda_j^2 \mathbf{z}_j^\top \mathbf{A}^{-2} \mathbf{z}_j \leq c_6 n_0 \frac{\sum_{j>\ell} \lambda_j^2}{(\lambda_{k+1} r_k(\mathbf{\Sigma}))^2}.$$

Thus, putting both summations together and taking $c > \max(c_0, c_4, c_6)$ complete the proof. $\qquad\square$

### B.2.2 Lower Bound on Single-Task Variance

To establish a lower bound on the trace term in Equation (12), we first present a preliminary result that extends a lower bound on individual random variables, each holding with equal probability, to a unified lower bound on their entire sum.

**Lemma 11** (Lemma 9, Bartlett et al. [5]). *Suppose $p \leq \infty$. Let $\{\eta_j\}_{j=1}^p$ be a sequence of non-negative random variables and $\{t_j\}_{j=1}^p$ be a sequence of non-negative real numbers (at least one of which is strictly positive) such that for some $\delta \in (0,1)$ and any $j \in [p]$, $\mathbb{P}(\eta_j > t_j) \geq 1 - \delta$. Then,*

$$\mathbb{P}\left(\sum_{j=1}^p \eta_j \geq \frac{1}{2} \sum_{j=1}^p t_j\right) \geq 1 - 2\delta.$$

Next lemma provides the lower bound by first establishing a general bound that holds regardless of $r_k(\mathbf{\Sigma})$ and then refining it for the case where $r_k(\mathbf{\Sigma})$ is at least $n_0$, using Lemma 11.

**Lemma 12** (Lemma 10, Bartlett et al. [5]). *Let Assumption 1 hold. There exist universal constants $b, c \geq 1$ such that if*

$$0 \leq k \leq n_0/c, \quad r_k(\mathbf{\Sigma}) \geq bn_0,$$

*then with probability at least $1 - 10e^{-n_0/c}$,*

$$\frac{\sigma_0^2}{n_0} \mathrm{Tr}\left(\hat{\mathbf{\Sigma}}^\dagger \mathbf{\Sigma}\right) \;\geq\; \frac{1}{cb^2} \sigma_0^2 \min_{\ell \leq k}\left(\frac{\ell}{n_0} + \frac{b^2 n_0 \sum_{j>\ell} \lambda_j^2}{\left(\lambda_{k+1} r_k(\mathbf{\Sigma})\right)^2}\right).$$

*Proof.* Fix $j \geq 1$ and $k \in [0, n_0/C]$. By Lemma 7, there exists a universal constant $C_1$ such that with probability at least $1 - 2e^{-n_0/C_1}$,

$$\mu_{k+1}(\mathbf{A}_{-j}) \leq C_1\left(\sum_{i>k} \lambda_i + n_0\lambda_{k+1}\right).$$

Let $\mathcal{G}_j$ be the span of the bottom $(n_0 - k)$ eigenvectors of $\mathbf{A}_{-j}$ and $\mathbf{\Pi}_{\mathcal{G}_j}$ be the projection onto the orthogonal complement of $\mathcal{G}_j$. Then, we have

$$\mathbf{z}_j^\top \mathbf{A}_{-j}^{-1} \mathbf{z}_j \geq (\mathbf{\Pi}_{\mathcal{G}_j} \mathbf{z}_j)^\top \mathbf{A}_{-j}^{-1} (\mathbf{\Pi}_{\mathcal{G}_j} \mathbf{z}_j) \geq \mu_{k+1}(\mathbf{A}_{-j})^{-1} \|\mathbf{\Pi}_{\mathcal{G}_j} \mathbf{z}_j\|^2 \geq \frac{\|\mathbf{\Pi}_{\mathcal{G}_j} \mathbf{z}_j\|^2}{C_1 \left( \sum_{i>k} \lambda_i + n_0 \lambda_{k+1} \right)}.$$

By Corollary 8, with probability at least $1 - 3e^{-t}$,

$$\|\mathbf{\Pi}_{\mathcal{G}_j} \mathbf{z}_j\|^2 \geq n_0/C_2,$$

provided that $t \leq n_0/C_0$ for some sufficiently large $C_0 < C$. Thus, with probability at least $1 - 5e^{-n_0/C_3}$,

$$\mathbf{z}_j^\top \mathbf{A}_{-j}^{-1} \mathbf{z}_j \geq \frac{n_0}{C_3 \left( \sum_{i>k} \lambda_i + n_0 \lambda_{k+1} \right)}$$

$$\implies 1 + \lambda_j \mathbf{z}_j^\top \mathbf{A}_{-j}^{-1} \mathbf{z}_j \leq \left( 1 + \frac{C_3 \left( \sum_{i>k} \lambda_i + n_0 \lambda_{k+1} \right)}{n_0 \lambda_j} \right) \lambda_j \mathbf{z}_j^\top \mathbf{A}_{-j}^{-1} \mathbf{z}_j,$$

and taking the squared reciprocals of both sides and then multiplying them by $\lambda_j^2 \mathbf{z}_j^\top \mathbf{A}_{-j}^{-2} \mathbf{z}_j$ yield

$$\frac{\lambda_j^2 \mathbf{z}_j^\top \mathbf{A}_{-j}^{-2} \mathbf{z}_j}{(1 + \lambda_j \mathbf{z}_j^\top \mathbf{A}_{-j}^{-1} \mathbf{z}_j)^2} \geq \left( 1 + \frac{C_3 \left( \sum_{i>k} \lambda_i + n_0 \lambda_{k+1} \right)}{n_0 \lambda_j} \right)^{-2} \frac{\mathbf{z}_j^\top \mathbf{A}_{-j}^{-2} \mathbf{z}_j}{(\mathbf{z}_j^\top \mathbf{A}_{-j}^{-1} \mathbf{z}_j)^2}.$$

Also, by the Cauchy-Schwarz inequality and the same high-probability event from Lemma 7, we have

$$\frac{\mathbf{z}_j^\top \mathbf{A}_{-j}^{-2} \mathbf{z}_j}{(\mathbf{z}_j^\top \mathbf{A}_{-j}^{-1} \mathbf{z}_j)^2} \geq \frac{\mathbf{z}_j^\top \mathbf{A}_{-j}^{-2} \mathbf{z}_j}{\|\mathbf{A}_{-j}^{-1} \mathbf{z}_j\|^2 \|\mathbf{z}_j\|^2} = \frac{1}{\|\mathbf{z}_j\|^2} \geq \frac{1}{C_4 n_0}.$$

Thus, taking $C$ sufficiently large, for any $j \geq 1$ and $k \in [0, n_0/C]$, with probability at least $1 - 5e^{-n/C}$,

$$\frac{\lambda_j^2 \mathbf{z}_j^\top \mathbf{A}_{-j}^{-2} \mathbf{z}_j}{(1 + \lambda_j \mathbf{z}_j^\top \mathbf{A}_{-j}^{-1} \mathbf{z}_j)^2} \geq \frac{1}{Cn_0} \left( 1 + \frac{\sum_{i>k} \lambda_i + n_0 \lambda_{k+1}}{n_0 \lambda_j} \right)^{-2},$$

which establishes a general lower bound. We refine it by establishing a lower bound on the summation over $j \in [p]$ as follows: by Lemma 11, with probability at least $1 - 10e^{-n_0/c_1}$,

$$\begin{aligned}
\mathrm{Tr}\left( \mathbf{X}^\top (\mathbf{X}\mathbf{X}^\top)^{-2} \mathbf{X}\boldsymbol{\Sigma} \right) &= \sum_{j=1}^{p} \frac{\lambda_j^2 \mathbf{z}_j^\top \mathbf{A}_{-j}^{-2} \mathbf{z}_j}{(1 + \lambda_j \mathbf{z}_j^\top \mathbf{A}_{-j}^{-1} \mathbf{z}_j)^2} \\
&\geq \frac{1}{c_1 n_0} \sum_{j=1}^{p} \left( 1 + \frac{\sum_{i>k} \lambda_i + n_0 \lambda_{k+1}}{n_0 \lambda_j} \right)^{-2} \\
&\geq \frac{1}{c_2 n_0} \sum_{j=1}^{p} \min\left( 1, \frac{n_0^2 \lambda_j^2}{(\sum_{i>k} \lambda_i)^2}, \frac{\lambda_j^2}{\lambda_{k+1}^2} \right) \\
&\geq \frac{1}{c_2 b^2 n_0} \sum_{j=1}^{p} \min\left( 1, \left( \frac{bn_0}{r_k(\boldsymbol{\Sigma})} \right)^2 \frac{\lambda_j^2}{\lambda_{k+1}^2}, \frac{\lambda_j^2}{\lambda_{k+1}^2} \right),
\end{aligned}$$

where $b \geq 1$ is as specified in Lemma 7. Thus, if $r_k(\boldsymbol{\Sigma}) \geq bn_0$, then

$$\begin{aligned}
\mathrm{Tr}\left( \mathbf{X}^\top (\mathbf{X}\mathbf{X}^\top)^{-2} \mathbf{X}\boldsymbol{\Sigma} \right) &\geq \frac{1}{c_2 b^2} \sum_{j=1}^{p} \min\left( \frac{1}{n_0}, \frac{b^2 n_0 \lambda_j^2}{(\lambda_{k+1} r_k(\boldsymbol{\Sigma}))^2} \right) \\
&= \frac{1}{c_2 b^2} \min_{\ell \leq k} \left( \frac{\ell}{n_0} + \frac{b^2 n_0 \sum_{j>\ell} \lambda_j^2}{(\lambda_{k+1} r_k(\boldsymbol{\Sigma}))^2} \right),
\end{aligned}$$

where the equality follows from the fact that $\lambda_j$ are non-increasing. Taking $c > \max(c_1, c_2)$ completes the proof. $\qquad\square$

### B.2.3 Matching Upper and Lower Bounds

Combining Lemmas 10 and 12, if $r_k(\mathbf{\Sigma}) \geq bn_0$ for some $k \leq n/c$, the variance $\mathcal{V}_{\mathrm{M}}^{(0)}$ is within a constant factor of

$$\sigma_0^2 \min_{\ell \leq k} \left( \frac{\ell}{n_0} + \frac{n_0 \sum_{j>\ell} \lambda_j^2}{\left(\lambda_{k+1} r_k(\mathbf{\Sigma})\right)^2} \right).$$

Choosing $k = k_0^*$, which is the smallest one among all qualifying $k$'s, simplifies the bound as follows.

**Lemma 13** (Lemma 11, Bartlett et al. [5]). *Let Assumption 1 hold. For any $b \geq 1$, if the minimum*

$$k_0^* = \min \left\{ k \geq 0 : r_k(\mathbf{\Sigma}) \geq bn_0 \right\}$$

*is well-defined, then*

$$\min_{\ell \leq k_0^*} \left( \frac{\ell}{bn_0} + \frac{bn_0 \sum_{j>\ell} \lambda_j^2}{\left(\lambda_{k_0^*+1} r_{k_0^*}(\mathbf{\Sigma})\right)^2} \right) = \frac{k_0^*}{bn_0} + \frac{bn_0 \sum_{j>k_0^*} \lambda_j^2}{\left(\lambda_{k_0^*+1} r_{k_0^*}(\mathbf{\Sigma})\right)^2} = \frac{k_0^*}{bn_0} + \frac{bn_0}{R_{k_0^*}(\mathbf{\Sigma})}.$$

*Proof.* Observe that

$$\begin{aligned}
\frac{\ell}{bn_0} + bn_0 \frac{\sum_{j>\ell} \lambda_j^2}{\left(\lambda_{k_0^*+1} r_{k_0^*}(\mathbf{\Sigma})\right)^2} &= \sum_{j=1}^{\ell} \frac{1}{bn_0} + \sum_{j>\ell} \frac{bn_0 \lambda_j^2}{\left(\lambda_{k_0^*+1} r_{k_0^*}(\mathbf{\Sigma})\right)^2} \\
&\geq \sum_{j=1}^{k_0^*} \min \left( \frac{1}{bn_0}, \frac{bn_0 \lambda_j^2}{\left(\lambda_{k_0^*+1} r_{k_0^*}(\mathbf{\Sigma})\right)^2} \right) + \sum_{j>k_0^*} \frac{bn_0 \lambda_j^2}{\left(\lambda_{k_0^*+1} r_{k_0^*}(\mathbf{\Sigma})\right)^2} \\
&= \sum_{j=1}^{\ell^*} \frac{1}{bn_0} + \sum_{j>\ell^*} \frac{bn_0 \lambda_j^2}{\left(\lambda_{k_0^*+1} r_{k_0^*}(\mathbf{\Sigma})\right)^2},
\end{aligned}$$

where $\ell^*$ denotes the largest value of $j \leq k_0^*$ for which

$$\frac{1}{bn_0} \leq \frac{bn_0 \lambda_j^2}{\left(\lambda_{k_0^*+1} r_{k_0^*}(\mathbf{\Sigma})\right)^2},$$

since $\lambda_j^2$ are non-increasing. This condition holds if and only if

$$\lambda_j \geq \frac{\lambda_{k_0^*+1} r_{k_0^*}(\mathbf{\Sigma})}{bn_0}.$$

By the definition of $k_0^*$, we have $r_{k_0^*-1}(\mathbf{\Sigma}) < bn_0$, which implies

$$r_{k_0^*}(\mathbf{\Sigma}) = \frac{\sum_{j>k_0^*} \lambda_j}{\lambda_{k_0^*+1}} = \frac{\sum_{j>k_0^*-1} \lambda_j - \lambda_{k_0^*}}{\lambda_{k_0^*+1}} = \frac{\lambda_{k_0^*}}{\lambda_{k_0^*+1}} \left( r_{k_0^*-1}(\mathbf{\Sigma}) - 1 \right) < \frac{\lambda_{k_0^*}}{\lambda_{k_0^*+1}} (bn_0 - 1),$$

so the minimizing $\ell$ is $k_0^*$. This completes the proof. $\qquad\square$

Using Lemma 13, we can obtain the upper and lower bounds that match up to a constant factor in the desired form that aligns with the formal definition of benign overfitting in Definition 2.

**Corollary 3.** *Let Assumptions 1 and 2 hold; that is, there exist universal constants $b_0, c_0 \geq 1$ such that the minimal index*

$$k_0^* = \min \left\{ k \geq 0 : r_k(\mathbf{\Sigma}) \geq b_0 n_0 \right\}$$

*is well-defined with $0 \leq k_0^* \leq n_0/c_0$. Then, with probability at least $1 - 7e^{-n_0/c_0}$,*

$$\frac{\sigma_0^2}{n_0} \mathrm{Tr}\left(\hat{\mathbf{\Sigma}}^\dagger \mathbf{\Sigma}\right) \leq c_0 \sigma_0^2 \min_{\ell \leq k_0^*} \left( \frac{\ell}{n_0} + \frac{n_0 \sum_{j>\ell} \lambda_j^2}{\left(\lambda_{k_0^*+1} r_{k_0^*}(\mathbf{\Sigma})\right)^2} \right) \asymp c_0 \sigma_0^2 \left( \frac{k_0^*}{n_0} + \frac{n_0}{R_{k_0^*}(\mathbf{\Sigma})} \right),$$

*and with probability at least $1 - 10e^{-n_0/c_0}$,*

$$\frac{\sigma_0^2}{n_0} \mathrm{Tr}\left(\hat{\mathbf{\Sigma}}^\dagger \mathbf{\Sigma}\right) \geq \frac{1}{c_0 b_0^2} \sigma_0^2 \min_{\ell \leq k_0^*} \left( \frac{\ell}{n_0} + \frac{b_0^2 n_0 \sum_{j>\ell} \lambda_j^2}{\left(\lambda_{k_0^*+1} r_{k_0^*}(\mathbf{\Sigma})\right)^2} \right) \asymp \frac{1}{c_0} \sigma_0^2 \left( \frac{k_0^*}{n_0} + \frac{n_0}{R_{k_0^*}(\mathbf{\Sigma})} \right).$$

# C   Proofs

Throughout our proofs, we assume without loss of generality as in Section B that each design $\mathbf{X}^{(q)} \in \mathbb{R}^{n_q \times p}$ is of full row rank, i.e., $\mathrm{rank}(\mathbf{X}^{(q)}) = n_q < p$, so that $\mathbf{X}^{(q)}\mathbf{X}^{(q)\top}$ is invertible. Thus, we write the matrices $\left(\mathbf{X}^{(q)}\mathbf{X}^{(q)\top}\right)^{\dagger}$ and $\left(\mathbf{X}^{(q)}\mathbf{X}^{(q)\top}\right)^{-1}$ interchangeably.

## C.1   Proof of Equation 2

*Proof.* The identity in Equation (2) is well-known; we provide a proof here for completeness. Recall that each single-task MNI is given by $\hat{\boldsymbol{\beta}}_{\mathrm{M}}^{(q)} = \mathbf{X}^{(q)\top}\left(\mathbf{X}^{(q)}\mathbf{X}^{(q)\top}\right)^{-1}\mathbf{y}^{(q)}$. Given an arbitrary interpolator $\tilde{\boldsymbol{\beta}}$ trained on the same dataset $(\mathbf{X}^{(q)}, \mathbf{y}^{(q)})$, we have

$$\mathbf{X}^{(q)}\left(\tilde{\boldsymbol{\beta}} - \hat{\boldsymbol{\beta}}_{\mathrm{M}}^{(q)}\right) = \mathbf{y}^{(q)} - \mathbf{y}^{(q)} = \mathbf{0}_{n_q}.$$

This implies

$$
\begin{aligned}
\left(\tilde{\boldsymbol{\beta}} - \hat{\boldsymbol{\beta}}_{\mathrm{M}}^{(q)}\right)^{\top}\hat{\boldsymbol{\beta}}_{\mathrm{M}}^{(q)} &= \left(\tilde{\boldsymbol{\beta}} - \hat{\boldsymbol{\beta}}_{\mathrm{M}}^{(q)}\right)^{\top}\mathbf{X}^{(q)\top}\left(\mathbf{X}^{(q)}\mathbf{X}^{(q)\top}\right)^{-1}\mathbf{y}^{(q)} \\
&= \left(\mathbf{X}^{(q)}\left(\tilde{\boldsymbol{\beta}} - \hat{\boldsymbol{\beta}}_{\mathrm{M}}^{(q)}\right)\right)^{\top}\left(\mathbf{X}^{(q)}\mathbf{X}^{(q)\top}\right)^{-1}\mathbf{y}^{(q)} \\
&= 0,
\end{aligned}
$$

i.e., $\left(\tilde{\boldsymbol{\beta}} - \hat{\boldsymbol{\beta}}_{\mathrm{M}}^{(q)}\right)$ and $\hat{\boldsymbol{\beta}}_{\mathrm{M}}^{(q)}$ are orthogonal. Thus,

$$\|\tilde{\boldsymbol{\beta}}\| = \|\hat{\boldsymbol{\beta}}_{\mathrm{M}}^{(q)} + \left(\tilde{\boldsymbol{\beta}} - \hat{\boldsymbol{\beta}}_{\mathrm{M}}^{(q)}\right)\| = \|\hat{\boldsymbol{\beta}}_{\mathrm{M}}^{(q)}\| + \|\tilde{\boldsymbol{\beta}} - \hat{\boldsymbol{\beta}}_{\mathrm{M}}^{(q)}\| \geq \|\hat{\boldsymbol{\beta}}_{\mathrm{M}}^{(q)}\|,$$

and the equality holds if and only if $\tilde{\boldsymbol{\beta}} = \hat{\boldsymbol{\beta}}_{\mathrm{M}}^{(q)}$; that is, the MNI has the minimum $\ell_2$-norm among all possible interpolators and is uniquely determined. This completes the proof. $\qquad\square$

## C.2   Proof of Equation 3

*Proof.* We present a detailed derivation of the TM estimate in Equation (3). First, we may write

$$
\begin{aligned}
\hat{\boldsymbol{\beta}}_{\mathrm{TM}}^{(q)} &:= \arg\min_{\boldsymbol{\beta}\in\mathbb{R}^p}\left\{\|\boldsymbol{\beta} - \hat{\boldsymbol{\beta}}_{\mathrm{M}}^{(q)}\| : \mathbf{X}^{(0)}\boldsymbol{\beta} = \mathbf{y}^{(0)}\right\} \\
&= \arg\min_{\boldsymbol{\beta}\in\mathbb{R}^p}\left\{\|\boldsymbol{\beta} - \hat{\boldsymbol{\beta}}_{\mathrm{M}}^{(q)}\| : \mathbf{X}^{(0)}\left(\boldsymbol{\beta} - \hat{\boldsymbol{\beta}}_{\mathrm{M}}^{(q)}\right) = \mathbf{y}^{(0)} - \mathbf{X}^{(0)}\hat{\boldsymbol{\beta}}_{\mathrm{M}}^{(q)}\right\} \\
&= \arg\min_{\boldsymbol{\theta}\in\mathbb{R}^p}\left\{\|\boldsymbol{\theta}\| : \mathbf{X}^{(0)}\boldsymbol{\theta} = \mathbf{y}^{(0)} - \mathbf{X}^{(0)}\hat{\boldsymbol{\beta}}_{\mathrm{M}}^{(q)}\right\}.
\end{aligned}
$$

Here, we have $\boldsymbol{\theta} = \boldsymbol{\beta} - \hat{\boldsymbol{\beta}}_{\mathrm{M}}^{(q)}$ and the minimizer $\boldsymbol{\theta}^*$ of $\boldsymbol{\theta}$ is given by $\boldsymbol{\theta}^* = \boldsymbol{\beta}^* - \hat{\boldsymbol{\beta}}_{\mathrm{M}}^{(q)}$, where $\boldsymbol{\beta}^* = \hat{\boldsymbol{\beta}}_{\mathrm{TM}}^{(q)}$ is the minimizer of $\boldsymbol{\beta}$. By the definition of MNI in Equation (2), the minimizer $\boldsymbol{\theta}^*$ is given by

$$\boldsymbol{\theta}^* = \mathbf{X}^{(0)\top}\left(\mathbf{X}^{(0)}\mathbf{X}^{(0)\top}\right)^{\dagger}\left(\mathbf{y}^{(0)} - \mathbf{X}^{(0)}\hat{\boldsymbol{\beta}}_{\mathrm{M}}^{(q)}\right).$$

Therefore, we have

$$
\begin{aligned}
\hat{\boldsymbol{\beta}}_{\mathrm{TM}}^{(q)} - \hat{\boldsymbol{\beta}}_{\mathrm{M}}^{(q)} &= \mathbf{X}^{(0)\top}\left(\mathbf{X}^{(0)}\mathbf{X}^{(0)\top}\right)^{\dagger}\left(\mathbf{y}^{(0)} - \mathbf{X}^{(0)}\hat{\boldsymbol{\beta}}_{\mathrm{M}}^{(q)}\right) \\
\implies \hat{\boldsymbol{\beta}}_{\mathrm{TM}}^{(q)} &= \hat{\boldsymbol{\beta}}_{\mathrm{M}}^{(q)} + \hat{\boldsymbol{\beta}}_{\mathrm{M}}^{(0)} - \mathbf{H}^{(0)}\hat{\boldsymbol{\beta}}_{\mathrm{M}}^{(q)} \\
&= \hat{\boldsymbol{\beta}}_{\mathrm{M}}^{(0)} + \left(\mathbf{I}_p - \mathbf{H}^{(0)}\right)\hat{\boldsymbol{\beta}}_{\mathrm{M}}^{(q)},
\end{aligned}
$$

where $\mathbf{H}^{(0)} = \mathbf{X}^{(0)\top}\left(\mathbf{X}^{(0)}\mathbf{X}^{(0)\top}\right)^{\dagger}\mathbf{X}^{(0)}$. This completes the proof. $\qquad\square$

## C.3   Proof of Lemma 1

*Proof.* By Equation (5), the excess risk of the TM estimate is given by

$$\mathcal{R}_{\mathrm{TM}}^{(q)} = \mathbb{E}_{\mathcal{E}}\left[\left(\hat{\boldsymbol{\beta}}_{\mathrm{TM}}^{(q)} - \boldsymbol{\beta}^{(0)}\right)^{\top}\boldsymbol{\Sigma}^{(0)}\left(\hat{\boldsymbol{\beta}}_{\mathrm{TM}}^{(q)} - \boldsymbol{\beta}^{(0)}\right) \mid \mathcal{X}\right],$$

where $(\mathcal{X}, \mathcal{E})$ are as specified in Assumption 1. For simplicity, we write

$$\mathbb{E}[\cdot] \equiv \mathbb{E}_{\mathcal{E}}[\cdot \mid \mathcal{X}], \qquad \mathrm{Cov}(\cdot) \equiv \mathrm{Cov}_{\mathcal{E}}(\cdot \mid \mathcal{X}),$$

so that we always take expectations over the randomness of noises $\mathcal{E}$ conditional on the designs $\mathcal{X}$, unless specified otherwise. Lemma 2 gives

$$\mathcal{R}_{\mathrm{TM}}^{(q)} = \underbrace{\left(\mathbb{E}\hat{\boldsymbol{\beta}}_{\mathrm{TM}}^{(q)} - \boldsymbol{\beta}^{(0)}\right)^{\top} \boldsymbol{\Sigma}^{(0)} \left(\mathbb{E}\hat{\boldsymbol{\beta}}_{\mathrm{TM}}^{(q)} - \boldsymbol{\beta}^{(0)}\right)}_{\mathcal{B}_{\mathrm{TM}}^{(q)}} + \underbrace{\mathrm{Tr}\left(\boldsymbol{\Sigma}^{(0)} \mathrm{Cov}\big(\hat{\boldsymbol{\beta}}_{\mathrm{TM}}^{(q)}\big)\right)}_{\mathcal{V}_{\mathrm{TM}}^{(q)}},$$

i.e., the excess risk decomposes into a sum of bias and variance.

**Bias of TM.** Recall from Equation (3) that

$$\hat{\boldsymbol{\beta}}_{\mathrm{TM}}^{(q)} = \hat{\boldsymbol{\beta}}_{\mathrm{M}}^{(0)} + \big(\mathbf{I}_p - \mathbf{H}^{(0)}\big)\hat{\boldsymbol{\beta}}_{\mathrm{M}}^{(q)},$$

where $\mathbf{H}^{(q)} = \mathbf{X}^{(q)\top}\big(\mathbf{X}^{(q)}\mathbf{X}^{(q)\top}\big)^{\dagger}\mathbf{X}^{(q)}$ is the orthogonal projection onto the row space of $\mathbf{X}^{(q)}$. Here, the expectation of the single-task MNI $\hat{\boldsymbol{\beta}}_{\mathrm{M}}^{(q)}$ is given by

$$\mathbb{E}\hat{\boldsymbol{\beta}}_{\mathrm{M}}^{(q)} = \mathbf{X}^{(q)\top}\big(\mathbf{X}^{(q)}\mathbf{X}^{(q)\top}\big)^{\dagger}\mathbb{E}\big[\mathbf{X}^{(q)}\boldsymbol{\beta}^{(q)} + \boldsymbol{\epsilon}^{(q)}\big] = \mathbf{H}^{(q)}\boldsymbol{\beta}^{(q)}, \quad q \in [Q]_0,$$

since $\boldsymbol{\epsilon}^{(q)}$ is mean-zero by Assumption 1. Thus, the following expectation holds:

$$\mathbb{E}\hat{\boldsymbol{\beta}}_{\mathrm{TM}}^{(q)} = \mathbf{H}^{(0)}\boldsymbol{\beta}^{(0)} + \big(\mathbf{I}_p - \mathbf{H}^{(0)}\big)\mathbf{H}^{(q)}\boldsymbol{\beta}^{(q)}$$

$$\implies \mathbb{E}\hat{\boldsymbol{\beta}}_{\mathrm{TM}}^{(q)} - \boldsymbol{\beta}^{(0)} = (\mathbf{I}_p - \mathbf{H}^{(0)})\big(\mathbf{H}^{(q)}\boldsymbol{\beta}^{(q)} - \boldsymbol{\beta}^{(0)}\big).$$

Denoting $\boldsymbol{\Pi}^{(0)} = \big(\mathbf{I}_p - \mathbf{H}^{(0)}\big)\boldsymbol{\Sigma}^{(0)}\big(\mathbf{I}_p - \mathbf{H}^{(0)}\big)$ where the orthogonal projections $\mathbf{H}^{(q)}$ and $\mathbf{I}_p - \mathbf{H}^{(q)}$ are symmetric, we have

$$\mathcal{B}_{\mathrm{TM}}^{(q)} = (\mathbf{H}^{(q)}\boldsymbol{\beta}^{(q)} - \boldsymbol{\beta}^{(0)})^{\top}\boldsymbol{\Pi}^{(0)}(\mathbf{H}^{(q)}\boldsymbol{\beta}^{(q)} - \boldsymbol{\beta}^{(0)}), \quad q \neq 0,$$

where

$$\mathbf{H}^{(q)}\boldsymbol{\beta}^{(q)} - \boldsymbol{\beta}^{(0)} = \mathbf{H}^{(q)}(\boldsymbol{\beta}^{(0)} + \boldsymbol{\delta}^{(q)}) - \boldsymbol{\beta}^{(0)} = \mathbf{H}^{(q)}\boldsymbol{\delta}^{(q)} - \big(\mathbf{I}_p - \mathbf{H}^{(q)}\big)\boldsymbol{\beta}^{(0)}.$$

Thus, we have

$$\begin{aligned}
\mathcal{B}_{\mathrm{TM}}^{(q)} = {}& \boldsymbol{\beta}^{(0)\top}\big(\mathbf{I}_p - \mathbf{H}^{(q)}\big)\boldsymbol{\Pi}^{(0)}\big(\mathbf{I}_p - \mathbf{H}^{(q)}\big)\boldsymbol{\beta}^{(0)} + \boldsymbol{\delta}^{(q)\top}\mathbf{H}^{(q)}\boldsymbol{\Pi}^{(0)}\mathbf{H}^{(q)}\boldsymbol{\delta}^{(q)} \\
& - 2\boldsymbol{\delta}^{(q)\top}\mathbf{H}^{(q)}\boldsymbol{\Pi}^{(0)}\big(\mathbf{I}_p - \mathbf{H}^{(q)}\big)\boldsymbol{\beta}^{(0)},
\end{aligned}$$

which completes the derivation of $\mathcal{B}_{\mathrm{TM}}^{(q)}$ in Lemma 1.

**Variance of TM.** As for the variance $\mathcal{V}_{\mathrm{TM}}^{(q)}$, we may write

$$\begin{aligned}
\mathrm{Cov}\big(\hat{\boldsymbol{\beta}}_{\mathrm{TM}}^{(q)}\big) &= \mathrm{Cov}\Big(\hat{\boldsymbol{\beta}}_{\mathrm{M}}^{(0)} + \big(\mathbf{I}_p - \mathbf{H}^{(0)}\big)\hat{\boldsymbol{\beta}}_{\mathrm{M}}^{(q)}\Big) \\
&= \mathrm{Cov}\big(\hat{\boldsymbol{\beta}}_{\mathrm{M}}^{(0)}\big) + \big(\mathbf{I}_p - \mathbf{H}^{(0)}\big)\mathrm{Cov}\big(\hat{\boldsymbol{\beta}}_{\mathrm{M}}^{(q)}\big)\big(\mathbf{I}_p - \mathbf{H}^{(0)}\big),
\end{aligned}$$

where the second equation holds since $\hat{\boldsymbol{\beta}}_{\mathrm{M}}^{(0)}$ and $\hat{\boldsymbol{\beta}}_{\mathrm{M}}^{(q)}$ are independent. Here, the covariance of the single-task MNI $\hat{\boldsymbol{\beta}}_{\mathrm{M}}^{(q)}$ is given by

$$\begin{aligned}
\mathrm{Cov}\big(\hat{\boldsymbol{\beta}}_{\mathrm{M}}^{(q)}\big) &= \mathrm{Cov}\Big(\mathbf{X}^{(q)\top}\big(\mathbf{X}^{(q)}\mathbf{X}^{(q)\top}\big)^{\dagger}\mathbf{y}^{(q)}\Big) \\
&= \mathrm{Cov}\Big(\big(\mathbf{X}^{(q)\top}\mathbf{X}^{(q)}\big)^{\dagger}\mathbf{X}^{(q)\top}\mathbf{y}^{(q)}\Big) \\
&= \big(\mathbf{X}^{(q)\top}\mathbf{X}^{(q)}\big)^{\dagger}\mathbf{X}^{(q)\top}\mathrm{Cov}\big(\boldsymbol{\epsilon}^{(q)}\big)\mathbf{X}^{(q)}\big(\mathbf{X}^{(q)\top}\mathbf{X}^{(q)}\big)^{\dagger} \\
&= \sigma_q^2\big(\mathbf{X}^{(q)\top}\mathbf{X}^{(q)}\big)^{\dagger}, \quad q \in [Q]_0,
\end{aligned}$$

where the last equality holds since $\mathrm{Cov}\big(\boldsymbol{\epsilon}^{(q)}\big) = \sigma_q^2\mathbf{I}_{n_q}$ by Assumption 1 and $\mathbf{M}^{\dagger}\mathbf{M}\mathbf{M}^{\dagger} = \mathbf{M}^{\dagger}$ for any $\mathbf{M}$ by the definition of Moore-Penrose inverse. Denoting $\hat{\boldsymbol{\Sigma}}^{(q)} = (1/n_q)\mathbf{X}^{(q)\top}\mathbf{X}^{(q)}$, we have

$$\mathrm{Cov}\big(\hat{\boldsymbol{\beta}}_{\mathrm{M}}^{(q)}\big) = \frac{\sigma_q^2}{n_q}\hat{\boldsymbol{\Sigma}}^{(q)\dagger}, \quad q \in [Q]_0,$$

which yields

$$\operatorname{Cov}\big(\hat{\boldsymbol{\beta}}_{\mathrm{TM}}^{(q)}\big) = \frac{\sigma_0^2}{n_0}\hat{\boldsymbol{\Sigma}}^{(0)\dagger} + \frac{\sigma_q^2}{n_q}\big(\mathbf{I}_p - \mathbf{H}^{(0)}\big)\hat{\boldsymbol{\Sigma}}^{(q)\dagger}\big(\mathbf{I}_p - \mathbf{H}^{(0)}\big).$$

Plugging this into $\mathcal{V}_{\mathrm{TM}}^{(q)}$ yields

$$
\begin{aligned}
\mathcal{V}_{\mathrm{TM}}^{(q)} &= \operatorname{Tr}\Big(\operatorname{Cov}\big(\hat{\boldsymbol{\beta}}_{\mathrm{TM}}^{(q)}\big)\boldsymbol{\Sigma}^{(0)}\Big) \\
&= \frac{\sigma_0^2}{n_0}\operatorname{Tr}\Big(\hat{\boldsymbol{\Sigma}}^{(0)\dagger}\boldsymbol{\Sigma}^{(0)}\Big) + \frac{\sigma_q^2}{n_q}\operatorname{Tr}\Big(\big(\mathbf{I}_p - \mathbf{H}^{(0)}\big)\hat{\boldsymbol{\Sigma}}^{(q)\dagger}\big(\mathbf{I}_p - \mathbf{H}^{(0)}\big)\boldsymbol{\Sigma}^{(0)}\Big) \\
&= \frac{\sigma_0^2}{n_0}\operatorname{Tr}\Big(\hat{\boldsymbol{\Sigma}}^{(0)\dagger}\boldsymbol{\Sigma}^{(0)}\Big) + \frac{\sigma_q^2}{n_q}\operatorname{Tr}\Big(\hat{\boldsymbol{\Sigma}}^{(q)\dagger}\boldsymbol{\Pi}^{(0)}\Big).
\end{aligned}
$$

This completes the derivation of $\mathcal{V}_{\mathrm{TM}}^{(q)}$ in Lemma 1. $\qquad\square$

### C.4  Proof of Theorem 1

*Proof.* Since we assume i.i.d. standard Gaussian design for $(\mathbf{X}^{(0)}, \mathbf{X}^{(q)})$ in Theorem 1, we have $\boldsymbol{\Sigma}^{(0)} = \boldsymbol{\Sigma}^{(q)} = \mathbf{I}_p$ and $\boldsymbol{\Pi}^{(0)} = (\mathbf{I}_p - \mathbf{H}^{(0)})\boldsymbol{\Sigma}^{(0)}(\mathbf{I}_p - \mathbf{H}^{(0)}) = \mathbf{I}_p - \mathbf{H}^{(0)}$ for idempotent $\mathbf{H}^{(0)}$.

**Bias of TM.**  Under the standard Gaussianity assumption, the bias of the target and transfer tasks, which are provided in Equation (6) and Lemma 1 respectively, are reduced to as follows:

$$
\begin{aligned}
\mathcal{B}_{\mathrm{M}}^{(0)} &= \boldsymbol{\beta}^{(0)\top}\big(\mathbf{I}_p - \mathbf{H}^{(0)}\big)\boldsymbol{\beta}^{(0)}, \\
\mathcal{B}_{\mathrm{TM}}^{(q)} &= \boldsymbol{\beta}^{(0)\top}\big(\mathbf{I}_p - \mathbf{H}^{(q)}\big)\big(\mathbf{I}_p - \mathbf{H}^{(0)}\big)\big(\mathbf{I}_p - \mathbf{H}^{(q)}\big)\boldsymbol{\beta}^{(0)} + \boldsymbol{\delta}^{(q)\top}\mathbf{H}^{(q)}\big(\mathbf{I}_p - \mathbf{H}^{(0)}\big)\mathbf{H}^{(q)}\boldsymbol{\delta}^{(q)} \\
&\quad - 2\boldsymbol{\delta}^{(q)\top}\mathbf{H}^{(q)}\big(\mathbf{I}_p - \mathbf{H}^{(0)}\big)\big(\mathbf{I}_p - \mathbf{H}^{(q)}\big)\boldsymbol{\beta}^{(0)}.
\end{aligned}
$$

First, by Lemma 3, the expected bias of target task is derived as

$$\mathbb{E}_{\mathcal{X}}\mathcal{B}_{\mathrm{M}}^{(0)} = \mathbb{E}_{\mathbf{X}^{(0)}}\big\|\big(\mathbf{I}_p - \mathbf{H}^{(0)}\big)\boldsymbol{\beta}^{(0)}\big\|^2 = \left(\frac{p - n_0}{p}\right)\big\|\boldsymbol{\beta}^{(0)}\big\|^2.$$

Next, to derive the expected bias risk of the transfer task, we again use Lemma 3. The expectation of the first term in $\mathcal{B}_{\mathrm{TM}}^{(q)}$ is obtained as follows:

$$
\begin{aligned}
&\mathbb{E}_{(\mathbf{X}^{(0)}, \mathbf{X}^{(q)})}\Big[\boldsymbol{\beta}^{(0)\top}(\mathbf{I}_p - \mathbf{H}^{(q)})(\mathbf{I}_p - \mathbf{H}^{(0)})(\mathbf{I}_p - \mathbf{H}^{(q)})\boldsymbol{\beta}^{(0)}\Big] \\
&= \mathbb{E}_{\mathbf{X}^{(q)}}\mathbb{E}_{\mathbf{X}^{(0)}}\Big[\boldsymbol{\beta}^{(0)\top}(\mathbf{I}_p - \mathbf{H}^{(q)})(\mathbf{I}_p - \mathbf{H}^{(0)})(\mathbf{I}_p - \mathbf{H}^{(q)})\boldsymbol{\beta}^{(0)} \mid \mathbf{X}^{(q)}\Big] \\
&= \left(\frac{p - n_0}{p}\right)\mathbb{E}_{\mathbf{X}^{(q)}}\big\|(\mathbf{I}_p - \mathbf{H}^{(q)})\boldsymbol{\beta}^{(0)}\big\|^2 \\
&= \left(\frac{p - n_0}{p}\right)\left(\frac{p - n_q}{p}\right)\big\|\boldsymbol{\beta}^{(0)}\big\|^2, \quad q \neq 0,
\end{aligned}
$$

where we use the law of total expectation in the second line and the equality

$$\mathbb{E}_{\mathbf{X}^{(0)}}\big[\mathbf{I}_p - \mathbf{H}^{(0)}\big] = \left(\frac{p - n_0}{p}\right)\mathbf{I}_p$$

given by Lemma 3 to obtain the third line. Applying the law of total expectation in the same way, we derive the expectation of the second term in $\mathcal{B}_{\mathrm{TM}}^{(q)}$ as follows:

$$
\begin{aligned}
\mathbb{E}_{(\mathbf{X}^{(0)}, \mathbf{X}^{(q)})}\Big[\boldsymbol{\delta}^{(q)\top}\mathbf{H}^{(q)}\big(\mathbf{I}_p - \mathbf{H}^{(0)}\big)\mathbf{H}^{(q)}\boldsymbol{\delta}^{(q)}\Big] &= \left(\frac{p - n_0}{p}\right)\mathbb{E}_{\mathbf{X}^{(q)}}\big\|\mathbf{H}^{(q)}\boldsymbol{\delta}^{(q)}\big\|^2 \\
&= \left(\frac{p - n_0}{p}\right)\frac{n_q}{p}\big\|\boldsymbol{\delta}^{(q)}\big\|^2, \quad q \neq 0.
\end{aligned}
$$

Lastly, the expectation of the third term in $\mathcal{B}_{\mathrm{TM}}^{(q)}$ vanishes, since

$$\mathbb{E}_{(\mathbf{X}^{(0)}, \mathbf{X}^{(q)})} \left[ -2\boldsymbol{\delta}^{(q)\top} \mathbf{H}^{(q)} (\mathbf{I}_p - \mathbf{H}^{(0)})(\mathbf{I}_p - \mathbf{H}^{(q)})\boldsymbol{\beta}^{(0)} \right]$$

$$= -2\boldsymbol{\delta}^{(q)\top} \mathbb{E}_{\mathbf{X}^{(q)}} \mathbb{E}_{\mathbf{X}^{(0)}} \left[ \mathbf{H}^{(q)}(\mathbf{I}_p - \mathbf{H}^{(0)})(\mathbf{I}_p - \mathbf{H}^{(q)}) \mid \mathbf{X}^{(q)} \right] \boldsymbol{\beta}^{(0)}$$

$$= -2\left(\frac{p - n_0}{p}\right) \boldsymbol{\delta}^{(q)\top} \mathbb{E}_{\mathbf{X}^{(q)}} \left[ \mathbf{H}^{(q)}(\mathbf{I}_p - \mathbf{H}^{(q)}) \right] \boldsymbol{\beta}^{(0)},$$

and $\mathbf{H}^{(q)}(\mathbf{I}_p - \mathbf{H}^{(q)}) = \mathbf{0}_{p \times p}$ almost surely as $\mathbf{H}^{(q)}$ is idempotent. Combining the results, we have

$$\mathbb{E}_{\mathcal{X}} \mathcal{B}_{\mathrm{TM}}^{(q)} = \left(\frac{p - n_0}{p}\right)\left(\frac{p - n_q}{p}\right) \|\boldsymbol{\beta}^{(0)}\|^2 + \left(\frac{p - n_0}{p}\right) \frac{n_q}{p} \|\boldsymbol{\delta}^{(q)}\|^2$$

$$= \frac{p - n_0}{p} \left(\frac{p - n_q}{p} \|\boldsymbol{\beta}^{(0)}\|^2 + \frac{n_q}{p} \|\boldsymbol{\delta}^{(q)}\|^2\right),$$

which completes the derivation of the expected bias in Theorem 1. Note that Lemma 3 requires $p > n_0 + 1$ and $p > n_q + 1$, which is assumed in Theorem 1.

**Variance of TM.** Under the standard Gaussianity assumption, we can simplify the variances in Equation (6) and Lemma 1 as follows:

$$\mathcal{V}_{\mathrm{M}}^{(0)} = \frac{\sigma_0^2}{n_0} \mathrm{Tr}\left(\hat{\boldsymbol{\Sigma}}^{(0)\dagger}\right) = \sigma_0^2 \mathrm{Tr}\left((\mathbf{X}^{(0)} \mathbf{X}^{(0)\top})^{-1}\right),$$

$$\mathcal{V}_{\mathrm{TM}}^{(q)} = \underbrace{\frac{\sigma_q^2}{n_q} \mathrm{Tr}\left(\hat{\boldsymbol{\Sigma}}^{(q)\dagger}(\mathbf{I}_p - \mathbf{H}^{(0)})\right)}_{\text{variance inflation} = \mathcal{V}_{\uparrow}^{(q)}} + \mathcal{V}_{\mathrm{M}}^{(0)},$$

since, for each $q \in [Q]_0$, we have

$$\hat{\boldsymbol{\Sigma}}^{(q)\dagger} = \left(\frac{\mathbf{X}^{(q)\top} \mathbf{X}^{(q)}}{n_q}\right)^{\dagger} = n_q \mathbf{X}^{(q)\dagger}(\mathbf{X}^{(q)\top})^{\dagger} = n_q \mathbf{X}^{(q)\top}(\mathbf{X}^{(q)} \mathbf{X}^{(q)\top})^{-2} \mathbf{X}^{(q)},$$

and trace is invariant to cyclic permutation. As we assume $p > (n_0 + 1) \vee (n_q + 1)$ in Theorem 1, we may use the result from Section 2.2 in Belkin et al. [8]: for each $q \in [Q]_0$, $(\mathbf{X}^{(q)} \mathbf{X}^{(q)\top})^{-1}$ follows inverse-Wishart distribution with scale matrix $\mathbf{I}_{n_q}$ and $p$ degrees-of-freedom. This implies that each diagonal entry of $(\mathbf{X}^{(q)} \mathbf{X}^{(q)\top})^{-1}$

$$\left[(\mathbf{X}^{(q)} \mathbf{X}^{(q)\top})^{-1}\right]_{ii}, \quad i \in [n_q],$$

has a reciprocal that follows the $\chi^2$-distribution with $p - (n_q - 1)$ degrees-of-freedom. Therefore, we have

$$\mathbb{E}\left[(\mathbf{X}^{(q)} \mathbf{X}^{(q)\top})^{-1}\right]_{ii} = \frac{1}{p - (n_q + 1)}, \quad i \in [n_q],$$

and

$$\mathbb{E}_{\mathcal{X}} \mathcal{V}_{\mathrm{M}}^{(0)} = \sigma_0^2 \mathbb{E} \mathrm{Tr}\left((\mathbf{X}^{(0)} \mathbf{X}^{(0)\top})^{-1}\right) = \sigma_0^2 \mathbb{E} \sum_{i=1}^{n_0} \left[(\mathbf{X}^{(q)} \mathbf{X}^{(q)\top})^{-1}\right]_{ii} = \frac{\sigma_0^2 n_0}{p - (n_0 + 1)}.$$

Now it remains to derive the expectation of variance inflation. The law of total expectation gives

$$\mathbb{E}_{(\mathbf{X}^{(0)}, \mathbf{X}^{(q)})} \left[\frac{\sigma_q^2}{n_q} \mathrm{Tr}\left(\hat{\boldsymbol{\Sigma}}^{(q)\dagger}(\mathbf{I}_p - \mathbf{H}^{(0)})\right)\right] = \frac{\sigma_q^2}{n_q} \mathbb{E}_{\mathbf{X}^{(q)}} \mathrm{Tr}\left(\mathbb{E}_{\mathbf{X}^{(0)}} \left[\hat{\boldsymbol{\Sigma}}^{(q)\dagger}(\mathbf{I}_p - \mathbf{H}^{(0)}) \mid \mathbf{X}^{(q)}\right]\right)$$

$$= \frac{\sigma_q^2}{n_q} \left(\frac{p - n_0}{p}\right) \mathbb{E}_{\mathbf{X}^{(q)}} \mathrm{Tr}\left(n_q (\mathbf{X}^{(q)} \mathbf{X}^{(q)\top})^{-1}\right)$$

$$= \sigma_q^2 \left(\frac{p - n_0}{p}\right) \frac{n_q}{p - (n_q + 1)},$$

which completes the derivation of the expected variance in Theorem 1. $\qquad\square$

## C.5 Proof of Corollary 1

*Proof.* From Theorem 1, we have

$$\mathbb{E}_{\mathcal{X}}\mathcal{R}_{\mathrm{M}}^{(0)} = \frac{p-n_0}{p}\|\boldsymbol{\beta}^{(0)}\|^2 + \frac{\sigma_0^2 n_0}{p-(n_0+1)},$$

$$\mathbb{E}_{\mathcal{X}}\mathcal{R}_{\mathrm{TM}}^{(q)} = \frac{p-n_0}{p}\left(\frac{p-n_q}{p}\|\boldsymbol{\beta}^{(0)}\|^2 + \frac{n_q}{p}\|\boldsymbol{\delta}^{(q)}\|^2\right) + \left(\frac{p-n_0}{p}\right)\frac{\sigma_q^2 n_q}{p-(n_q+1)} + \frac{\sigma_0^2 n_0}{p-(n_0+1)}.$$

Thus, it follows that

$$\mathbb{E}_{\mathcal{X}}\mathcal{R}_{\mathrm{TM}}^{(q)} < \mathbb{E}_{\mathcal{X}}\mathcal{R}_{\mathrm{M}}^{(0)}$$

$$\iff \frac{p-n_0}{p}\left(\frac{p-n_q}{p}\|\boldsymbol{\beta}^{(0)}\|^2 + \frac{n_q}{p}\|\boldsymbol{\delta}^{(q)}\|^2\right) + \left(\frac{p-n_0}{p}\right)\frac{\sigma_q^2 n_q}{p-(n_q+1)} < \frac{p-n_0}{p}\|\boldsymbol{\beta}^{(0)}\|^2$$

$$\iff \frac{n_q}{p}\|\boldsymbol{\delta}^{(q)}\|^2 + \frac{\sigma_q^2 n_q}{p-(n_q+1)} < \frac{n_q}{p}\|\boldsymbol{\beta}^{(0)}\|^2$$

$$\iff \frac{1}{p}\left(\|\boldsymbol{\beta}^{(0)}\|^2 - \|\boldsymbol{\delta}^{(q)}\|^2\right) > \frac{\sigma_q^2}{p-(n_q+1)}. \tag{13}$$

Using the definitions $\mathrm{SNR}_q = \|\boldsymbol{\beta}^{(0)}\|^2/\sigma_q^2 > 0$ and $\mathrm{SSR}_q = \|\boldsymbol{\delta}^{(q)}\|^2/\|\boldsymbol{\beta}^{(0)}\|^2 \geq 0$, we may write

$$\|\boldsymbol{\beta}^{(0)}\|^2 = \sigma_q^2\,\mathrm{SNR}_q, \qquad \|\boldsymbol{\delta}^{(q)}\|^2 = \sigma_q^2\,\mathrm{SNR}_q\,\mathrm{SSR}_q,$$

and therefore, inequality (13) can be written as

$$\mathrm{SNR}_q\,(1-\mathrm{SSR}_q) > \frac{p}{p-(n_q+1)}. \tag{14}$$

From inequality (14), observe that its right-hand side is always positive, which makes $\mathrm{SSR}_q < 1$ a necessary condition for $\mathbb{E}_{\mathcal{X}}\mathcal{R}_{\mathrm{TM}}^{(q)} < \mathbb{E}_{\mathcal{X}}\mathcal{R}_{\mathrm{M}}^{(0)}$. Also, the right-hand side minimized with respect to $n_q \in [1, p-1]$ is $p/(p-2)$, which makes $\mathrm{SNR}_q\,(1-\mathrm{SSR}_q) > p/(p-2)$ another necessary condition.

Provided the condition $\mathrm{SSR}_q < 1$, define the following function of $n_q$:

$$\widetilde{\Delta}(n_q) := \mathbb{E}_{\mathcal{X}}\mathcal{R}_{\mathrm{TM}}^{(q)} - \mathbb{E}_{\mathcal{X}}\mathcal{R}_{\mathrm{M}}^{(0)}$$

$$= \frac{p-n_0}{p}\left(\frac{p-n_q}{p}\|\boldsymbol{\beta}^{(0)}\|^2 + \frac{n_q}{p}\|\boldsymbol{\delta}^{(q)}\|^2\right) + \left(\frac{p-n_0}{p}\right)\frac{\sigma_q^2 n_q}{p-(n_q+1)} - \frac{p-n_0}{p}\|\boldsymbol{\beta}^{(0)}\|^2$$

$$= \left(\frac{p-n_0}{p}\right)\left\{\frac{n_q}{p}\left(\|\boldsymbol{\delta}^{(q)}\|^2 - \|\boldsymbol{\beta}^{(0)}\|^2\right) + n_q\frac{\sigma_q^2}{p-(n_q+1)}\right\}$$

$$= \underbrace{\left(\frac{p-n_0}{p}\right)\|\boldsymbol{\beta}^{(0)}\|^2}_{=:\,B^{(0)}} n_q\left(\frac{\mathrm{SSR}_q - 1}{p} + \frac{1}{\mathrm{SNR}_q(p-n_q-1)}\right),$$

where we write $B^{(0)} = \left(\frac{p-n_0}{p}\right)\|\boldsymbol{\beta}^{(0)}\|^2 > 0$ and define $\Delta(n_q) := -\widetilde{\Delta}(n_q)$ as in Corollary 1. Holding all quantities other than $n_q$ constant and differentiating $\widetilde{\Delta}$ with respect to $n_q$, we obtain

$$\frac{\partial\widetilde{\Delta}}{\partial n_q} = B^{(0)}\left(\frac{\mathrm{SSR}_q - 1}{p} + \frac{1}{\mathrm{SNR}_q(p-n_q-1)}\right) + B^{(0)}n_q\left(\frac{1}{\mathrm{SNR}_q(p-n_q-1)^2}\right)$$

$$= B^{(0)}\left(\frac{\mathrm{SSR}_q - 1}{p} + \frac{1}{\mathrm{SNR}_q(p-n_q-1)} + \frac{n_q}{\mathrm{SNR}_q(p-n_q-1)^2}\right).$$

For the first-order condition $\frac{\partial \widetilde{\Delta}}{\partial n_q} = 0$, multiplying both sides by $p \cdot \text{SNR}_q(p - n_q - 1)^2/B^{(0)} > 0$, we obtain

$$(\text{SSR}_q - 1)\text{SNR}_q(p - n_q^* - 1)^2 + p(p - n_q^* - 1) + pn_q^* = 0$$

$$\implies (\text{SSR}_q - 1)\text{SNR}_q(p - n_q^* - 1)^2 + p(p - 1) = 0$$

$$\implies p - n_q^* - 1 = \sqrt{\frac{p(p-1)}{\text{SNR}_q(1 - \text{SSR}_q)}} > 0$$

$$\implies n_q^* = p - 1 - \sqrt{\frac{p(p-1)}{\text{SNR}_q(1 - \text{SSR}_q)}}.$$

Here, since $\text{SSR}_q < 1$ and $\text{SNR}_q > 0$, $n_q^*$ is strictly less than $p - 1$. Furthermore, we have $n_q^* \geq 1$ if and only if

$$(p - 2)^2 \geq \frac{p(p-1)}{\text{SNR}_q(1 - \text{SSR}_q)} \iff \text{SNR}_q(1 - \text{SSR}_q) \geq \frac{p(p-1)}{(p - 2)^2}.$$

To check the second-order condition at $n_q^*$, consider the second derivative of $\widetilde{\Delta}$ given by

$$\frac{\partial^2 \widetilde{\Delta}}{\partial n_q^2} = \frac{\partial \widetilde{\Delta}}{\partial n_q}\left[ B^{(0)}\left( \frac{\text{SSR}_q - 1}{p} + \frac{1}{\text{SNR}_q(p - n_q - 1)} + \frac{n_q}{\text{SNR}_q(p - n_q - 1)^2} \right) \right]$$

$$= \frac{B^{(0)}}{\text{SNR}_q}\left( \frac{1}{(p - n_q - 1)^2} + \frac{(p - n_q - 1)^2 + n_q \cdot 2(p - n_q - 1)}{(p - n_q - 1)^4} \right)$$

$$= \frac{B^{(0)}}{\text{SNR}_q}\left( \frac{2(p - n_q - 1)^2 + 2n_q(p - n_q - 1)}{(p - n_q - 1)^4} \right)$$

$$= \frac{2B^{(0)}}{\text{SNR}_q}\left( \frac{p - 1}{(p - n_q - 1)^3} \right) > 0, \quad \forall n_q \in [1, p - 1).$$

That is, $\widetilde{\Delta}(n_q)$ is strictly convex on the interval $[1, p - 1)$; conversely, $\Delta(n_q) = \mathbb{E}_{\mathcal{X}}\mathcal{R}_{\text{M}}^{(0)} - \mathbb{E}_{\mathcal{X}}\mathcal{R}_{\text{TM}}^{(q)}$ is strictly concave on the same interval, and thus the improvement in expected excess risk $\Delta(n_q)$ strictly increases on $n_q \in [1, n_q^*]$ and strictly decreases on $n_q \in (n_q^*, p - 1)$.

Now it remains to show that $\Delta(n_q^*) > 0$. Recall from the first-order condition that we have

$$\left.\frac{\partial \widetilde{\Delta}}{\partial n_q}\right|_{n_q = n_q^*} = B^{(0)}\left( \frac{\text{SSR}_q - 1}{p} + \frac{1}{\text{SNR}_q(p - n_q^* - 1)} + \frac{n_q^*}{\text{SNR}_q(p - n_q^* - 1)^2} \right) = 0$$

$$\implies B^{(0)}\left( \frac{\text{SSR}_q - 1}{p} + \frac{1}{\text{SNR}_q(p - n_q^* - 1)} \right) = -\frac{B^{(0)}n_q^*}{\text{SNR}_q(p - n_q^* - 1)^2}.$$

Plugging this into $\widetilde{\Delta}(n_q)$ at $n_q = n_q^*$ and flipping the sign yield the maximal improvement value

$$\Delta(n_q^*) = \left( \frac{p - n_0}{p} \right)\left( \frac{\left( p - 1 - \sqrt{\frac{p(p-1)}{\text{SNR}_q(1 - \text{SSR}_q)}} \right)^2 (1 - \text{SSR}_q)}{p(p - 1)} \right) \|\boldsymbol{\beta}^{(0)}\|^2$$

$$= \left( \frac{p - n_0}{p} \right)\left( \frac{(n_q^*)^2(1 - \text{SSR}_q)}{p(p - 1)} \right) \|\boldsymbol{\beta}^{(0)}\|^2 > 0.$$

This completes the proof. $\qquad\square$

### C.6  Proof of Theorem 2

*Proof.* We first recall the notations $\hat{\boldsymbol{\Sigma}}^{(q)} = (1/n_q)\mathbf{X}^{(q)\top}\mathbf{X}^{(q)}$, $\mathbf{H}^{(q)} = \mathbf{X}^{(q)\top}(\mathbf{X}^{(q)}\mathbf{X}^{(q)\top})^{-1}\mathbf{X}^{(q)}$, and $\boldsymbol{\Pi}^{(0)} = (\mathbf{I}_p - \mathbf{H}^{(0)})\boldsymbol{\Sigma}^{(0)}(\mathbf{I}_p - \mathbf{H}^{(0)})$.

**Upper bound on bias.** First, notice that the bias of the target-only MNI in Equation (6) can be written as

$$\mathcal{B}_{\mathrm{M}}^{(0)} = \boldsymbol{\beta}^{(0)\top}(\mathbf{I}_p - \mathbf{H}^{(0)})\boldsymbol{\Sigma}^{(0)}(\mathbf{I}_p - \mathbf{H}^{(0)})\boldsymbol{\beta}^{(0)}$$
$$= \boldsymbol{\beta}^{(0)\top}(\mathbf{I}_p - \mathbf{H}^{(0)})(\boldsymbol{\Sigma}^{(0)} - \hat{\boldsymbol{\Sigma}}^{(0)})(\mathbf{I}_p - \mathbf{H}^{(0)})\boldsymbol{\beta}^{(0)},$$

since

$$(\mathbf{I}_p - \mathbf{H}^{(q)})\hat{\boldsymbol{\Sigma}}^{(q)} = \frac{1}{n_q}\left(\mathbf{I}_p - \mathbf{X}^{(q)\top}(\mathbf{X}^{(q)}\mathbf{X}^{(q)\top})^{-1}\mathbf{X}^{(q)}\right)\mathbf{X}^{(q)\top}\mathbf{X}^{(q)} = \mathbf{0}_{p\times p}, \quad q \in [Q]_0.$$

This implies

$$\mathcal{B}_{\mathrm{M}}^{(0)} \leq \left\|(\mathbf{I}_p - \mathbf{H}^{(0)})(\boldsymbol{\Sigma}^{(0)} - \hat{\boldsymbol{\Sigma}}^{(0)})(\mathbf{I}_p - \mathbf{H}^{(0)})\right\|\left\|\boldsymbol{\beta}^{(0)}\right\|^2$$
$$\leq \left\|\boldsymbol{\Sigma}^{(0)} - \hat{\boldsymbol{\Sigma}}^{(0)}\right\|\left\|\boldsymbol{\beta}^{(0)}\right\|^2,$$

where $\left\|\mathbf{H}^{(q)}\right\| = \left\|\mathbf{I}_p - \mathbf{H}^{(q)}\right\| = 1$ for all $q \in [Q]_0$. By Lemma 6, there exists a constant $u > 0$ such that with probability at least $1 - 2e^{-\delta}$ for $\delta \geq \log(2)$,

$$\left\|\boldsymbol{\Sigma}^{(0)} - \hat{\boldsymbol{\Sigma}}^{(0)}\right\| \leq u\left(\sqrt{\frac{r_0(\boldsymbol{\Sigma}^{(0)}) + \delta}{n_0}} + \frac{r_0(\boldsymbol{\Sigma}^{(0)}) + \delta}{n_0}\right)\left\|\boldsymbol{\Sigma}^{(0)}\right\|, \tag{15}$$

which completes the proof for the upper bound on $\mathcal{B}_{\mathrm{M}}^{(0)}$; note that the constant $u > 0$ in Lemma 6 depends only on the sub-Gaussian parameter $\nu_x$ in Assumption 1. As in Bartlett et al. [5], we assume that $\nu_x$ is fixed and therefore regard $u$ as a universal constant.

Next, recall that the bias of the TM estimate in Lemma 1 is given by

$$\mathcal{B}_{\mathrm{TM}}^{(q)} = \underbrace{\boldsymbol{\delta}^{(q)\top}\mathbf{H}^{(q)}\boldsymbol{\Pi}^{(0)}\mathbf{H}^{(q)}\boldsymbol{\delta}^{(q)}}_{=:\,U_1\,\geq\,0} + \underbrace{\boldsymbol{\beta}^{(0)\top}(\mathbf{I}_p - \mathbf{H}^{(q)})\boldsymbol{\Pi}^{(0)}(\mathbf{I}_p - \mathbf{H}^{(q)})\boldsymbol{\beta}^{(0)}}_{=:\,U_2\,\geq\,0}$$
$$- 2\boldsymbol{\delta}^{(q)\top}\mathbf{H}^{(q)}\boldsymbol{\Pi}^{(0)}(\mathbf{I}_p - \mathbf{H}^{(q)})\boldsymbol{\beta}^{(0)},$$

where the Cauchy-Schwartz and AM-GM inequalities yield

$$2\left|\boldsymbol{\delta}^{(q)\top}\mathbf{H}^{(q)}\boldsymbol{\Pi}^{(0)}(\mathbf{I}_p - \mathbf{H}^{(q)})\boldsymbol{\beta}^{(0)}\right| \leq 2\sqrt{U_1 U_2} \leq U_1 + U_2,$$

and therefore

$$\mathcal{B}_{\mathrm{TM}}^{(q)} \leq 2(U_1 + U_2).$$

So it suffices to upper-bound the terms $U_1$ and $U_2$, and their concentration inequalities can be obtained similarly. Notice that $U_1$ is bounded above by

$$U_1 = \boldsymbol{\delta}^{(q)\top}\mathbf{H}^{(q)}(\mathbf{I}_p - \mathbf{H}^{(0)})(\boldsymbol{\Sigma}^{(0)} - \hat{\boldsymbol{\Sigma}}^{(0)})(\mathbf{I}_p - \mathbf{H}^{(0)})\mathbf{H}^{(q)}\boldsymbol{\delta}^{(q)}$$
$$\leq \left\|\mathbf{H}^{(q)}(\mathbf{I}_p - \mathbf{H}^{(0)})(\boldsymbol{\Sigma}^{(0)} - \hat{\boldsymbol{\Sigma}}^{(0)})(\mathbf{I}_p - \mathbf{H}^{(0)})\mathbf{H}^{(q)}\right\|\left\|\boldsymbol{\delta}^{(q)}\right\|^2$$
$$\leq \left\|\boldsymbol{\Sigma}^{(0)} - \hat{\boldsymbol{\Sigma}}^{(0)}\right\|\left\|\boldsymbol{\delta}^{(q)}\right\|^2,$$

where the concentration inequality for $\|\boldsymbol{\Sigma}^{(0)} - \hat{\boldsymbol{\Sigma}}^{(0)}\|$ has already been established in inequality (15). Here, we only substitute the variable $\delta$ in (15) by $\eta_1 \geq \log(2)$: with probability at least $1 - 2e^{-\eta_1}$,

$$U_1 \leq u\left(\sqrt{\frac{r_0(\boldsymbol{\Sigma}^{(0)}) + \eta_1}{n_0}} + \frac{r_0(\boldsymbol{\Sigma}^{(0)}) + \eta_1}{n_0}\right)\left\|\boldsymbol{\Sigma}^{(0)}\right\|\left\|\boldsymbol{\delta}^{(q)}\right\|^2. \tag{16}$$

Next, recall that by Assumption 1, there exists a universal constant $C_{\boldsymbol{\Sigma}^{(q)}} > 0$ such that

$$\left\|\boldsymbol{\Sigma}^{(0)}(\boldsymbol{\Sigma}^{(q)})^{-1}\right\| \leq C_{\boldsymbol{\Sigma}^{(q)}},$$

with which we define the following set of matrices:

$$\boldsymbol{\Theta} := \left\{\mathbf{A} \in \mathbb{R}^{p\times p} : \left\|\mathbf{A}\right\| \leq C_{\boldsymbol{\Sigma}^{(q)}}\right\}.$$

Consider any $\mathbf{A} \in \boldsymbol{\Theta}$. We have

$$
\begin{aligned}
\big\| \big( \mathbf{I}_p - \mathbf{H}^{(q)} \big) & \big( \mathbf{I}_p - \mathbf{H}^{(0)} \big) \boldsymbol{\Sigma}^{(0)} \big( \mathbf{I}_p - \mathbf{H}^{(0)} \big) \big( \mathbf{I}_p - \mathbf{H}^{(q)} \big) \big\| \\
&\leq \big\| \big( \mathbf{I}_p - \mathbf{H}^{(q)} \big) \boldsymbol{\Sigma}^{(0)} \big( \mathbf{I}_p - \mathbf{H}^{(q)} \big) \big\| \\
&= \big\| \big( \mathbf{I}_p - \mathbf{H}^{(q)} \big) \big( \boldsymbol{\Sigma}^{(0)} - \mathbf{A} \hat{\boldsymbol{\Sigma}}^{(q)} \big) \big( \mathbf{I}_p - \mathbf{H}^{(q)} \big) \big\| \\
&\leq \min_{\mathbf{A} \in \boldsymbol{\Theta}} \big\| \boldsymbol{\Sigma}^{(0)} - \mathbf{A} \hat{\boldsymbol{\Sigma}}^{(q)} \big\|.
\end{aligned}
$$

This implies

$$
\begin{aligned}
U_2 &= \boldsymbol{\beta}^{(0)\top} \big( \mathbf{I}_p - \mathbf{H}^{(q)} \big) \big( \mathbf{I}_p - \mathbf{H}^{(0)} \big) \boldsymbol{\Sigma}^{(0)} \big( \mathbf{I}_p - \mathbf{H}^{(0)} \big) \big( \mathbf{I}_p - \mathbf{H}^{(q)} \big) \boldsymbol{\beta}^{(0)} \\
&\leq \big\| \big( \mathbf{I}_p - \mathbf{H}^{(q)} \big) \big( \mathbf{I}_p - \mathbf{H}^{(0)} \big) \boldsymbol{\Sigma}^{(0)} \big( \mathbf{I}_p - \mathbf{H}^{(0)} \big) \big( \mathbf{I}_p - \mathbf{H}^{(q)} \big) \big\| \big\| \boldsymbol{\beta}^{(0)} \big\|^2 \\
&\leq \min_{\mathbf{A} \in \boldsymbol{\Theta}} \big\| \boldsymbol{\Sigma}^{(0)} - \mathbf{A} \hat{\boldsymbol{\Sigma}}^{(q)} \big\| \big\| \boldsymbol{\beta}^{(0)} \big\|^2 \\
&\leq \min_{\mathbf{A} \in \boldsymbol{\Theta}} \Big( \big\| \boldsymbol{\Sigma}^{(0)} - \mathbf{A} \boldsymbol{\Sigma}^{(q)} \big\| + \big\| \mathbf{A} \boldsymbol{\Sigma}^{(q)} - \mathbf{A} \hat{\boldsymbol{\Sigma}}^{(q)} \big\| \Big) \big\| \boldsymbol{\beta}^{(0)} \big\|^2 \\
&= \Big\| \boldsymbol{\Sigma}^{(0)} \big( \boldsymbol{\Sigma}^{(q)} \big)^{-1} \big( \boldsymbol{\Sigma}^{(q)} - \hat{\boldsymbol{\Sigma}}^{(q)} \big) \Big\| \big\| \boldsymbol{\beta}^{(0)} \big\|^2 \quad \text{with the minimizer } \mathbf{A}^* = \boldsymbol{\Sigma}^{(0)} \big( \boldsymbol{\Sigma}^{(q)} \big)^{-1} \\
&\leq \big\| \boldsymbol{\Sigma}^{(0)} \big( \boldsymbol{\Sigma}^{(q)} \big)^{-1} \big\| \big\| \boldsymbol{\Sigma}^{(q)} - \hat{\boldsymbol{\Sigma}}^{(q)} \big\| \big\| \boldsymbol{\beta}^{(0)} \big\|^2 \\
&\leq C_{\boldsymbol{\Sigma}^{(q)}} \big\| \boldsymbol{\Sigma}^{(q)} - \hat{\boldsymbol{\Sigma}}^{(q)} \big\| \big\| \boldsymbol{\beta}^{(0)} \big\|^2,
\end{aligned}
$$

and the tail bound on $\big\| \boldsymbol{\Sigma}^{(q)} - \hat{\boldsymbol{\Sigma}}^{(q)} \big\|$ holds similarly by Lemma 6. That is, there exists a universal constant $u > 0$ such that with probability at least $1 - 2e^{-\eta_2}$ for $\eta_2 \geq \log(2)$,

$$
U_2 \leq u \left( \sqrt{\frac{r_0(\boldsymbol{\Sigma}^{(q)}) + \eta_2}{n_q}} + \frac{r_0(\boldsymbol{\Sigma}^{(q)}) + \eta_2}{n_q} \right) C_{\boldsymbol{\Sigma}^{(q)}} \big\| \boldsymbol{\Sigma}^{(q)} \big\| \big\| \boldsymbol{\beta}^{(0)} \big\|^2. \tag{17}
$$

Now, let $\mathcal{U}_1(\eta_1)$ and $\mathcal{U}_2(\eta_2)$ denote the events (16) and (17) respectively, emphasizing the dependency on $\eta_1$ and $\eta_2$ of the corresponding inequalities. Then, it follows that

$$
\mathbb{P}\Big( \mathcal{U}_1(\eta_1) \Big) \geq 1 - 2e^{-\eta_1}, \qquad \mathbb{P}\Big( \mathcal{U}_2(\eta_2) \Big) \geq 1 - 2e^{-\eta_2}.
$$

By taking the union bound, we have

$$
\begin{aligned}
\mathbb{P}\left( \bigcap_{t=1}^{2} \mathcal{U}_t(\eta_t) \right) &\geq 1 - \sum_{t=1}^{2} \mathbb{P}\Big( \mathcal{U}_t(\eta_t)^c \Big) \\
&\geq 1 - 2e^{-\eta_1} - 2e^{-\eta_2}.
\end{aligned}
$$

Taking $\eta_1 \geq \log(4)$, $\eta_2 \geq \log(4)$, and $\eta = -\log\left( \frac{e^{-\eta_1} + e^{-\eta_2}}{2} \right)$ yields

$$
\mathbb{P}\left( \bigcap_{t=1}^{2} \mathcal{U}_t(\eta_t) \right) \geq 1 - 4e^{-\eta}, \quad \eta \geq \log(4).
$$

Thus, with probability at least $1 - 4e^{-\eta}$ for $\eta \geq \log(4)$,

$$
\begin{aligned}
U_1 + U_2 \ \leq \ & u \left( \sqrt{\frac{r_0(\boldsymbol{\Sigma}^{(0)}) + \eta}{n_0}} + \frac{r_0(\boldsymbol{\Sigma}^{(0)}) + \eta}{n_0} \right) \big\| \boldsymbol{\Sigma}^{(0)} \big\| \big\| \boldsymbol{\delta}^{(q)} \big\|^2 \\
& + u \left( \sqrt{\frac{r_0(\boldsymbol{\Sigma}^{(q)}) + \eta}{n_q}} + \frac{r_0(\boldsymbol{\Sigma}^{(q)}) + \eta}{n_q} \right) C_{\boldsymbol{\Sigma}^{(q)}} \big\| \boldsymbol{\Sigma}^{(q)} \big\| \big\| \boldsymbol{\beta}^{(0)} \big\|^2,
\end{aligned}
$$

which completes the proof for the upper bound on $\mathcal{B}_{\mathrm{TM}}^{(q)}$.

**Upper bound on variance inflation.** By Lemma 4, the variance inflation $\mathcal{V}_\uparrow^{(q)}$ is bounded above by

$$
\frac{\sigma_q^2}{n_q} \mathrm{Tr}\left( \hat{\boldsymbol{\Sigma}}^{(q)\dagger} \boldsymbol{\Pi}^{(0)} \right)
$$

$$
= \frac{\sigma_q^2}{n_q} \mathrm{Tr}\left( \left(\boldsymbol{\Sigma}^{(q)}\right)^{1/2} \hat{\boldsymbol{\Sigma}}^{(q)\dagger} \left(\boldsymbol{\Sigma}^{(q)}\right)^{1/2} \left(\boldsymbol{\Sigma}^{(q)}\right)^{-1/2} \boldsymbol{\Pi}^{(0)} \left(\boldsymbol{\Sigma}^{(q)}\right)^{-1/2} \right)
$$

$$
\leq \frac{\sigma_q^2}{n_q} \mathrm{Tr}\left( \hat{\boldsymbol{\Sigma}}^{(q)\dagger} \boldsymbol{\Sigma}^{(q)} \right) \left\| \left(\boldsymbol{\Sigma}^{(q)}\right)^{-1/2} \left(\mathbf{I}_p - \mathbf{H}^{(0)}\right) \left(\boldsymbol{\Sigma}^{(0)} - \hat{\boldsymbol{\Sigma}}^{(0)}\right) \left(\mathbf{I}_p - \mathbf{H}^{(0)}\right) \left(\boldsymbol{\Sigma}^{(q)}\right)^{-1/2} \right\|
$$

$$
\leq \frac{\sigma_q^2}{n_q} \mathrm{Tr}\left( \hat{\boldsymbol{\Sigma}}^{(q)\dagger} \boldsymbol{\Sigma}^{(q)} \right) \left\| \left(\boldsymbol{\Sigma}^{(q)}\right)^{-1/2} \right\|^2 \left\| \boldsymbol{\Sigma}^{(0)} - \hat{\boldsymbol{\Sigma}}^{(0)} \right\|,
$$

where $\left\| \left(\boldsymbol{\Sigma}^{(q)}\right)^{-1/2} \right\|^2 = \lambda_1\left( \left(\boldsymbol{\Sigma}^{(q)}\right)^{-1} \right) = \left( \lambda_p^{(q)} \right)^{-1}$. By Corollary 3, there exist universal constants $b_q, c_q \geq 1$ as specified in Assumption 2 such that with probability at least $1 - 7e^{-n_q/c_q}$,

$$
\frac{\sigma_q^2}{n_q} \mathrm{Tr}\left( \hat{\boldsymbol{\Sigma}}^{(q)\dagger} \boldsymbol{\Sigma}^{(q)} \right) \lesssim c_q \sigma_q^2 \left( \frac{k_q^*}{n_q} + \frac{n_q}{R_{k_q^*}(\boldsymbol{\Sigma}^{(q)})} \right),
$$

and by Lemma 6, with probability at least $1 - 2e^{-\xi}$ for $\xi \geq \log(2)$,

$$
\left\| \boldsymbol{\Sigma}^{(0)} - \hat{\boldsymbol{\Sigma}}^{(0)} \right\| \leq u \left( \sqrt{ \frac{r_0(\boldsymbol{\Sigma}^{(0)}) + \xi}{n_0} } + \frac{r_0(\boldsymbol{\Sigma}^{(0)}) + \xi}{n_0} \right) \left\| \boldsymbol{\Sigma}^{(0)} \right\|.
$$

Combining the results, with probability at least $1 - 7e^{-n_q/c_q} - 2e^{-\xi}$ for $\xi \geq \log(2)$,

$$
\frac{\sigma_q^2}{n_q} \mathrm{Tr}\left( \hat{\boldsymbol{\Sigma}}^{(q)\dagger} \boldsymbol{\Pi}^{(0)} \right)
$$

$$
\lesssim \sigma_q^2 \left( \frac{k_q^*}{n_q} + \frac{n_q}{R_{k_q^*}(\boldsymbol{\Sigma}^{(q)})} \right) \left( \sqrt{ \frac{r_0(\boldsymbol{\Sigma}^{(0)}) + \xi}{n_0} } + \frac{r_0(\boldsymbol{\Sigma}^{(0)}) + \xi}{n_0} \right) \left( \lambda_p^{(q)} \right)^{-1} \left\| \boldsymbol{\Sigma}^{(0)} \right\|.
$$

**Lower bound on excess risk.** Even when the TM estimate induces positive transfer, i.e., $\mathcal{R}_{\mathrm{TM}}^{(q)} < \mathcal{R}_{\mathrm{M}}^{(0)}$, which is sufficient for $\mathcal{B}_{\mathrm{TM}}^{(q)} < \mathcal{B}_{\mathrm{M}}^{(0)}$, its excess risk is bounded below by

$$
\mathcal{R}_{\mathrm{TM}}^{(q)} = \mathcal{B}_{\mathrm{TM}}^{(q)} + \mathcal{V}_{\mathrm{TM}}^{(q)}
$$

$$
= \mathcal{B}_{\mathrm{TM}}^{(q)} + \mathcal{V}_\uparrow^{(q)} + \mathcal{V}_{\mathrm{M}}^{(0)}
$$

$$
\geq \mathcal{V}_{\mathrm{M}}^{(0)},
$$

and by Corollary 3, there exist universal constants $b_0, c_0 \geq 1$ as specified in Assumption 2 such that with probability at least $1 - 10e^{-n_0/c_0}$,

$$
\mathcal{V}_{\mathrm{M}}^{(0)} \asymp \sigma_0^2 \left( \frac{k_0^*}{n_0} + \frac{n_0}{R_{k_0^*}(\boldsymbol{\Sigma}^{(0)})} \right),
$$

which implies $\mathcal{R}_{\mathrm{TM}}^{(q)} \gtrsim \sigma_0^2 \left( \frac{k_0^*}{n_0} + \frac{n_0}{R_{k_0^*}(\boldsymbol{\Sigma}^{(0)})} \right)$ on the same high-probability event. This completes the proof. $\qquad\square$

### C.7 Proof of Corollary 2

Before proceeding to the proof, we illustrate a case where $\tau^*$ in Equation (7) satisfies $\tau^* \ll n_0$. A canonical example with a "benign" spectrum is given by the spiked covariance model [24], where the target covariance $\boldsymbol{\Sigma}^{(0)}$ has eigenvalues $\lambda_1^{(0)} \geq \ldots \geq \lambda_s^{(0)} > \lambda_{s+1}^{(0)} \geq \ldots \geq \lambda_p^{(0)}$ such that

$$
\lambda_j^{(0)} \asymp 1, \quad j \leq s; \qquad \lambda_j^{(0)} \asymp \varepsilon, \quad j > s.
$$

Here, the first $s$ largest eigenvalues are referred to as *spikes*, while the rest $p - s$ eigenvalues are *non-spikes*; a similar structure is considered by Mallinar et al. [42]. Suppose the covariance satisfies $p \gg n_0$, $p\varepsilon \ll n_0$, and $s \ll n_0$. Then, we have $\tau^* = s$ and $\tau^* = k_0^*$ indeed, where $k_0^*$ is as specified in Assumption 2. In this case, $\mathbf{\Sigma}^{(0)}$ is "benign" according to Definition 2, since

$$\frac{r_0(\mathbf{\Sigma}^{(0)})}{n_0} \asymp \frac{s + (p - s)\varepsilon}{n_0} \to 0, \qquad \frac{s}{n_0} \to 0, \qquad \frac{n_0}{R_s(\mathbf{\Sigma}^{(0)})} \asymp \frac{n_0}{p - s} \to 0.$$

We now proceed to the proof of Corollary 2.

*Proof.* Recall the high-probability upper bounds (ignoring constant factors) on the bias and variance inflation of the TM estimate obtained in Theorem 2:

$$\overline{\mathcal{B}} := \left( \sqrt{\frac{r_0(\mathbf{\Sigma}^{(0)}) + \eta}{n_0}} + \frac{r_0(\mathbf{\Sigma}^{(0)}) + \eta}{n_0} \right) \|\mathbf{\Sigma}^{(0)}\| \|\boldsymbol{\delta}^{(q)}\|^2$$

$$+ \left( \sqrt{\frac{r_0(\mathbf{\Sigma}^{(q)}) + \eta}{n_q}} + \frac{r_0(\mathbf{\Sigma}^{(q)}) + \eta}{n_q} \right) C_{\mathbf{\Sigma}^{(q)}} \|\mathbf{\Sigma}^{(q)}\| \|\boldsymbol{\beta}^{(0)}\|^2, \tag{18}$$

$$\overline{\mathcal{V}} := \sigma_q^2 \left( \frac{k_q^*}{n_q} + \frac{n_q}{R_{k_q^*}(\mathbf{\Sigma}^{(q)})} \right) \left( \sqrt{\frac{r_0(\mathbf{\Sigma}^{(0)}) + \xi}{n_0}} + \frac{r_0(\mathbf{\Sigma}^{(0)}) + \xi}{n_0} \right) (\lambda_p^{(q)})^{-1} \|\mathbf{\Sigma}^{(0)}\|. \tag{19}$$

**Case (A) of Corollary 2.** If there is no covariate shift, i.e, $\mathbf{\Sigma}^{(q)} = \mathbf{\Sigma}^{(0)}$, the upper bounds $\overline{\mathcal{B}}$ and $\overline{\mathcal{V}}$ in (18) and (19) are reduced to as follows:

$$\overline{\mathcal{B}}_{\text{HM}} := \left( \sqrt{\frac{r_0(\mathbf{\Sigma}^{(0)}) + \eta}{n_0}} + \frac{r_0(\mathbf{\Sigma}^{(0)}) + \eta}{n_0} \right) \|\mathbf{\Sigma}^{(0)}\| \|\boldsymbol{\delta}^{(q)}\|^2$$

$$+ \left( \sqrt{\frac{r_0(\mathbf{\Sigma}^{(0)}) + \eta}{n_q}} + \frac{r_0(\mathbf{\Sigma}^{(0)}) + \eta}{n_q} \right) \|\mathbf{\Sigma}^{(0)}\| \|\boldsymbol{\beta}^{(0)}\|^2, \tag{20}$$

$$\overline{\mathcal{V}}_{\text{HM}} := \sigma_q^2 \left( \frac{k_0^*}{n_q} + \frac{n_q}{R_{k_0^*}(\mathbf{\Sigma}^{(0)})} \right) \left( \sqrt{\frac{r_0(\mathbf{\Sigma}^{(0)}) + \xi}{n_0}} + \frac{r_0(\mathbf{\Sigma}^{(0)}) + \xi}{n_0} \right) (\lambda_p^{(0)})^{-1} \|\mathbf{\Sigma}^{(0)}\|, \tag{21}$$

where $C_{\mathbf{\Sigma}^{(0)}} = \|\mathbf{\Sigma}^{(0)}(\mathbf{\Sigma}^{(0)})^{-1}\| = 1$ for $\overline{\mathcal{B}}_{\text{HM}}$. Under the covariate shift (A) in Corollary 2, we have

$$\lambda_j^{(q)} = \alpha \lambda_j^{(0)}, \quad j \in [p],$$

for some $\alpha > 1$ and the top $\tau^*$ eigenvectors (corresponding to the $\tau^*$ largest eigenvalues) of $\mathbf{\Sigma}^{(q)}$ and $\mathbf{\Sigma}^{(0)}$ align where $\tau^*$ is defined in Equation (7). Under this covariate shift, the upper bound $\overline{\mathcal{B}}$ in (18) becomes

$$\overline{\mathcal{B}}_{\text{HT}} := \left( \sqrt{\frac{r_0(\mathbf{\Sigma}^{(0)}) + \eta}{n_0}} + \frac{r_0(\mathbf{\Sigma}^{(0)}) + \eta}{n_0} \right) \|\mathbf{\Sigma}^{(0)}\| \|\boldsymbol{\delta}^{(q)}\|^2$$

$$+ \alpha C_{\mathbf{\Sigma}^{(q)}} \left( \sqrt{\frac{r_0(\mathbf{\Sigma}^{(0)}) + \eta}{n_q}} + \frac{r_0(\mathbf{\Sigma}^{(0)}) + \eta}{n_q} \right) \|\mathbf{\Sigma}^{(0)}\| \|\boldsymbol{\beta}^{(0)}\|^2, \tag{22}$$

since

$$r_0(\mathbf{\Sigma}^{(q)}) = \frac{\text{Tr}(\mathbf{\Sigma}^{(q)})}{\|\mathbf{\Sigma}^{(q)}\|} = \frac{\alpha \text{Tr}(\mathbf{\Sigma}^{(0)})}{\alpha \|\mathbf{\Sigma}^{(0)}\|} = r_0(\mathbf{\Sigma}^{(0)}), \qquad \|\mathbf{\Sigma}^{(q)}\| = \alpha \|\mathbf{\Sigma}^{(0)}\|.$$

To derive the upper bound $C_{\mathbf{\Sigma}^{(q)}}$ on $\|\mathbf{\Sigma}^{(0)}(\mathbf{\Sigma}^{(q)})^{-1}\|$, we first recall the spectral decompositions of $\mathbf{\Sigma}^{(0)}$ and $\mathbf{\Sigma}^{(q)}$ in Assumption 1 with simplified notations:

$$\mathbf{\Sigma}^{(0)} = \mathbf{V} \mathbf{\Lambda} \mathbf{V}^\top, \qquad \mathbf{\Sigma}^{(q)} = \mathbf{U}(\alpha \mathbf{\Lambda}) \mathbf{U}^\top,$$

where $\boldsymbol{\Lambda} = \mathrm{Diag}(\lambda_1^{(0)}, \lambda_2^{(0)}, \ldots, \lambda_p^{(0)})$, and $\mathbf{V}^{(q)} \in \mathbb{R}^{p \times p}$ and $\mathbf{U} \in \mathbb{R}^{p \times p}$ are orthogonal matrices whose columns consist of the eigenvectors of $\boldsymbol{\Sigma}^{(0)}$ and $\boldsymbol{\Sigma}^{(q)}$ respectively. We partition $\mathbf{V}$ and $\mathbf{U}$ by their leading $\tau^*$ and the rest $(p - \tau^*)$ columns as follows:

$$\mathbf{V} = \begin{pmatrix} \mathbf{V}_{\tau^*} & \mathbf{V}_{-\tau^*} \end{pmatrix}, \qquad \mathbf{U} = \begin{pmatrix} \mathbf{U}_{\tau^*} & \mathbf{U}_{-\tau^*} \end{pmatrix} = \begin{pmatrix} \mathbf{V}_{\tau^*} & \mathbf{U}_{-\tau^*} \end{pmatrix},$$

where $\mathbf{V}_{\tau^*} \in \mathbb{R}^{p \times \tau^*}$ (resp. $\mathbf{U}_{\tau^*} \in \mathbb{R}^{p \times \tau^*}$) consists of the leading $\tau^*$ eigenvectors of $\boldsymbol{\Sigma}^{(0)}$ (resp. $\boldsymbol{\Sigma}^{(q)}$) and

$$\mathbf{V}_{\tau^*} = \mathbf{U}_{\tau^*}$$

due to the alignment of the leading $\tau^*$ eigenvector pairs of $\boldsymbol{\Sigma}^{(q)}$ and $\boldsymbol{\Sigma}^{(0)}$. For

$$\boldsymbol{\Sigma}^{(0)} \big(\boldsymbol{\Sigma}^{(q)}\big)^{-1} = \mathbf{V}\boldsymbol{\Lambda}\mathbf{V}^{\top}\mathbf{U}(\alpha\boldsymbol{\Lambda})^{-1}\mathbf{U}^{\top} = \frac{1}{\alpha}\mathbf{V}\boldsymbol{\Lambda}\mathbf{V}^{\top}\mathbf{U}\boldsymbol{\Lambda}^{-1}\mathbf{U}^{\top},$$

we have

$$\mathbf{V}^{\top}\mathbf{U} = \begin{pmatrix} \mathbf{V}_{\tau^*}^{\top}\mathbf{U}_{\tau^*} & \mathbf{V}_{\tau^*}^{\top}\mathbf{U}_{-\tau^*} \\ \mathbf{V}_{-\tau^*}^{\top}\mathbf{U}_{\tau^*} & \mathbf{V}_{-\tau^*}^{\top}\mathbf{U}_{-\tau^*} \end{pmatrix} = \begin{pmatrix} \mathbf{I}_{\tau^*} & \mathbf{0}_{\tau^* \times (p-\tau^*)} \\ \mathbf{0}_{(p-\tau^*) \times \tau^*} & \mathbf{V}_{-\tau^*}^{\top}\mathbf{U}_{-\tau^*} \end{pmatrix}.$$

Denoting $\boldsymbol{\Lambda}_{\tau^*} = \mathrm{Diag}(\lambda_1^{(0)}, \ldots, \lambda_{\tau^*}^{(0)})$ and $\boldsymbol{\Lambda}_{-\tau^*} = \mathrm{Diag}(\lambda_{\tau^*+1}^{(0)}, \ldots, \lambda_p^{(0)})$ yields

$$\boldsymbol{\Sigma}^{(0)} \big(\boldsymbol{\Sigma}^{(q)}\big)^{-1} = \frac{1}{\alpha}\mathbf{V}\boldsymbol{\Lambda}\begin{pmatrix} \mathbf{I}_{\tau^*} & \mathbf{0}_{\tau^* \times (p-\tau^*)} \\ \mathbf{0}_{(p-\tau^*) \times \tau^*} & \mathbf{V}_{-\tau^*}^{\top}\mathbf{U}_{-\tau^*} \end{pmatrix}\boldsymbol{\Lambda}^{-1}\mathbf{U}^{\top}$$

$$= \frac{1}{\alpha}\mathbf{V}\begin{pmatrix} \mathbf{I}_{\tau^*} & \mathbf{0}_{\tau^* \times (p-\tau^*)} \\ \mathbf{0}_{(p-\tau^*) \times \tau^*} & \boldsymbol{\Lambda}_{-\tau^*}\mathbf{V}_{-\tau^*}^{\top}\mathbf{U}_{-\tau^*}(\boldsymbol{\Lambda}_{-\tau^*})^{-1} \end{pmatrix}\mathbf{U}^{\top},$$

where

$$\big\|\boldsymbol{\Lambda}_{-\tau^*}\mathbf{V}_{-\tau^*}^{\top}\mathbf{U}_{-\tau^*}(\boldsymbol{\Lambda}_{-\tau^*})^{-1}\big\| \leq \big\|\boldsymbol{\Lambda}_{-\tau^*}\big\|\big\|\mathbf{V}_{-\tau^*}^{\top}\big\|\big\|\mathbf{U}_{-\tau^*}\big\|\big\|(\boldsymbol{\Lambda}_{-\tau^*})^{-1}\big\|$$

$$\leq \frac{\lambda_{\tau^*+1}}{\lambda_p}.$$

Therefore, we have

$$\big\|\boldsymbol{\Sigma}^{(0)} \big(\boldsymbol{\Sigma}^{(q)}\big)^{-1}\big\| \leq \frac{1}{\alpha}\big\|\mathbf{V}\big\|\left\|\begin{pmatrix} \mathbf{I}_{\tau^*} & \mathbf{0}_{\tau^* \times (p-\tau^*)} \\ \mathbf{0}_{(p-\tau^*) \times \tau^*} & \boldsymbol{\Lambda}_{-\tau^*}\mathbf{V}_{-\tau^*}^{\top}\mathbf{U}_{-\tau^*}(\boldsymbol{\Lambda}_{-\tau^*})^{-1} \end{pmatrix}\right\|\big\|\mathbf{U}^{\top}\big\|$$

$$\leq \frac{1}{\alpha}\Big(\big\|\mathbf{I}_\tau^*\big\| \vee \big\|\boldsymbol{\Lambda}_{-\tau^*}\mathbf{V}_{-\tau^*}^{\top}\mathbf{U}_{-\tau^*}(\boldsymbol{\Lambda}_{-\tau^*})^{-1}\big\|\Big)$$

$$\leq \frac{1}{\alpha}\Big(\lambda_{\tau^*+1}^{(0)}/\lambda_p^{(0)}\Big),$$

and by taking $C_{\boldsymbol{\Sigma}^{(q)}} = \alpha^{-1}\big(\lambda_{\tau^*+1}^{(0)}/\lambda_p^{(0)}\big)$, the upper bound $\overline{\mathcal{B}}_{\mathrm{HT}}$ in Equation (22) becomes

$$\overline{\mathcal{B}}_{\mathrm{HT}} = \left(\sqrt{\frac{r_0(\boldsymbol{\Sigma}^{(0)}) + \eta}{n_0}} + \frac{r_0(\boldsymbol{\Sigma}^{(0)}) + \eta}{n_0}\right)\big\|\boldsymbol{\Sigma}^{(0)}\big\|\big\|\boldsymbol{\delta}^{(q)}\big\|^2$$

$$+ \frac{\lambda_{\tau^*+1}^{(0)}}{\lambda_p^{(0)}}\left(\sqrt{\frac{r_0(\boldsymbol{\Sigma}^{(0)}) + \eta}{n_q}} + \frac{r_0(\boldsymbol{\Sigma}^{(0)}) + \eta}{n_q}\right)\big\|\boldsymbol{\Sigma}^{(0)}\big\|\big\|\boldsymbol{\beta}^{(0)}\big\|^2, \qquad (23)$$

where

$$\frac{\lambda_{\tau^*+1}^{(0)}}{\lambda_p^{(0)}} \asymp 1$$

by the construction of $\tau^*$ in Equation (7). Therefore, comparing $\overline{\mathcal{B}}_{\mathrm{HM}}$ in (20) under the homogeneous covariate case and $\overline{\mathcal{B}}_{\mathrm{HT}}$ in (23) under the covariate shift case (A), it follows that

$$\overline{\mathcal{B}}_{\mathrm{HM}} \asymp \overline{\mathcal{B}}_{\mathrm{HT}},$$

i.e., the upper bounds are within a constant factor independent of $\alpha$.

In contrast, under this covariate shift, the upper bound $\overline{\mathcal{V}}$ in (19) is adjusted to $\overline{\mathcal{V}}_{\text{HT}}$ as follows:

$$\overline{\mathcal{V}}_{\text{HT}} := \sigma_q^2 \left( \frac{k_0^*}{n_q} + \frac{n_q}{R_{k_0^*}(\mathbf{\Sigma}^{(0)})} \right) \left( \sqrt{\frac{r_0(\mathbf{\Sigma}^{(0)}) + \xi}{n_0}} + \frac{r_0(\mathbf{\Sigma}^{(0)}) + \xi}{n_0} \right) \frac{\|\mathbf{\Sigma}^{(0)}\|}{\alpha \lambda_p^{(0)}}$$

$$= \frac{1}{\alpha} \overline{\mathcal{V}}_{\text{HM}},$$

since $\mathbf{\Sigma}^{(q)}$ and $\mathbf{\Sigma}^{(0)}$ have the same effective ranks (Definition 1) for all $k \geq 0$, and therefore they share the same minimum

$$k_q^* = k_0^*,$$

provided Assumption 2 for $\mathbf{\Sigma}^{(0)}$. Thus, the upper bound on the variance inflation is reduced by a factor of $\alpha > 1$, when compared to the upper bound $\overline{\mathcal{V}}_{\text{HM}}$ in (21).

To summarize, the following holds for the covariate shift (A) in Corollary 2 when compared to the homogeneous case $\mathbf{\Sigma}^{(q)} = \mathbf{\Sigma}^{(0)}$:

- The upper bound on the bias of the TM estimate remains identical up to a constant factor independent of $\alpha$.

- The upper bound on the variance inflation of the TM estimate is reduced by a factor of $\alpha > 1$.

**Case (B) of Corollary 2.** Suppose $\mathbf{X}^{(q)} = \mathbf{Z}^{(q)}(\mathbf{\Sigma}^{(q)})^{1/2}$ is the $q$-th source design matrix under the homogeneous case where $\mathbf{\Sigma}^{(q)} = \mathbf{\Sigma}^{(0)}$, where $\mathbf{Z}^{(q)}$ is as specified in Assumption 1. Recall that the bias and variance inflation of the TM estimate in Lemma 1 are given by

$$\mathcal{B}_{\text{TM}}^{(q)} = \boldsymbol{\beta}^{(0)\top} (\mathbf{I}_p - \mathbf{H}^{(q)}) \mathbf{\Pi}^{(0)} (\mathbf{I}_p - \mathbf{H}^{(q)}) \boldsymbol{\beta}^{(0)} + \boldsymbol{\delta}^{(q)\top} \mathbf{H}^{(q)} \mathbf{\Pi}^{(0)} \mathbf{H}^{(q)} \boldsymbol{\delta}^{(q)}$$

$$- 2\boldsymbol{\delta}^{(q)\top} \mathbf{H}^{(q)} \mathbf{\Pi}^{(0)} (\mathbf{I}_p - \mathbf{H}^{(q)}) \boldsymbol{\beta}^{(0)},$$

$$\mathcal{V}_{\uparrow}^{(q)} = \sigma_q^2 \text{Tr} \left( \mathbf{X}^{(q)\top} (\mathbf{X}^{(q)} \mathbf{X}^{(q)\top})^{-2} \mathbf{X}^{(q)} \mathbf{\Pi}^{(0)} \right),$$

where

$$\mathbf{H}^{(q)} = \mathbf{X}^{(q)\top} (\mathbf{X}^{(q)} \mathbf{X}^{(q)\top})^{-1} \mathbf{X}^{(q)}.$$

Let $\widetilde{\mathbf{X}}^{(q)} = \mathbf{Z}^{(q)} (\widetilde{\mathbf{\Sigma}}^{(q)})^{1/2}$ be the source design under the covariate shift (B) in Corollary 2 where

$$\widetilde{\mathbf{\Sigma}}^{(q)} = \alpha \mathbf{\Sigma}^{(0)}, \quad \alpha > 1,$$

so that

$$\widetilde{\mathbf{X}}^{(q)} = \alpha^{1/2} \mathbf{X}^{(q)}.$$

Compared to the homogeneity case, the bias $\mathcal{B}_{\text{TM}}^{(q)}$ under this covariate shift remains unchanged, since

$$\widetilde{\mathbf{H}}^{(q)} = \widetilde{\mathbf{X}}^{(q)\top} (\widetilde{\mathbf{X}}^{(q)} \widetilde{\mathbf{X}}^{(q)\top})^{-1} \widetilde{\mathbf{X}}^{(q)} = \mathbf{H}^{(q)}.$$

On the other hand, the variance inflation $\mathcal{V}_{\uparrow}^{(q)}$ is reduced by a factor of $\alpha$, since

$$\widetilde{\mathbf{X}}^{(q)\top} \left( \widetilde{\mathbf{X}}^{(q)} \widetilde{\mathbf{X}}^{(q)\top} \right)^{-2} \widetilde{\mathbf{X}}^{(q)} = \alpha^{-1} \mathbf{X}^{(q)\top} \left( \mathbf{X}^{(q)} \mathbf{X}^{(q)\top} \right)^{-2} \mathbf{X}^{(q)}.$$

To summarize, the following holds for the covariate shift (B) in Corollary 2 when compared to the homogeneous case $\mathbf{\Sigma}^{(q)} = \mathbf{\Sigma}^{(0)}$:

- The exact bias of the TM estimate remains the same.

- The exact variance inflation of the TM estimate in case is reduced by a factor of $\alpha > 1$.

This completes the proof. $\qquad\square$

# D  Computation Algorithm

---

**Algorithm 1:** Informative-Weighted Transfer MNI (WTM)

---

**Input:** all datasets $\{(\mathbf{X}^{(q)}, \mathbf{y}^{(q)})\}_{q=0}^{Q}$, number of folds $K$, detection threshold $\varepsilon_0 > 0$

**Output:** estimated index set of informative sources $\widehat{\mathcal{I}}$, the WTM estimate $\hat{\boldsymbol{\beta}}_{\mathrm{WTM}}$

$\{(\mathbf{X}^{(0)[k]}, \mathbf{y}^{(0)[k]})\}_{k=1}^{K} \leftarrow$ randomly partition the target dataset into $K$ folds of equal size $n_0/K$

**for** $q = 1$ **to** $Q$ **do**

   **for** $k = 1$ **to** $K$ **do**

      $(\mathbf{X}^{(0)[-k]}, \mathbf{y}^{(0)[-k]}) \leftarrow \{(\mathbf{X}^{(0)[k]}, \mathbf{y}^{(0)[k]})\}_{k=1}^{K} \setminus (\mathbf{X}^{(0)[k]}, \mathbf{y}^{(0)[k]})$

      $\hat{\boldsymbol{\beta}}_{\mathrm{M}}^{(0)[-k]} \leftarrow$ MNI trained on the left-out target folds $(\mathbf{X}^{(0)[-k]}, \mathbf{y}^{(0)[-k]})$

      $\hat{\boldsymbol{\beta}}_{\mathrm{TM}}^{(q)[-k]} \leftarrow q$-th TM pre-trained with $(\mathbf{X}^{(q)}, \mathbf{y}^{(q)})$ and fine-tuned on $(\mathbf{X}^{(0)[-k]}, \mathbf{y}^{(0)[-k]})$

      $\widehat{\mathcal{L}}^{[k]}\big(\hat{\boldsymbol{\beta}}_{\mathrm{M}}^{(0)[-k]}\big) \leftarrow \frac{1}{n_0/K}\big\|\mathbf{y}^{(0)[k]} - \mathbf{X}^{(0)[k]}\hat{\boldsymbol{\beta}}_{\mathrm{M}}^{(0)[-k]}\big\|^2$: per-fold loss of target-only MNI

      $\widehat{\mathcal{L}}^{[k]}\big(\hat{\boldsymbol{\beta}}_{\mathrm{TM}}^{(q)[-k]}\big) \leftarrow \frac{1}{n_0/K}\big\|\mathbf{y}^{(0)[k]} - \mathbf{X}^{(0)[k]}\hat{\boldsymbol{\beta}}_{\mathrm{TM}}^{(q)[-k]}\big\|^2$: per-fold loss of $q$-th TM

   **end**

   $\widehat{\mathcal{L}}_{\mathrm{TM}}^{(q)} \leftarrow \frac{1}{K}\sum_{k=1}^{K}\widehat{\mathcal{L}}^{[k]}\big(\hat{\boldsymbol{\beta}}_{\mathrm{TM}}^{(q)[-k]}\big)$: terminal CV loss of $q$-th TM

**end**

$\widehat{\mathcal{L}}_{\mathrm{M}}^{(0)} \leftarrow \frac{1}{K}\sum_{k=1}^{K}\widehat{\mathcal{L}}^{[k]}\big(\hat{\boldsymbol{\beta}}_{\mathrm{M}}^{(0)[-k]}\big)$: terminal CV loss of target-only MNI

$\hat{\sigma}_{\mathcal{L}} \leftarrow \sqrt{\frac{1}{K-1}\sum_{k=1}^{K}\big(\widehat{\mathcal{L}}^{[k]}(\hat{\boldsymbol{\beta}}_{\mathrm{M}}^{(0)[-k]}) - \widehat{\mathcal{L}}_{\mathrm{M}}^{(0)}\big)^2}$

$\widehat{\mathcal{I}} \leftarrow \big\{q \in [Q] : \widehat{\mathcal{L}}_{\mathrm{TM}}^{(q)} - \widehat{\mathcal{L}}_{\mathrm{M}}^{(0)} \le \varepsilon_0(\hat{\sigma}_{\mathcal{L}} \vee 0.01)\big\}$

**if** $\widehat{\mathcal{I}} = \emptyset$, *i.e., no informative source is detected* **then**

   $\hat{\boldsymbol{\beta}}_{\mathrm{WTM}} \leftarrow \hat{\boldsymbol{\beta}}_{\mathrm{M}}^{(0)}$: no knowledge transfer

**else**

   **for** $i \in \widehat{\mathcal{I}}$ **do**

      $\hat{\boldsymbol{\beta}}_{\mathrm{TM}}^{(i)} \leftarrow i$-th TM pre-trained with informative $(\mathbf{X}^{(i)}, \mathbf{y}^{(i)})$ and fine-tuned on $(\mathbf{X}^{(0)}, \mathbf{y}^{(0)})$

      $w_i \leftarrow \big(\widehat{\mathcal{L}}_0^{(i)}\big)^{-1}$: weight initialized by the inverse CV loss of $i$-th TM

   **end**

   $w_i \leftarrow w_i/\big(\sum_{i \in \widehat{\mathcal{I}}} w_i\big)$ for all $i \in \widehat{\mathcal{I}}$: weight normalization

**end**

$\hat{\boldsymbol{\beta}}_{\mathrm{WTM}} \leftarrow \sum_{i \in \widehat{\mathcal{I}}} w_i \hat{\boldsymbol{\beta}}_{\mathrm{TM}}^{(i)}$

---

# E  Additional Numerical Experiments

This section presents additional experimental results along with their implications. The experimental setups are identical in Section 5. The average excess risk over 50 independent simulations is plotted against each value of $p \in \{300, 400, \ldots, 1000\}$. When implementing Algorithm 1, we set $K = 5$ and $\varepsilon_0 = 1/2$. As for the ground truth in Equation (1) given by

$$\mathbf{y}^{(q)} = \mathbf{X}^{(q)}\boldsymbol{\beta}^{(q)} + \boldsymbol{\epsilon}^{(q)} = \Big(\mathbf{Z}^{(q)}(\boldsymbol{\Sigma}^{(q)})^{1/2}\Big)\Big(\boldsymbol{\beta}^{(0)} + \boldsymbol{\delta}^{(q)}\Big) + \boldsymbol{\epsilon}^{(q)}, \quad q \in [Q]_0,$$

where $\boldsymbol{\delta}^{(0)} \equiv \mathbf{0}_p$, all parametric configurations are as specified in Section 5. See Appendix F.1 for more details on the generating process of the deterministic parameters $\{\boldsymbol{\beta}^{(0)}, (\boldsymbol{\delta}^{(q)}, \boldsymbol{\Sigma}^{(q)})\}_{q=0}^{Q}$.

## E.1  Benign Overfitting Experiments

Here, we replicate the experimental setup used for Figure 1.(a) under benign overfitting, but now compare to three single-source versions of the pooled-MNI [56]. Specifically, for each of $Q = 3$

source datasets, we employ the following forms of pooled-MNI:

$$\hat{\boldsymbol{\beta}}_{\text{PM}}^{(1)} := \arg\min_{\boldsymbol{\beta}\in\mathbb{R}^p}\Big\{\|\boldsymbol{\beta}\| : \mathbf{X}^{(q)}\boldsymbol{\beta} = \mathbf{y}^{(q)}, \quad q \in \{0,1\}\Big\},$$

$$\hat{\boldsymbol{\beta}}_{\text{PM}}^{(2)} := \arg\min_{\boldsymbol{\beta}\in\mathbb{R}^p}\Big\{\|\boldsymbol{\beta}\| : \mathbf{X}^{(q)}\boldsymbol{\beta} = \mathbf{y}^{(q)}, \quad q \in \{0,2\}\Big\},$$

$$\hat{\boldsymbol{\beta}}_{\text{PM}}^{(3)} := \arg\min_{\boldsymbol{\beta}\in\mathbb{R}^p}\Big\{\|\boldsymbol{\beta}\| : \mathbf{X}^{(q)}\boldsymbol{\beta} = \mathbf{y}^{(q)}, \quad q \in \{0,3\}\Big\}.$$

That is, we consider each pooled-MNI as a single-source transfer task, by having each $\hat{\boldsymbol{\beta}}_{\text{PM}}^{(q)}$ interpolate the target and only the $q$-th source.

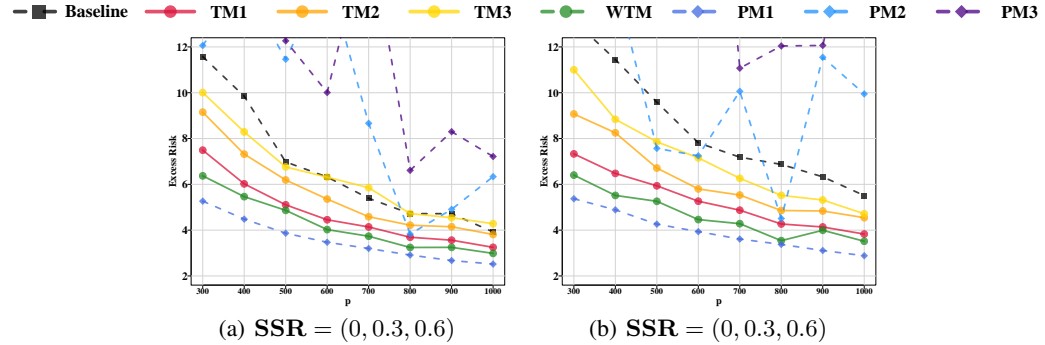

Figure 3: We compare the performance of our proposed methods to single-source-pooled-MNIs under model shift, with the same setting as Figure 1.(a). Figure (a) re-uses the exact seeds for Figure 1.(a), so the excess risk curves for $\hat{\boldsymbol{\beta}}_{\text{M}}^{(0)}$, $\hat{\boldsymbol{\beta}}_{\text{TM}}^{(q)}$, and $\hat{\boldsymbol{\beta}}_{\text{WTM}}$ are identical across the two figures. In contrast, (b) uses distinct seeds, hence its curves differ from those in Figure 1.(a).

In Fig. 3, the first source has no distribution shift, as $\boldsymbol{\Sigma}^{(1)} = \boldsymbol{\Sigma}^{(0)} = \boldsymbol{\Lambda}^{(0)}$ where $\boldsymbol{\Lambda}^{(0)}$ is as specified in Section 5 and $\text{SSR}_1 = 0$. While $\hat{\boldsymbol{\beta}}_{\text{TM}}^{(1)}$ pre-trained with the first source clearly outperforms the target-only MNI, the first-source-pooled-MNI $\hat{\boldsymbol{\beta}}_{\text{PM}}^{(1)}$ even performs better. This confirms the strength of the early-fusion approach of Song et al. [56] for leveraging source data when the source and target distributions are exactly identical.

However, the effectiveness of our approach is much more pronounced in terms of its robustness to model shift. Once model shift is introduced for the second and third sources with $\text{SSR}_2 = 0.3$ and $\text{SSR}_3 = 0.6$ respectively, the depicted curves change markedly. Both corresponding pooled-MNIs, $\hat{\boldsymbol{\beta}}_{\text{PM}}^{(2)}$ and $\hat{\boldsymbol{\beta}}_{\text{PM}}^{(3)}$, not only undergo negative transfer but also show unstable convergence in excess risk. On the other hand, both TM estimates $\hat{\boldsymbol{\beta}}_{\text{TM}}^{(2)}$ and $\hat{\boldsymbol{\beta}}_{\text{TM}}^{(3)}$ exhibit steadily decreasing excess risk with $\hat{\boldsymbol{\beta}}_{\text{TM}}^{(2)}$ consistently inducing positive transfer. Even under model shift severe as $\text{SSR}_3 = 0.6$, $\hat{\boldsymbol{\beta}}_{\text{TM}}^{(3)}$ still matches (Fig. 3.(a)) or surpasses (Fig. 3.(b)) the target-only MNI. Overall, under the present experimental design, the model shift severe as $\text{SSR} = 0.6$ appears to mark a threshold for the TM estimate in inducing positive transfer, whereas it deteriorates the performance of pooled-MNI below the target-only baseline.

### E.2  Harmless Interpolation Experiments

We extend the experiments in Section 5.2 to the case where $Q = 1$ and the target sample size varies. We first set $n_1$ to be the optimal transfer size $n_1^*$ for $\hat{\boldsymbol{\beta}}_{\text{TM}}^{(1)}$ as identified in Corollary 1:

$$n_1^* = \left\lfloor p - 1 - \sqrt{\frac{p(p-1)}{\text{SNR}(1 - \text{SSR}_1)}} \right\rfloor, \tag{24}$$

where we use the floor function to ensure the optimum be an integer.

While $n_0 = 50$ is fixed throughout Section 5.2, we now optimize $n_0$ for the pooled-MNI $\hat{\boldsymbol{\beta}}_{\text{PM}}$ as suggested by Song et al. [56]. Specifically, under the same isotropic Gaussian setting as our

experiment, Corollary 3.4 in Song et al. [56] gives

$$n_0^* = \underset{n_0 : \, n_0 + n_1^* < p}{\arg\min} \, \mathcal{R}(\hat{\boldsymbol{\beta}}_{\mathrm{PM}}) = \left\lfloor p - n_1^* - \sqrt{\frac{p^2}{\mathrm{SNR}} + n_1^* \cdot \mathrm{SSR}_1} \right\rfloor \vee 0. \tag{25}$$

That is, given each value of $p$, $n_1^*$, SNR, and $\mathrm{SSR}_1$, the optimal target sample size that minimizes the excess risk of $\hat{\boldsymbol{\beta}}_{\mathrm{PM}}$ is determined by $n_0^*$, which can be zero. The constraint in Equation (25) ensures that the total training sample size $n_0^* + n_1^*$ is less than $p$, so $\hat{\boldsymbol{\beta}}_{\mathrm{PM}}$ always operates in an overparameterized regime enabling interpolation.

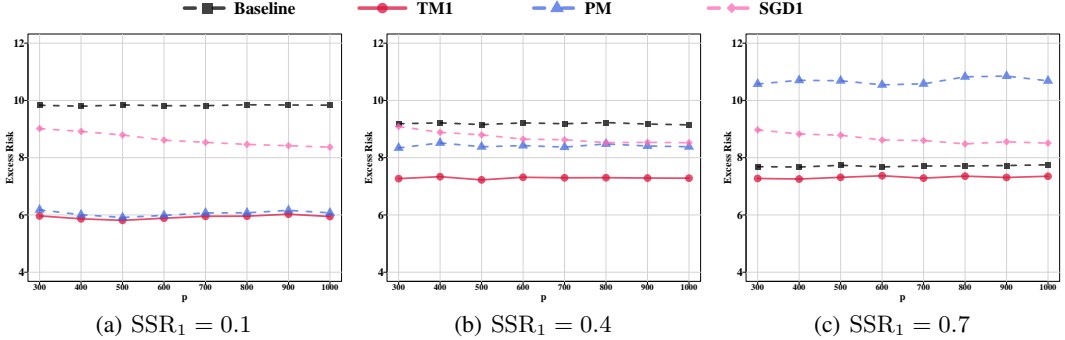

(a) $\mathrm{SSR}_1 = 0.1$      (b) $\mathrm{SSR}_1 = 0.4$      (c) $\mathrm{SSR}_1 = 0.7$

Figure 4: As in Figure 2.(a), the target and source covariates are i.i.d. $\mathcal{N}(\mathbf{0}_p, \mathbf{I}_p)$ with SNR $= 10$. Once $n_1 = n_1^*$ is determined by (24), the corresponding target sample size $n_0^*$ follows from (25).

Table 1: Optimal target and source sample sizes $(n_0^*, n_1^*)$ for each value of $p$ in Figure 4

| Figure | Optimality | Sample size for each $p \in \{300, 400, \dots, 1000\}$ | | | | | | | |
|---|---|---|---|---|---|---|---|---|---|
| (a) | $n_0^*$ | 6 | 8 | 9 | 11 | 13 | 14 | 16 | 18 |
| | $n_1^*$ | 199 | 265 | 332 | 399 | 465 | 532 | 599 | 665 |
| (b) | $n_0^*$ | 28 | 38 | 46 | 55 | 65 | 74 | 84 | 93 |
| | $n_1^*$ | 176 | 235 | 295 | 354 | 413 | 472 | 531 | 590 |
| (c) | $n_0^*$ | 78 | 105 | 131 | 157 | 183 | 209 | 235 | 262 |
| | $n_1^*$ | 126 | 168 | 210 | 252 | 295 | 337 | 379 | 421 |

We observe an interesting pattern from Table 1: for each value of $p$, the total sample size $n_0^* + n_1^*$ remains identical across Figures 4.(a)-(c). (In some instances, the totals differ by 1 due to the use of floor function when computing $n_0^*$ and $n_1^*$.) To provide context, once $n_1^*$ source samples are provided, $\hat{\boldsymbol{\beta}}_{\mathrm{PM}}$ appears to select an appropriate number of target samples $n_0^*$ to balance the overall quality of the training inputs. Specifically, in Fig. 4.(a), the pooled-MNI already has a large amount of high-quality inputs with SSR $= 0.1$ obtained from the source, so it requires fewer additional target samples. In contrast, in Fig. 4.(c), the training inputs are highly "corrupted" by source data of poor quality with SSR $= 0.7$, and consequently, the model pools much more target samples to "dilute" the corruption. This results in a significant increase in $n_0^*$ compared to Fig. 4.(a). This decision-making process of the pooled-MNI appears natural, considering that the estimate is an early-fusion approach [56], where the source is integrated at the input level and then jointly learned with the target data.

When comparing $\hat{\boldsymbol{\beta}}_{\mathrm{TM}}^{(1)}$ with $\hat{\boldsymbol{\beta}}_{\mathrm{PM}}$, their performances are nearly identical in Fig. 4.(a), with our method performing slightly better. In fact, if SSR $= 0$, i.e., no model shift, we have $n_0^* = 0$ by Equation (25). In that case, both PM and TM estimates are reduced to the source-only MNI $\hat{\boldsymbol{\beta}}_{\mathrm{M}}^{(1)}$ trained on $n_1^*$ samples, resulting in identical performance. However, once under model shift, $\hat{\boldsymbol{\beta}}_{\mathrm{TM}}^{(1)}$ is consistently superior even when the model shift is so low that it is favorable for $\hat{\boldsymbol{\beta}}_{\mathrm{PM}}$, as illustrated in Fig. 4.(a) where SSR $= 0.1$. The superior performance of $\hat{\boldsymbol{\beta}}_{\mathrm{TM}}^{(1)}$ is further highlighted in Fig. 4.(c) where model shift is severe as SSR $= 0.7$, as it is the only method that consistently induces positive transfer.

Throughout Fig. 4, the performance of SGD remains consistent regardless of the degree of model shift. This is plausible because, while the total sample size $n_0^* + n_1^*$ is fixed for each value of $p$, the estimate pre-trains with fewer source samples and instead fine-tunes with more target samples in proportion to the degree of model shift.

## F    Experimental Details

This section provides further details on the experiments presented in Section 5 and Appendix E. All numerical experiments are conducted by R using a standard laptop (ASUS ZenBook UX331UN, Intel(R) Core(TM) i7-8550U CPU @ 1.80GHz, 16 GB 2133 MHz DDR3). All sourcecodes we release are scripted from scratch, relying on standard R packages such as MASS and mvtnorm.

For the evaluation metric, we compute the excess risk of an estimate $\hat{\boldsymbol{\beta}}_t$ trained at the $t$-th simulation run by

$$\left(\hat{\boldsymbol{\beta}}_t - \boldsymbol{\beta}^{(0)}\right)^\top \boldsymbol{\Sigma}^{(0)}\left(\hat{\boldsymbol{\beta}}_t - \boldsymbol{\beta}^{(0)}\right),$$

for $t \in [50]$, where 50 is the number of independent simulations conducted. The deterministic parameters $\boldsymbol{\beta}^{(0)}$ and $\boldsymbol{\Sigma}^{(0)}$ are generated once and kept fixed across the 50 simulations, guaranteeing they are agnostic to the replicate index $t$. We then report the average excess risk

$$\frac{1}{50}\sum_{t=1}^{50}\left(\hat{\boldsymbol{\beta}}_t - \boldsymbol{\beta}^{(0)}\right)^\top \boldsymbol{\Sigma}^{(0)}\left(\hat{\boldsymbol{\beta}}_t - \boldsymbol{\beta}^{(0)}\right).$$

To ensure the stable convergence of SGD [69] we test as a benchmark transfer task, we tune the initial pre-training and fine-tuning learning rates for each experimental setup. While Wu et al. [69] adopted an initial learning rate of $0.1$ (both for pre-training and fine-tuning) in their experiments, we find this value leads to divergence across all of our experimental setups. Accordingly, we reduce the learning rates based on preliminary implementations; for example, we use $0.001$ for both pre-training and fine-tuning in Fig. 2.(a).

### F.1    Parameter Generation Process

In the formulation of the ground truth model

$$\mathbf{y}^{(q)} = \left(\mathbf{Z}^{(q)}(\boldsymbol{\Sigma}^{(q)})^{1/2}\right)\left(\boldsymbol{\beta}^{(0)} + \boldsymbol{\delta}^{(q)}\right) + \boldsymbol{\epsilon}^{(q)}, \quad q \in [Q]_0,$$

where $\boldsymbol{\delta}^{(0)} \equiv \mathbf{0}_p$, only the random quantities $\{(\mathbf{Z}^{(q)}, \boldsymbol{\epsilon}^{(q)})\}_{q=0}^Q$ vary across the 50 simulations. The deterministic parameters $\{\boldsymbol{\beta}^{(0)}, (\boldsymbol{\delta}^{(q)}, \boldsymbol{\Sigma}^{(q)})\}_{q=0}^Q$ remain fixed across all runs. While these fixed quantities are sampled from specific probability distributions, we generate them using a fixed random seed, thereby ensuring their deterministic nature.

**Model coefficients.** As specified in Section 5, the target coefficient $\boldsymbol{\beta}^{(0)}$ and the contrast vectors $\{\boldsymbol{\delta}^{(q)}\}_{q=1}^Q$ are given by

$$\boldsymbol{\beta}^{(0)} = S^{1/2}\mathbf{u}^{(0)}, \qquad \boldsymbol{\delta}^{(q)} = (\mathrm{SSR}_q \cdot S)^{1/2}\mathbf{u}^{(q)},$$

where $(S, \mathrm{SSR}_q)$ are pre-specified hyperparameters. Each $p$-dimensional vector $\mathbf{u}^{(q)}$ is independently sampled from $\mathcal{N}(\mathbf{0}_p, \mathbf{I}_p)$ and then normalized so that $\|\mathbf{u}^{(q)}\| = 1$ for all $q \in [Q]_0$. Each $\mathbf{u}^{(q)}$ is held fixed across the 50 replications.

**Spectral decomposition of covariance.** Random sampling for covariance generation is used only in producing Figures 1.(b) and 1.(c). In these figures, the source covariances are defined via the spectral decomposition

$$\boldsymbol{\Sigma}^{(q)} = \mathbf{V}^{(q)}\boldsymbol{\Lambda}^{(q)}\mathbf{V}^{(q)\top}, \quad q \in [Q],$$

where $\{\boldsymbol{\Lambda}^{(q)}\}_{q=1}^Q$ are fixed as specified in Section 5, and $\{\mathbf{V}^{(q)}\}_{q=1}^Q$ are independently sampled from the orthogonal group $\mathcal{O}_p := \{\mathbf{Q} \in \mathbb{R}^{p \times p} : \mathbf{Q}^\top\mathbf{Q} = \mathbf{Q}\mathbf{Q}^\top = \mathbf{I}_p\}$. Specifically, we first independently sample the matrix $\mathbf{G}^{(q)} \in \mathbb{R}^{p \times p}$ with i.i.d. standard Gaussian entries (which is then fixed across the 50 simulations), perform its QR decomposition $\mathbf{G}^{(q)} = \mathbf{Q}^{(q)}\mathbf{R}^{(q)}$ and assign $\mathbf{V}^{(q)} = \mathbf{Q}^{(q)}$. Consequently, the matrices $\{\mathbf{V}^{(q)}\}_{q=1}^Q$ are held fixed.

## F.2 Variability of Experiment Results

Table 2: Mean and standard deviation (in parenthesis) of excess risk

| Figure | Estimate | $p$ | | | | | | | |
|---|---|---|---|---|---|---|---|---|---|
| | | 300 | 400 | 500 | 600 | 700 | 800 | 900 | 1000 |
| 1.(a) | Target-only MNI | 11.577 (1.686) | 9.846 (1.734) | 6.978 (1.008) | 6.321 (1.187) | 5.389 (0.796) | 4.721 (1.030) | 4.719 (0.743) | 3.920 (0.701) |
| | TM1 | 7.491 (1.396) | 6.021 (1.019) | 5.105 (0.915) | 4.453 (0.713) | 4.140 (0.709) | 3.690 (0.687) | 3.563 (0.503) | 3.249 (0.560) |
| | TM2 | 9.151 (1.602) | 7.318 (1.404) | 6.191 (1.153) | 5.356 (0.905) | 4.586 (0.812) | 4.216 (0.713) | 4.148 (0.688) | 3.810 (0.695) |
| | TM3 | 10.006 (2.021) | 8.289 (1.378) | 6.760 (1.128) | 6.313 (1.164) | 5.855 (0.787) | 4.724 (0.934) | 4.543 (0.795) | 4.282 (0.678) |
| | WTM | 6.370 (1.423) | 5.462 (1.154) | 4.865 (1.083) | 4.024 (0.584) | 3.735 (0.382) | 3.244 (0.553) | 3.249 (0.469) | 2.986 (0.882) |
| | Pooled-MNI | 71.942 (16.986) | 39.543 (3.745) | 12.258 (1.772) | 12.788 (1.184) | 11.343 (1.484) | 4.805 (0.665) | 5.795 (0.857) | 5.638 (0.582) |
| | SGD1 | 14.864 (1.379) | 11.359 (1.069) | 8.749 (0.957) | 7.558 (0.742) | 6.421 (0.467) | 5.627 (0.581) | 4.956 (0.427) | 4.470 (0.447) |
| | SGD2 | 16.129 (1.751) | 13.576 (1.983) | 9.251 (1.024) | 9.444 (1.322) | 7.095 (1.020) | 5.521 (0.608) | 4.986 (0.471) | 5.519 (0.937) |
| | SGD3 | 18.504 (3.544) | 18.032 (3.918) | 10.328 (1.336) | 7.729 (0.970) | 7.766 (1.359) | 5.744 (0.789) | 6.384 (1.169) | 5.488 (0.683) |
| 1.(b) | Target-only MNI | 11.577 (1.686) | 9.846 (1.734) | 6.978 (1.008) | 6.321 (1.187) | 5.389 (0.796) | 4.721 (1.030) | 4.719 (0.743) | 3.920 (0.701) |
| | TM1 | 9.744 (1.836) | 7.975 (1.395) | 6.638 (1.179) | 5.591 (0.975) | 5.024 (0.832) | 4.531 (0.861) | 4.198 (0.741) | 3.858 (0.624) |
| | TM2 | 9.416 (1.695) | 7.695 (1.469) | 6.772 (1.217) | 6.047 (1.073) | 5.066 (0.771) | 4.592 (0.949) | 4.467 (0.717) | 3.860 (0.616) |
| | TM3 | 20.849 (4.591) | 14.105 (2.828) | 10.661 (1.985) | 8.655 (1.765) | 6.819 (1.165) | 5.995 (1.067) | 5.352 (0.938) | 4.600 (0.778) |
| | WTM | 8.648 (1.404) | 7.614 (1.858) | 6.288 (0.865) | 5.638 (1.115) | 4.853 (0.649) | 4.457 (1.181) | 4.464 (0.754) | 3.614 (0.478) |
| | Pooled-MNI | 75.355 (22.828) | 22.625 (4.058) | 14.484 (4.170) | 9.085 (1.799) | 7.697 (1.164) | 6.592 (1.320) | 5.476 (0.948) | 4.884 (0.738) |
| | SGD1 | 20.011 (2.690) | 14.223 (1.551) | 11.006 (1.328) | 9.630 (1.108) | 8.247 (0.963) | 6.970 (0.877) | 6.293 (0.616) | 5.710 (0.606) |
| | SGD2 | 19.957 (2.149) | 15.045 (1.341) | 11.291 (1.136) | 9.973 (1.102) | 8.543 (0.817) | 7.317 (0.835) | 6.500 (0.602) | 5.824 (0.531) |
| | SGD3 | 20.247 (2.226) | 15.198 (1.301) | 11.462 (1.135) | 9.916 (1.092) | 8.545 (0.820) | 7.332 (0.831) | 6.515 (0.591) | 5.763 (0.529) |
| 1.(c) | Target-only MNI | 11.577 (1.686) | 9.846 (1.734) | 6.978 (1.008) | 6.321 (1.187) | 5.389 (0.796) | 4.721 (1.030) | 4.719 (0.743) | 3.920 (0.701) |
| | TM1 | 9.735 (1.845) | 7.977 (1.397) | 6.638 (1.181) | 5.589 (0.972) | 5.021 (0.833) | 4.531 (0.862) | 4.198 (0.741) | 3.857 (0.622) |
| | TM2 | 9.361 (1.680) | 7.657 (1.469) | 6.747 (1.219) | 6.038 (1.072) | 5.048 (0.774) | 4.570 (0.940) | 4.454 (0.708) | 3.840 (0.621) |
| | TM3 | 11.031 (2.251) | 8.763 (1.668) | 7.170 (1.170) | 6.295 (1.104) | 5.361 (0.873) | 4.720 (0.832) | 4.392 (0.721) | 3.813 (0.564) |
| | WTM | 8.296 (1.351) | 7.357 (1.671) | 6.018 (0.846) | 5.480 (1.076) | 4.814 (0.667) | 4.238 (0.956) | 4.376 (0.783) | 3.541 (0.521) |

| | | | | | | | | |
|---|---|---|---|---|---|---|---|---|
| | Pooled-MNI | 26.400 (6.839) | 9.480 (1.586) | 7.341 (1.525) | 5.910 (1.004) | 5.306 (0.727) | 4.540 (0.842) | 4.373 (0.782) | 3.777 (0.464) |
| | SGD1 | 20.181 (2.567) | 14.498 (1.478) | 11.164 (1.275) | 9.715 (1.083) | 8.336 (0.918) | 7.060 (0.844) | 6.353 (0.597) | 5.772 (0.623) |
| | SGD2 | 20.174 (2.162) | 15.208 (1.355) | 11.422 (1.140) | 10.029 (1.114) | 8.603 (0.813) | 7.379 (0.825) | 6.542 (0.593) | 5.883 (0.591) |
| | SGD3 | 20.405 (2.225) | 15.328 (1.320) | 11.561 (1.140) | 9.983 (1.106) | 8.606 (0.818) | 7.391 (0.822) | 6.554 (0.585) | 5.833 (0.594) |
| 2.(a) | Target-only MNI | 8.510 (0.259) | 8.839 (0.299) | 9.166 (0.169) | 9.249 (0.119) | 9.346 (0.135) | 9.440 (0.130) | 9.490 (0.124) | 9.555 (0.091) |
| | TM1 | 7.790 (0.426) | 8.344 (0.291) | 8.698 (0.292) | 8.808 (0.252) | 9.014 (0.251) | 9.112 (0.186) | 9.222 (0.167) | 9.314 (0.170) |
| | TM2 | 6.715 (0.555) | 7.146 (0.466) | 7.253 (0.445) | 7.318 (0.335) | 7.403 (0.318) | 7.407 (0.324) | 7.555 (0.320) | 7.606 (0.318) |
| | WTM | 5.526 (1.040) | 5.831 (0.900) | 5.962 (0.797) | 5.923 (0.951) | 6.213 (0.936) | 6.251 (1.006) | 6.363 (0.959) | 6.342 (1.051) |
| | Pooled-MNI | 33.144 (6.900) | 15.647 (2.941) | 12.361 (1.540) | 11.092 (1.204) | 9.842 (0.580) | 9.663 (0.697) | 9.413 (0.573) | 9.117 (0.266) |
| | SGD1 | 9.424 (0.103) | 9.445 (0.110) | 9.506 (0.118) | 9.496 (0.103) | 9.531 (0.108) | 9.549 (0.080) | 9.603 (0.098) | 9.619 (0.093) |
| | SGD2 | 8.886 (0.160) | 8.833 (0.182) | 8.786 (0.139) | 8.626 (0.149) | 8.676 (0.161) | 8.613 (0.160) | 8.660 (0.135) | 8.655 (0.152) |
| 2.(b) | Target-only MNI | 8.510 (0.259) | 8.839 (0.299) | 9.166 (0.169) | 9.249 (0.119) | 9.346 (0.135) | 9.440 (0.130) | 9.490 (0.124) | 9.555 (0.091) |
| | TM1 | 7.640 (0.413) | 8.219 (0.288) | 8.607 (0.289) | 8.741 (0.239) | 8.958 (0.251) | 9.059 (0.186) | 9.168 (0.171) | 9.273 (0.168) |
| | TM2 | 5.654 (0.439) | 6.016 (0.359) | 6.083 (0.322) | 6.147 (0.275) | 6.253 (0.268) | 6.255 (0.263) | 6.343 (0.253) | 6.387 (0.273) |
| | WTM | 4.858 (0.651) | 5.290 (0.677) | 5.313 (0.243) | 5.239 (0.307) | 5.535 (0.268) | 5.520 (0.280) | 5.774 (0.759) | 5.901 (1.133) |
| | Pooled-MNI | 24.454 (5.173) | 12.022 (2.586) | 9.331 (1.132) | 8.486 (0.822) | 7.776 (0.541) | 7.576 (0.550) | 7.481 (0.334) | 7.266 (0.287) |
| | SGD1 | 9.468 (0.094) | 9.482 (0.101) | 9.540 (0.106) | 9.524 (0.096) | 9.554 (0.099) | 9.565 (0.076) | 9.607 (0.093) | 9.623 (0.085) |
| | SGD2 | 9.015 (0.143) | 8.957 (0.156) | 8.905 (0.122) | 8.745 (0.134) | 8.764 (0.142) | 8.679 (0.144) | 8.692 (0.123) | 8.640 (0.137) |
| 2.(c) | Target-only MNI | 8.525 (0.261) | 8.830 (0.302) | 9.178 (0.168) | 9.251 (0.117) | 9.345 (0.137) | 9.435 (0.137) | 9.491 (0.128) | 9.551 (0.089) |
| | TM1 | 7.647 (0.420) | 8.192 (0.281) | 8.592 (0.271) | 8.747 (0.247) | 8.960 (0.256) | 9.059 (0.177) | 9.166 (0.193) | 9.261 (0.158) |
| | TM2 | 4.813 (0.376) | 5.073 (0.271) | 5.142 (0.343) | 5.222 (0.259) | 5.366 (0.333) | 5.297 (0.243) | 5.368 (0.228) | 5.451 (0.246) |
| | WTM | 3.943 (0.864) | 4.178 (0.902) | 4.279 (0.937) | 3.974 (0.335) | 4.328 (0.359) | 4.153 (0.316) | 4.188 (0.190) | 4.058 (0.244) |
| 3.(a) | Target-only MNI | 11.577 (1.686) | 9.846 (1.734) | 6.978 (1.008) | 6.321 (1.187) | 5.389 (0.796) | 4.721 (1.030) | 4.719 (0.743) | 3.920 (0.701) |
| | TM1 | 7.491 (1.396) | 6.021 (1.019) | 5.105 (0.915) | 4.453 (0.713) | 4.140 (0.709) | 3.690 (0.687) | 3.563 (0.503) | 3.249 (0.560) |
| | TM2 | 9.151 (1.602) | 7.318 (1.404) | 6.191 (1.153) | 5.356 (0.905) | 4.586 (0.812) | 4.216 (0.713) | 4.148 (0.688) | 3.810 (0.695) |
| | TM3 | 10.006 (2.021) | 8.289 (1.378) | 6.760 (1.128) | 6.313 (1.164) | 5.855 (0.787) | 4.724 (0.934) | 4.543 (0.795) | 4.282 (0.678) |

| | | | | | | | | |
|---|---|---|---|---|---|---|---|---|
| | WTM | 6.370 (1.423) | 5.462 (1.154) | 4.865 (1.083) | 4.024 (0.584) | 3.735 (0.382) | 3.244 (0.553) | 3.249 (0.469) | 2.986 (0.882) |
| | Pooled-MNI1 | 5.266 (0.540) | 4.481 (0.435) | 3.867 (0.459) | 3.470 (0.389) | 3.199 (0.318) | 2.913 (0.380) | 2.672 (0.283) | 2.519 (0.299) |
| | Pooled-MNI2 | 12.062 (1.703) | 14.689 (1.827) | 11.461 (1.627) | 15.202 (1.983) | 8.666 (1.406) | 3.829 (0.449) | 4.907 (0.615) | 6.335 (0.873) |
| | Pooled-MNI3 | 35.867 (4.602) | 43.077 (5.516) | 12.266 (1.418) | 10.009 (1.290) | 17.337 (2.274) | 6.611 (1.047) | 8.298 (1.338) | 7.215 (1.084) |
| 3.(b) | Target-only MNI | 13.150 (1.665) | 11.451 (2.422) | 9.587 (1.894) | 7.813 (1.015) | 7.186 (1.073) | 6.870 (1.382) | 6.328 (1.089) | 5.511 (1.017) |
| | TM1 | 7.329 (1.229) | 6.477 (1.073) | 5.936 (1.252) | 5.263 (1.098) | 4.871 (0.804) | 4.269 (0.742) | 4.138 (0.693) | 3.829 (0.766) |
| | TM2 | 9.073 (1.643) | 8.249 (1.407) | 6.711 (1.391) | 5.800 (1.148) | 5.532 (1.092) | 4.857 (0.664) | 4.836 (0.759) | 4.544 (0.903) |
| | TM3 | 11.002 (1.969) | 8.839 (1.590) | 7.858 (1.456) | 7.154 (1.305) | 6.263 (1.120) | 5.519 (1.066) | 5.317 (1.073) | 4.714 (1.002) |
| | WTM | 6.403 (1.144) | 5.516 (1.139) | 5.260 (1.467) | 4.462 (0.576) | 4.282 (1.054) | 3.541 (0.510) | 3.994 (1.194) | 3.515 (0.596) |
| | Pooled-MNI1 | 5.371 (0.620) | 4.887 (0.506) | 4.262 (0.416) | 3.935 (0.474) | 3.611 (0.381) | 3.377 (0.432) | 3.110 (0.409) | 2.883 (0.325) |
| | Pooled-MNI2 | 16.679 (1.930) | 14.829 (1.912) | 7.570 (0.869) | 7.246 (1.117) | 10.062 (1.460) | 4.507 (0.477) | 11.547 (1.624) | 9.949 (1.487) |
| | Pooled-MNI3 | 16.011 (2.094) | 16.383 (2.609) | 14.135 (2.164) | 30.615 (3.426) | 11.072 (1.285) | 12.042 (1.826) | 12.065 (1.423) | 19.334 (2.228) |
| 4.(a) | Target-only MNI | 9.831 (0.106) | 9.798 (0.116) | 9.843 (0.075) | 9.815 (0.085) | 9.816 (0.067) | 9.847 (0.061) | 9.840 (0.067) | 9.835 (0.056) |
| | TM1 | 5.966 (0.500) | 5.864 (0.467) | 5.811 (0.380) | 5.886 (0.380) | 5.953 (0.364) | 5.958 (0.372) | 6.026 (0.304) | 5.945 (0.299) |
| | Pooled-MNI | 6.179 (0.551) | 6.006 (0.525) | 5.913 (0.396) | 5.984 (0.383) | 6.071 (0.360) | 6.074 (0.395) | 6.161 (0.325) | 6.072 (0.341) |
| | SGD1 | 9.016 (0.149) | 8.916 (0.134) | 8.791 (0.120) | 8.613 (0.145) | 8.535 (0.148) | 8.464 (0.129) | 8.418 (0.154) | 8.367 (0.154) |
| 4.(b) | Target-only MNI | 9.186 (0.246) | 9.216 (0.201) | 9.157 (0.180) | 9.215 (0.159) | 9.186 (0.164) | 9.226 (0.135) | 9.175 (0.137) | 9.147 (0.141) |
| | TM1 | 7.269 (0.638) | 7.336 (0.552) | 7.225 (0.401) | 7.316 (0.332) | 7.297 (0.394) | 7.301 (0.380) | 7.289 (0.293) | 7.284 (0.293) |
| | Pooled-MNI | 8.341 (0.872) | 8.516 (0.728) | 8.381 (0.648) | 8.423 (0.547) | 8.369 (0.508) | 8.480 (0.548) | 8.408 (0.435) | 8.382 (0.418) |
| | SGD1 | 9.091 (0.129) | 8.885 (0.160) | 8.796 (0.149) | 8.647 (0.164) | 8.626 (0.155) | 8.526 (0.196) | 8.535 (0.151) | 8.522 (0.167) |
| 4.(c) | Target-only MNI | 7.686 (0.381) | 7.673 (0.299) | 7.734 (0.276) | 7.679 (0.276) | 7.710 (0.299) | 7.709 (0.258) | 7.726 (0.212) | 7.751 (0.218) |
| | TM1 | 7.274 (0.618) | 7.254 (0.523) | 7.313 (0.463) | 7.368 (0.376) | 7.286 (0.350) | 7.354 (0.351) | 7.309 (0.319) | 7.351 (0.338) |
| | Pooled-MNI | 10.574 (1.025) | 10.703 (0.945) | 10.688 (0.917) | 10.545 (0.784) | 10.579 (0.680) | 10.824 (0.728) | 10.850 (0.737) | 10.684 (0.806) |
| | SGD1 | 8.971 (0.178) | 8.826 (0.160) | 8.785 (0.132) | 8.615 (0.179) | 8.599 (0.171) | 8.481 (0.182) | 8.555 (0.196) | 8.508 (0.149) |

