# OpenReview forum: "Transfer Learning for Benign Overfitting in High-Dimensional Linear Regression"
_NeurIPS.cc/2025/Conference — NeurIPS 2025 spotlight_

### Official Review · Reviewer_zKSu · 2025-06-06

**Clarity:** 3
**Significance:** 4
**Originality:** 4
**Rating:** 5
**Confidence:** 3

**Summary:**

The paper presents a novel algorithm for transfer learning in an overparameterized linear regression setting where data is available from both a transfer and target task (“transfer MNI”). The authors provide extensive theoretical analysis characterizing the bias and variance of the transfer MNI estimator, the conditions for positive transfer, generalization bounds for transfer MNI, and so-called “free-lunch” covariate shifts.

**Questions:**

- How does violation of each of the conditions in Assumption 1 affect your results?
- How does violation of the assumption that the linear model is well-specified affect your results? This seems particularly practically important given that overparameterization can be especially beneficial in cases of model misspecification (c.f., arXiv:2109.02355).
- It was unclear to me whether and how the results applied to settings where the learner transferred knowledge from multiple sources. The analysis in section 3 appears to concern exclusively $\hat{\beta}^{(q)}\_{\mathrm{TM}}$, which I understand to be the estimator obtained with single-source transfer from the $q^{\mathrm{th}}$ task. How does transfer from multiple sources affect the results in section 3? In particular, to what extent do these results apply to $\hat{\beta}\_{\mathrm{WTM}}$, the estimator described in section 4.2?

**Ethical Concerns:**

["NO or VERY MINOR ethics concerns only"]

**Final Justification:**

The paper provides extensive theoretical analysis of and an algorithm to operate in an important although somewhat restrictive setting. As stated in my original review, I find this to be a high-quality and original paper that tackles a practically significant question. My main concern was lack of discussion of the restrictiveness of the authors’ assumptions.

In their rebuttal, the authors discussed the restrictiveness of the conditions in Assumption 1 and the effect on their results of violation of the assumption that the linear model is well-specified. Including such discussion in the paper would address my main concern, and demonstrate the potentially broader significance of the results.

**Limitations:**

I would have liked to see more discussion of the restrictiveness of the authors’ assumptions (see my Questions to the authors).

**Quality:**

4

**Strengths And Weaknesses:**

Overall, I find this to be a high-quality, significant, and original paper. The question of when, in an overparameterized learning setting, knowledge transfer from a source task improves performance in a target task is a practically significant question. The authors provide extensive theoretical analysis characterizing the bias and variance of the transfer MNI estimator, the conditions for positive transfer, generalization bounds for transfer MNI, and so-called “free-lunch” covariate shifts. They also propose and empirically assess an algorithm for identifying and incorporating informative sources.

The paper could be clearer. In particular, there are several typos in the paper. Some examples:
- Line 148: “covariates” is misspelled.
- Line 171: The sentence appears incomplete.
- Line 281: “severe” is misspelled.

Another minor point: $\mathcal{R}^{(q)}\_{\mathrm{TM}}$ and $\mathcal{R}^{(0)}\_{\mathrm{M}}$ are used in Corollary 1 and Theorem 2 without definition.

I would have also liked to see more discussion of the restrictiveness of the authors’ assumptions (in particular, Assumption 1 and the assumption that the linear model is well-specified), and the effect of violation of these assumptions on the authors’ results.

---

> ### Author Rebuttal · Authors · 2025-07-31
>
> Dear Reviewer zKSu,
>
> We deeply appreciate your thorough analysis and acknowledging the significance and originality of our findings. We would like to address your questions as follows.
>
> ***
> **Q1. Some minor typos and definitions**
>
> Thank you for pointing these typos. We have fixed the typos and will ensure to incorporate the changes to the camera-ready version once accepted. In Line 171, our full sentence is as follows: "Based on Theorem 1, we formalize the regime where the TM outperforms the target task and specify the optimal transfer size that maximizes the improvement in excess risk over the baseline."
>
> On the other hand, the notations $\mathcal{B}\_{TM}^{(q)}$ and $\mathcal{V}\_{TM}^{(q)}$ follow from Lemma 1 in our mauniscript, which stand for the bias and variance of TM respectively. The notation $\mathcal{X}$ is defined in Assumption 1 as the collection of all designs, so the expression $\mathbb{E}\_{\mathcal{X}} \mathcal{B}\_{TM}^{(q)}$ in Theorem 1 and Corollary 1 is the expected bias of TM (expectation over designs) and thus $\mathbb{E}\_{\mathcal{X}} \mathcal{V}\_{TM}^{(q)}$ is the expected variance of TM.
>
> ***
> **Q2. How restrictive are the conditions in Assumption 1?**
>
> Thank you for raising this question. We address each condition separately as follows.
> * **Sub-Gaussian covariates**: Encompassing a wide class of distribution, this is a standard assumption in relevant studies on transfer learning, including, but not limited to, [1], [2], and [3]. On the other hand, we also note that investigating the generalization of single-task min-$\ell_1$-norm interpolant is primarily based on Gaussian covariates; please see references [8]-[11] in our response to Reviewer E9gm.
>
> * **Spectral decomposition**: Although written in "Assumption", this would rather be a definition. As noted just before Assumption 1 in our manuscript, our analysis and numerical validations do not necessarily impose simultaneous diagonalizability between the target and source covariances. Hence our condition is strictly weaker.
>
> * **Bounded operator norm**: This trivially holds when $\Sigma^{(0)}=\Sigma^{(q)}$ with $||\Sigma^{(0)}(\Sigma^{(q)})^{-1}||_{op}=1$. We kindly ask the reviewer to check our response to Q1 of Reviewer KZWN regarding the generality of this assumption. A counter-example is when $\Sigma^{(0)}=I_p$ but $\Sigma^{(q)}$ is a spiked covariance with decaying eigenvalues, where the mismatch would make transfer harmful. Nevertheless, our source detection procedure is designed to filter out such negative transfers.
>
> * **Sub-Gaussian noise**: Our assumption is again milder than the Gaussian requirement (e.g., [4] and [5]). In fact, this can be readily extended to a noise having i.i.d. non-sub-Gaussian components with zero mean and variance $\sigma^2$. This is because the precise definition of excess risk (Eq. (4) of our paper) takes expectation over noise, and its expression involves the form $Tr(M \mathbb{E}\epsilon\epsilon^\top)]=\mathbb E[\epsilon^\top M\epsilon]$ for some conditioned (fixed) matrix $M$. While taking expectation over $\epsilon$ simply reduces it to $\sigma^2 Tr(M)$, sub-Gaussianity of $\epsilon$ provides a high-probability upper bound (see Lemma S.2 in [6]) on $\epsilon^\top M \epsilon$, thus guaranteeing a high-probability upper bound on the *instance-wise* variance of estimate.
>
> ***
> **Q3. How would model-misspecification affect the performance of TM?**
>
> Thank you for raising this intriguing question. While our current analysis focuses on well-specified models (as is the standard for much of transfer learning studies, e.g., [1], [2], and [7]), a precise theoretical characterization of TM under model misspecification remains beyond the present scope. Nevertheless, we would like to offer some intuition based on the structural transfer of TM and the findings discussed in your citation [8].
>
> We first kindly recall the decomposition structure of TM estimate, which is given by
> $$
> \hat\beta_{TM}^{(q)}=\hat\beta_{M}^{(0)}+(I_p-H^{(0)})\hat\beta_{M}^{(q)},
> $$
> where $I_p-H^{(0)}$ is the orthogonal projection onto the null space of target design, and $\hat\beta_{M}^{(0)}$ and $\hat\beta_{M}^{(q)}$ are independent MNIs exclusively trained on the target and source datasets respectively. Let $S:=row(X^{(0)})=Im(H^{(0)})$ and $S^\perp=Ker(H^{(0)})$. Then, any vector uniquely decomposes as $\beta = (\beta)\_S + (\beta)\_{S^\perp}$ and our TM estimate satisfies
> $$
> (\hat\beta_{TM}^{(q)})\_S=\hat\beta_M^{(0)},\quad(\hat\beta_{TM}^{(q)})\_{S^\perp}=(I_p-H^{(0)})\hat\beta_M^{(q)}.
> $$
> That is, TM "preserves" the signal learned by the target-only MNI on the target-supported subspace and "carries over" source information only in the null-space where the target offers no information. Before procedding, we kindly ask the reviewer to see how this “projection + carry-over’’ principle is further supported in our reply to **Reviewer bZxw (Q3)**.
>
> When the true regression function lies outside every linear span we may fit, the usual squared bias of an estimate splits into two terms $B_1+B_2$:
> $$
> \text{bias}^2 =\text{mis-specification bias}^2 + \text{in-class bias}^2,
> $$
> which precisely matches Eq. (22) in [8]. $B_1$ is unavoidable unless we enlarge the feature space; $B_2$ behaves as in our well-specified analysis. [8] show that, as we add extra (low-energy) features, $B_1$ and its associated variance fall faster than the new estimation error rises, so the test error curve drops again in the over-parameterized regime (double descent). Empirically they observe the misspecification components decreasing decisively beyond $p >n$.
>
> Here, we kindly request to check the three terms $U_1+U_2+U_3$ comprising the bias of TM, explained in our response to **Reviewer bZxw (Q2)**. Under the mis-specification, $U_1$ now splits into the usual leakage plus the true target signal that lies neither in the span of target or source space and lives outside the linearity. Therefore, an irreducible approximation error arises on $U_1$. On the other hand, if the source carries a good approximation of the mis-specified part, TM can still copy it into $S^\perp$ and provides some bias improvement. If the source is also mis-specified, $U_2$ now imports the source’s model approximation error, which corresponds to the phenomenon “amplified misspecification’’ termed by [8] and makes TM prone to negative transfer.
>
> Favorably, we anticipate that our free-lunch covariate shifts can still play a role, as their requirements match the remedial properties suggested by [8]: there are only a few number of (such as $\tau^\ast$ in Section 3.3 of our paper) high-energy directions and the rest $p-\tau^\ast$ low-energy directions soak up noise. With the partial/full directional alignments in free-lunch covariate shifts, TM would benefit from reduced variance inflation independent of mis-specification bias.
>
> ***
> **Q4. Drawing a theoretical connection for WTM estimate**
>
> You are absolutely right; as noted, the result in Section 3 is based on the excess risk of TM, which is fundamentally a single-source transfer method. We appreciate this opportunity to elaborate further on a promising direction for future research, as briefly outlined in Section 6 of our manuscript.
>
> To do so, we suggest some sketch of the theoretical properties of WTM. First, we recall that $I$ is the true (oracle) set of informative sources as identified in Eq. (7) of our paper. Accordingly, we define the oracle WTM (oWTM) as
> $$
> \hat\beta\_{oWTM}:=\sum\_{i \in I}w_i \hat\beta\_{TM}^{(i)},
> $$
> i.e., oWTM ensembles truly informative TM estimates, while in practice our WTM is based on the set $\hat{I}$ estimated by our cross-validation algorithm. Then, the following would hold for the oWTM:
> $$
> \mathcal{R}(\hat\beta_{oWTM})=\mathbb{E}\left[\left(\sum_{i \in I}w_i\hat\beta\_{TM}^{(i)}-\beta^{(0)}\right)^\top\Sigma^{(0)}\left(\sum_{i \in I}w_i\hat\beta\_{TM}^{(i)}-\beta^{(0)}\right)\right] \leq \mathbb{E}\left[\sum_{i \in I}w_i \left(\hat\beta\_{TM}^{(i)}-\beta^{(0)}\right)^\top\Sigma^{(0)}\left(\hat\beta\_{TM}^{(i)}-\beta^{(0)}\right)\right] \leq \sum_{i \in I}\mathbb{E}\left[\left(\hat\beta\_{TM}^{(i)}- \beta^{(0)}\right)^\top\Sigma^{(0)}\left(\hat\beta\_{TM}^{(i)}-\beta^{(0)}\right)\right] \leq |I| \max_{i \in I} \mathcal{R}(\hat\beta\_{TM}^{(i)}),
> $$
> where the first inequality follows from Jensen's inequality, and $|I|$ is the cardinality of $I \neq \emptyset$. This implies if $|I|$ is finite (which is true in general) and $\max_{i \in I}\mathcal{R}(\hat\beta\_{TM}^{(i)}) \ll \mathcal{R}(\hat\beta_{M}^{(0)})$, then oracle-WTM can achieve a robust positive transfer.
>
> Then, it remains to precisely characterize the consistency of source detection, i.e., to prove that the event $\hat{I} = I$ holds with high probability, which is outlined as our future research agenda. We anticipate that such theoretical guarantee will require a "separation condition" that distinguishes informative from non-informative sources. For example, for some "transferability threshold" $h >0$, write $\mathcal{I}_h := \\{q \in [Q]: ||\delta^{(q)}||_2 \leq h\\}$. Then, we may need the condition such as $||\delta^{(q)}||_2 \geq h\log(n_0)$ for $q \notin \mathcal{I}_h$ to infer detection consistency. Characterizing the minimal separation condition and matching it with bounds on excess risk when an uninformative source is transferred remains an open problem and is part of our planned research agenda.
>
> We sincerely hope our answers clarify your questions.
>
> ***
> **References**
>
> [1] arXiv:2404.00522.
>
> [2] arXiv:2106.12108.
>
> [3] arXiv:2404.01153.
>
> [4] arXiv:1903.09139.
>
> [5] arXiv:1805.10939.
>
> [6] arXiv:1906.11300.
>
> [7] arXiv:2406.13944.
>
> [8] arXiv:2109.02355.

---

> > ### Comment · Reviewer_zKSu · 2025-08-03
> >
> > Thank you for the thorough response, especially for your consideration of extensions of the theory to the cases of model misspecification and multiple sources. I think including a version of this discussion in the manuscript would demonstrate the potentially broader significance of the results and greatly enhance the manuscript. I maintain my recommendation to accept the paper.

---

> ### Author Response · Authors · 2025-08-04
>
> Dear Reviewer zKSu,
>
> We sincerely appreciate your constructive engagement throughout the review process and your support for the acceptance of our paper. Please feel free to reach out if any further question arises.
>
> Best regards,

---

### Official Review · Reviewer_bZxw · 2025-07-01

**Clarity:** 3
**Significance:** 3
**Originality:** 3
**Rating:** 5
**Confidence:** 3

**Summary:**

This paper studies transfer learning in linear models. The specific transfer learning set up is a form of "late-fusion knowledge transfer" with potentially multiple source tasks. The authors then study the effect of model shift and derive several results on the maximal risk improvement. Perhaps most interestingly, the authors propose a procedure to identify informative sources and a weighting scheme to use the contributions from multiple source tasks.

**Questions:**

What is the relationship between the free-lunch covariate shift and "feature overlap" suggested by recent works on transfer learning?

How scalable is the source detection algorithm in the non-linear setting? Are there efficient proxies for it?

What minimal numerical experiment could illustrate the informative weighted transfer in a non-linear setting?

**Ethical Concerns:**

["NO or VERY MINOR ethics concerns only"]

**Final Justification:**

The response resolves all my questions and I'm happy to maintain my accept score.

**Limitations:**

Yes

**Quality:**

4

**Strengths And Weaknesses:**

Quality:

Overall, the quality of the paper is high. The explanations of the results are crisp and the numerical results provide a nice demonstration of the theoretical results.

Clarity:

The paper is well-written and mostly developed logically. As is natural for a paper of this type, much of the technical detail is deferred to the Appendix. I would have prefered that the authors provide more commentary on the nature of the arguments in the main text. For example, some important objects are discussed in the main text without much context. Effective rank is a good example of this---it would be much clearer to explain what the role of effective rank will be in the context of the results and then give the definition. Overall, though, I think these are stylistic rather than substantive quibbles.

Some specific suggestions include:

- more motivation for late-fusion knowledge transfer when it is introduced
- Lemma 1, say where it is proved in the appendix, comment on the meaning of the bias term, and define $\delta^{(q)}$.

Significance:

Transfer learning is a relatively important topic given the popularity of foundation models, which indeed are often specialized for some task. Better theoretical understanding of this topic is crucially important and I think, in particular, the question of selecting good source tasks is a particularly relevant focus. On the other hand, the significance of the results is of course limited by the fact that the authors analyze only the case of linear models. If the authors could illustrate their results, even numerically, in a more realistic setting, that might improve the impact.

Originality:

There are quite a few recent works that derive similar results in either linear setting or in the context of linear probing. One thing that stood out is the "free-lunch covariate shift" results. These are quite striking but also similar to the results in: https://arxiv.org/abs/2202.10054, https://arxiv.org/abs/2206.12314, https://arxiv.org/pdf/2410.08194. I think the nature of the results should be better contextualized within this literature that is more focused on "feature learning".

---

> ### Author Rebuttal · Authors · 2025-07-30
>
> Dear Reviewer bZxw,
>
> We deeply appreciate your thoughtful comments and high appraisal of the overall quality and clarity of paper. We address your questions as follows.
> ***
> **Q1. Motivation for late-fusion knowledge transfer**
>
> Thank you for raising this important question. The term **late-fusion** contrasts with the **early-fusion** (Pooled MNI) approach to transfer learning for MNI proposed by [1]. Unlike [1], our method aligns with the typical pre-training and fine-tuning paradigm common in deep learning. One immediate advantage of our late-fusion approach is that it cleanly separates the contributions of target and source noise levels, $\sigma_0^2$ and $\sigma_q^2$. We will address your Question 3 to further substantiate the soundness of our approach.
> ***
> **Q2. Definition of $\delta^{(q)}$ and interpreting the bias of TM in Lemma 1**
>
> Please note that Line 111 in our manuscript defines the contrast vector $\delta^{(q)}:=\beta^{(q)}-\beta^{(0)}$ for each $q$-th source, which induces the model shift between the source and target. In Lemma 1, the bias of TM consists of three terms ($U_1+U_2+U_3$):
> $$
> ||(\Sigma^{(0)})^{1/2}(I-H^{(0)})(I-H^{(q)})\beta^{(0)} ||_2^2 +||(\Sigma^{(0)})^{1/2}H^{(q)}\delta^{(q)}||_2^2 +U_3,
> $$
> where $H^{(0)}$ and $H^{(q)}$ are orthogonal projections onto target and source design row space respectively. We omit the expression of cross-term $U_3$, as it is dominated by the first two terms with $2|U_3| \leq \sqrt{U_1U_2} \leq U_1+U_2$. The first term $U_1$ quantifies how much of the target signal $\beta^{(0)}$ lives in directions that neither target or source design matrix captures, as the term involves projections onto each null space, so it represents an irreducible error. It is zero only when the source design perfectly spans the signal, i.e., $H^{(q)}\beta^{(0)}=\beta^{(0)}$, which is not true in general. On the other hand, $U_2$ is the model shift energy $\delta^{(q)}$ living in directions that are seen by the source with $H^{(q)}$ but unseen by the target due to $I-H^{(0)}$. Obviously, a small model shift with low $||\delta^{(q)}||_2$  introduced by the source can lower $U_2$ (and thus $U_3$).
> ***
> **Q3. Relationship between the free-lunch covariate shift and "feature overlap"**
>
> We especially thank the reviewer for drawing this connection to the broader transfer learning literature, as it enables us to clarify how our free‑lunch covariate‑shift complements and strengthens existing feature‑overlap and LP/LP‑FT perspectives, thereby supporting the validity of our transfer approach.
>
> The analysis in [2] is framed around **concept/task drift** under a fixed input distribution: the geometry that governs transfer is the **feature-space overlap** between the source and target regression functions. In the linear model context, this overlap is parametrized by an angle $\theta\in[0,\pi/2]$ such that $\langle\beta^{(q)},\beta^{(0)}\rangle =||\beta^{(q)}||_2||\beta^{(0)}||_2 \cos\theta$. The main takeaway from [2] is that if the target function is well represented in the pretrained feature space (which manifest to large $\cos\theta$), transfer is better than target-only. This is rather a **feature-overlap/bias** concept, whereas our **free-lunch covariate shift** is characterized by a spectral relation between $\Sigma^{(q)}$ and $\Sigma^{(0)}$ that controls the variance of transfer. The setup of [2] keeps covariate distribution fixed across the target and source domains and varies the alignment of model parameter with the learned features. That said, the effect of free-lunch covariate shift is **complementary to** the feature-overlap perspective of [2]. An ideal overlap with large $\cos\theta$ would help achieve the bias of TM lower than the target task bias, and the free-lunch covariate shift can inhibit variance inflation. The two effects can play a role independently.
>
> Next, [3] investigates how fine-tuning (FT) can underperform linear probing (LP) out-of-distribution (OOD) when the pre-trained representation is strong and the shift is large, and they advocate **LP-FT** as a practical remedy. Their theoretical mechanism is that gradient updates under FT modify the feature extractor only in the training row-space and leave the orthogonal complement unchanged, potentially "distorting" features where the target test distribution places mass. Our Transfer MNI approach encodes an analogous "do not distort what the target does not constrain* principle in an analytical form and in a linear setting. To corroborate this intuition, let $S:=row(X^{(0)})=Im(H^{(0)})$ and $S^\perp=Ker(H^{(0)})$. Any vector decomposes uniquely as $\beta=\beta_S+\beta_{S^\perp}$ with $\beta_S:=H^{(0)}\beta$ and $\beta_{S^\perp}:=(I-H^{(0)})\beta$. Since our TM estimate is given by
> $$
> \hat\beta_{TM}^{(q)} = \hat\beta_{M}^{(0)} + (I-H^{(0)})\hat\beta_{M}^{(q)},
> $$
> our TM estimate exactly matches the target-only MNI $\hat\beta_{M}^{(0)}$ on $S$ and exactly preserves the source-pre-trained MNI $\hat\beta_{M}^{(q)}$ on $S^\perp$. This "projection + carryover'' structure is the linear analogue of the empirical prescription behind LP/LP-FT: fit only the target-supported directions; avoid altering directions unsupported by the target data.
>
> ***
> **Q4. Can the source detection algorithm be extended to non-linear settings?**
>
> To address your question, we evaluated the performance of our cross-validation procedure in kernel ridgeless regression (KRR) within an RKHS, operating in a nonlinear, interpolating regime. As in [4], we use the RBF kernel
> $$
> K(x,x'):= \exp(-||x-x'||_2^{2}/d),~x,x' \in\mathbb{R}^{d}
> $$
> for an ambient dimension $d$. Given a training dataset $\{(x_i,y_i)\}\_{i=1}^n$, let $K\in\mathbb{R}^{n\times n}$ be the Gram matrix with $(i,j)$-th entry $K(x_i,x_j)$. It is well-known that the minimum-RKHS-norm interpolator (KRR) is given by
> $$
> \hat{f}_M(\cdot) =\sum\_{i=1}^n \alpha_i K(\cdot,x_i)\quad\alpha_i = K^{-1}y_i.
> $$
> Let $\hat{f}\_M^{(0)}$ and $\hat{f}\_M^{(q)}$ be the KRR trained on target $(X^{(0)},y^{(0)})$ and $q$-th source $(X^{(q)},y^{(q)})$ respectively. Our TM finds the solution that minimizes the RKHS-norm distance from $\hat{f}\_M^{(q)}$ while interpolating target data, and a brief projection argument gives the analytical function form
> $$
> \hat f\_{TM}^{(q)}(x):=\hat{f}\_M^{(q)}(x)+k_0(x)^{\top}K_0^{-1}(y^{(0)}-K\_{0q}K_q^{-1}y^{(q)}),
> $$
> where $k_0(x)=(K(x,x^{(0)}_1),\dots,K(x,x^{(0)}\_{n_0}))^{\top}$, $K_0$ and $K_q$ are the target and source Gram matrices respectively, and the $(i,j)$-th entry of $K\_{0q}\in\mathbb{R}^{n_0 \times n_q}$ is $K(x_i^{(0)}, x_j^{(q)})$.
>
> In our simulation, we let $x_i\in\mathbb{R}^d$ be i.i.d. $\mathcal{N}(0,\Lambda^{(0)})$ with $d=500$, where $\Lambda^{(0)}$ is the "benign" covariance as specified in Section 5 of our manuscript. The true target function is given as in [4] by
> $$
> f_0(x)=\sum\_{l=1}^{100} 10K(x,\theta_l),\quad\theta_l\overset{i.i.d.}{\sim}\mathcal{N}(0,I_d).
> $$
> The three source functions are:
> $$
> f_1(x)=f_0(x),\quad f_2(x)=f_0(x)+\cos(5x_1),\quad f_3(x)=-f_0(x),
> $$
> where $x_1$ is the first component of $x$. Intuitively, $f_1$ would be informative, while the cosine perturbation in $f_2$ can make it less informative than $f_1$. With the anti-aligning signal, $f_3$ would be harmful to pre-train. Each data generation is given by
> $$
> y_i^{(q)}=f_q(x_i^{(q)})+\epsilon_i^{(q)},\quad i=1,...,n_q,\quad q=0,1,2,3,
> $$
> where $\epsilon_i^{(q)}$ are i.i.d. $N(0,\sigma^2)$ with $\sigma^2=1$. We set $n_0=25$ and $n_1=n_2=n_3=75$ and use $K=5$-fold CV as in our paper. The CV algorithm follows that in Appendix C, where we measure on the $k$-th target fold the validation MSEs of target-only interpolator and each TM. Averaging over the $K=5$ folds yields the CV losses $\widehat L\_M^{(0)}$ and $\widehat L\_{TM}^{(q)}$, which are compared to identify the set of informative sources $\hat{\mathcal{I}}$. The WTM function is obtained accordingly based on the weighted average of informative TMs, i.e., $\hat f\_{WTM} = \sum_\{i \in \hat{\mathcal{I}}} w_i \hat f\_{TM}^{(i)}$.
>
> For each Monte-Carlo seed, we draw $1,000$ i.i.d. unseen test points $x_t \sim \mathcal{N}(0,\Lambda^{(0)})$ and obtain the noiseless target response $f_0(x_t)$. We then record the RMSE of each estimate $\hat{f}$ given by
> $$
> \mathrm{RMSE}(\hat{f})= \frac{1}{1000}\sum\_{t=1}^{1000}(\hat{f}(x_t)-f_0(x_t))^2,
> $$
> which approximates the true excess risk $\mathcal{R}(\hat{f})=\mathbb E_x[(\hat{f}(x)-f_0(x))^2]$. We repeat this entire process across 100 independent simulation seeds and report the average of CV loss $\hat{L}$ and RMSE as follows:
>  |  | CV Loss | RMSE |
> |---|---|---|
> | Target-only | 2.567 | 1.081 |
> | TM1 | 1.896 | 0.973 |
> | TM2 | 2.129 | 1.072 |
> | TM3 | 6.279 | 1.796 |
> | WTM |  | 0.935 |
>
> Overall, our previous intuition on $f_1,f_2,f_3$ matches with the loss values. As the CV loss estimates the prediction risk $\mathcal{R}(\hat{f})+\sigma^2$ with $\sigma^2=1$, subtracting 1 from the losses put them close to the corresponding RMSEs, confirming that CV is an adequate proxy in this non-linear experiments. For TM3 that pre-trains the negative of true target signal, the sign error shows up more evidently with CV loss. But once we refit TM3 on all 25 target points to compute RMSE, the projection cancels part of the opposite signal, resulting in RMSE of only 1.796. Finally, we note that WTM consistently incorporated TM1 and TM2, and the pattern in which the RMSE of WTM is lower than that of each individual TM is consistent with what we observed in the linear MNI setting.
>
> We hope this experiment answers your question.
>
> ***
> **References**
>
> [1] Generalization error of min-norm interpolators in transfer learning. arXiv:2406.13944.
>
> [2] Features are fate: a theory of transfer learning in high-dimensional regression. arXiv:2410.08194.
>
> [3] Fine-Tuning can Distort Pretrained Features and Underperform Out-of-Distribution. arXiv:2202.10054
>
> [4] Just Interpolate: Kernel “Ridgeless” Regression Can Generalize. AoS 2020.

---

> > ### Comment · Reviewer_bZxw · 2025-08-04
> >
> > Thanks for the detailed response, this resolves all my questions and I'm happy to maintain my accept score.

---

> > > ### Author Response · Authors · 2025-08-04
> > >
> > > Dear Reviewer bZxw,
> > >
> > > We’re glad to have clarified your questions and truly appreciate your support for our paper. Please don’t hesitate to reach out if you have any further question.
> > >
> > > Best regards,

---

### Official Review · Reviewer_E9gm · 2025-07-05

**Clarity:** 3
**Significance:** 3
**Originality:** 3
**Rating:** 5
**Confidence:** 3

**Summary:**

The first paper I am aware of that demonstrates benign overfitting in transfer learning related models. The work provides a comprehensive overview of this model under the $\ell_2$ norm interpolator and studies it in great detail.

**Questions:**

Can you extend these results to a minimum norm interpolator that is not the $\ell_2$ one?

Can you extend to isotropic sub-Gaussian (even uniformly bounded entires from above and below, or isotropic log-concave i.i.d. ) covariates at least in the case of $\ell_2$ minimum norm interpolator (extending to sub-Gaussians was done recently to $\ell_p$ norm?

**Ethical Concerns:**

["NO or VERY MINOR ethics concerns only"]

**Final Justification:**

Solid work on this setting with potential impact

**Limitations:**

These results are only valid for the $\ell_2$ minimum norm interpolator with a low-rank matrix.

As I do think that the authors did a good and comprehensive work despite these limitations, I am willing to increase my score to accept, under a considerable discussion on $\ell_p$ norms (see Donhasuer et al., Kur et al., and Lecue et al. works) related to objects and the RKHS settings, and cite the relevant work. In those settings, benign overfitting was shown, so I am curious how to connect your work to those settings.

Please use an anonymous link to address this topic.

**Paper Formatting Concerns:**

I would move some of the simulations to the appendix and place some proof sketches. I do not think that these simulations give a key insight for the reader. Please space the content of this paper.

**Quality:**

3

**Strengths And Weaknesses:**

Strengh:
- \emph{extremely} comprehensive analysis
- The first paper to study this problem in this natural setting
- Potentially high-impact baseline work

Weakness:
- No novel new technical ideas that do not appear in the works of Bartlett and Koehler.
- only \ell_2 norm interpolator, i.e., a closed formula.

---

> ### Author Rebuttal · Authors · 2025-07-30
>
> Dear Reviewer E9gm,
>
> Thank you for your thoughtful comments and for recognizing the depth of our analysis and its potential as a high-impact baseline for interpolation-based transfer learning methods. We kindly note that we were not allowed to include any external links in our rebuttal, so we address your questions as follows.
> ***
> **Q1. Can the result under isotropic Gaussianity extend to isotropic sub-Gaussianity?**
>
> Our answer is yes, due to the so-called *universality* phenomenon that has been justified in other learning problems with i.i.d. designs (e.g., [1],[2]). To elaborate beyond this heuristic, we note that the main property we used to derive Theorem 1 under isotropic Gaussianity in our paper is
> $$
> \mathbb E H= \mathbb E X^\top(XX^\top)^{-1}X = (n/p)I_p,
> $$
> which follows from the rotational invariance of Gaussian distribution. Such concentration of the projection $H$ to $(n/p)I_p$ would hold for a general sub-Gaussian distribution as well.
>
> Suppose $X\in\mathbb R^{n \times p}$ has i.i.d. sub-Gaussian entries of zero mean and unit variance, so each row has covariance $I_p$. Let $z_i$ be the $i$-th column of $X$ for $i \in [p]$ and assume $X$ is of full rank. Let $A:=XX^\top=\sum_{i} z_iz_i^{\top}$, $A_{-i}:=\sum_{j\neq i}z_jz_j^{\top}$, and $H_{ij}$ be the $(i,j)$-th entry of $H$. Applying Sherman-Morrison-Woodbury (SMW) formula gives
> $$
> H_{ij} =\frac{z_i^\top A_{-i}^{-1}z_j}{1+z_i^\top A_{-i}^{-1}z_i}.
> $$
> Since $Tr(H)=\sum_{i=1}^p H_{ii}=n$ and the diagonals are i.i.d., we have $\mathbb E H_{ii}=n/p$ for all $i\in [p]$. For the off diagonal terms $H\_{ij}$ with $i\neq j$, observe that conditioning $A_{-i}^{-1}$ and ${z_j}$ that are independent of $z_i$, we have
> $$
> H_{ij}|(A_{-i}^{-1},z_j)\overset{d}{=}-H_{ij}|(A_{-i}^{-1},z_j)
> $$
> whenever $z_i$ follows a distribution symmetric about the origin, i.e., $z_i \overset{d}{=}-z_i$. This implies
> $$
> \mathbb E[H_{ij}|A_{-i}^{-1},z_j]=0
> $$
> and taking expectation once more gives $\mathbb EH_{ij}=0$ for all $i\neq j$. Thus, $\mathbb EH = (n/p)I_p$ and this result hold for **any** distribution symmetric about 0, but sub-Gaussianity (e.g., each entry of $X$ are i.i.d. $\mathrm{Unif}(-\sqrt{3},\sqrt{3})$ or Rademacher random variable) would warrant a sufficiently fast concentration of $H_{ij}$ to 0.
>
> To address a general, asymmetric sub-Gaussian distribution, let $G:=\sum_{k\neq i,j}z_kz_k^\top$, $\alpha:=z_i^\top G^{-1}z_i$, $\beta:= z_j^\top G^{-1} z_j$, and $\gamma:=z_i^\top G^{-1}z_j$ for $i\neq j$. SWM formula gives
> $$
> H_{ij} = z_i^\top(G+z_iz_i^\top+z_jz_j^\top)^{-1}z_j=\frac{\gamma}{(1+\alpha)(1+\beta)-\gamma^2}.
> $$
> Since $G$ is PSD, we have $(1+\alpha)(1+\beta)-\gamma^2 \geq 1$ and $\mathbb E|H_{ij}|\leq\mathbb E|\gamma| \leq(\mathbb E \gamma^2)^{1/2}.$ Since $z_i$ and $z_j$ are independent and  $\mathbb Ez_iz_i^\top=\mathbb Ez_jz_j^\top=I_n$, we also have $\mathbb E[\gamma^2|G]=Tr(G^{-2})$. Then, by [3], sub-Gaussianity gives
> $$
> \mathbb E\gamma^2 = \mathbb ETr(G^{-2})\lesssim\frac{n}{(\sqrt{p}-\sqrt{n})^4}\asymp\frac{n}{p^2},
> $$
> and thus, $\mathbb E|H_{ij}|\lesssim\frac{\sqrt{n}}{p}$. That is, the off-diagonals of $H$ concentrate to 0.
>
> In response to your inquiry regarding bounded or log-concave distributions, we evaluated the performance of target-only MNI and our methods under i.i.d. isotropic non-Gaussian designs. We used the same setting as in Figure 2(a) in our paper, changing only the input distribution. Specifically, we considered $\mathrm{Unif}(-\sqrt{3},\sqrt{3})$ and the Laplace distribution with location $0$ and scale $1/\sqrt{2}$. Both have mean zero and unit variance; the former is sub-Gaussian, while the latter is log-concave and sub-exponential but not sub-Gaussian. The excess risks across $p=300,500,700,900$ are as follows:
> |Estimate|Distribution|Excess Risk|
> |-|-|-|
> |Target-only MNI|Uniform|8.55 / 9.05 / 9.39 / 9.53|
> ||Laplace|8.51 / 9.05 / 9.38 / 9.52|
> |TM1|Uniform|7.88 / 8.65 / 9.04 / 9.26|
> ||Laplace|7.86 / 8.63 / 9.04 / 9.26|
> |TM2|Uniform|6.75 / 7.21 / 7.37 / 7.50|
> ||Laplace|6.78 / 7.21 / 7.36 / 7.55|
> |WTM|Uniform|5.14 / 5.97 / 5.94 / 6.12|
> ||Laplace|5.28 / 5.84 / 6.21 / 6.03|
>
> Compared to the excess risk table for Figure 2(a) on page 40 of our submission, the performance of each estimate remains nearly identical to that under the isotropic Gaussian case. We hope our explanation and empirical evidence answer your question.
> ***
> **Q2. Can we draw a connection to the RKHS setting or min-$\ell_p$-norm interpolant for $p\neq 2$?**
>
> We thank the reviewer for highlighting the connection between our setting and the extensive literature on benign overfitting under general norm constraints.  While a rich body of work analyzes the generalization of the min-RKHS-norm interpolant (**kernel ridgeless regression, KRR**), to the best of our knowledge, systematic results for KRR in transfer learning context are still lacking.
>
> [4] demonstrated that the KRR can benignly overfit in high dimensions due to an *implicit bias* shaped by kernel curvature, high ambient dimension $d$, and spectral decay of the empirical covariance and kernel matrices. In contrast, [5] proved that if $d$ is fixed, KRR with Laplace kernel is inconsistent for any bandwidth, pinpointing the "blessing of dimensionality’’ as essential for benign overfitting of KRR. [6] showed that for Gaussian kernels, ridgeless KRR with fixed $d$ is inconsistent even with optimal bandwidth tuning and can be *catastrophically overfitting*. They introduce a "taxonomy" overfitting depending on how $d=d_n$ grows with sample size $n$. [7] provide geometric upper bounds for KRR and establish a restricted isomorphism property that explains why certain data–dependent kernels (e.g., NTK) enter the benign-overfitting regime. Taken together, consistency hinges on a *spectral bias*, either implicit or explicit via geometry, and low-dimensionality demands additional regularization.
>
> Favorably, our TM estimate in linear setting admits a structure that can be extended to the RKHS; **we kindly refer the reviewer to our response to Q4 of Reviewer bZxw, where we present most mathematical expressions**. Let $\mathcal{H}$ denote a RKHS associated with a PD kernel $K:\mathcal{X}\times\mathcal{X}\to\mathbb{R}$. We fit respective KRRs $\hat f_M^{(0)}$ and $\hat f_M^{(q)}$ on target and source. Write $E_0:\mathcal{H}\to\mathbb{R}^{n_0}$ for evaluation on the target inputs and let $K_0=E_0E\_0^*$ be the target Gram matrix. The orthogonal projector onto the target representer span is
> $$
> P_0 = E_0^*K_0^{-1}E_0,
> $$
> which is a RKHS analogue of $H^{(0)}$ in our linear setting. Then, our "TM-RKHS" is of the analytical form
> $$
> \hat f_{TM}^{(q)}=\hat f_M^{(0)}+(I-P_0)\hat f_M^{(q)}.
> $$
> Let $S=Im(P_0)$ and $S^\perp=ker(P_0)$. Decomposing $\hat f=(\hat f)_S+(\hat f)\_{S^\perp}$ gives
> $$
> (\hat f\_{TM}^{(q)})_S=\hat f_M^{(0)},\quad(\hat f\_{TM}^{(q)})\_{S^\perp}=(I-P_0)\hat f_M^{(q)}.
> $$
> That is, TM "retains" the target-only KRR in target-supported directions and transfers the source KRR only where the target provides no information, mirroring the linear TM decomposition (**please check our response to Q3 of Reviewer bZxw for its validity**.) **We also numerically validate TM's performance in RKHS under the benign setting considered by [4] in our response to Q4 of  Reviewer bZxw.**
>
> Beyond the $\ell_2$ case, systematic theory exists mostly for $\ell_1$, i.e., **basis pursuit (BP)**. These works share two common grounds-isotropic Gaussian design and sparsity level $s$-but differ in how $s$ scales with dimension $d$. [8] consider linear sparsity $s=\epsilon d$ with $\epsilon<1$ and $d\asymp n$; [9] obtain consistency when $s\lesssim n/log(d/n)^5$ with $d\gg n$ but $d\leq \exp(n^{1/5})$; [10] require $s/\log(d/n)\lesssim n$. Notably, [11] identified regime where $\ell_2$-MNI can outperform BP even under sparsity.
>
> A natural analogue "Transfer BP (TBP)" in  the $\ell_1$-setting would be the anchored basis pursuit:
> $$
> \hat\beta_{TBP}^{(q)}:=\arg\min_{\beta}||\beta-\hat\beta_{BP}^{(q)}||_1\quad s.t.\quad X^{(0)}\beta=y^{(0)}.
> $$
> Under the usuaul restricted-eigenvalue (RE) condition on the target design, the interpolation constraint fixes the coordinates that the target data can identify,  while the $\ell_1$ anchor would keep the remaining coordinates close to the source solution. So heuristically, TBP would mirror the similar behavior as in the $\ell_2$-norm/RKHS transfer setting discussed earlier. It would be beneficial when target task is sparse enough, the source task itself is "$\ell_1$-benign" under one of the sparsity-dimension regimes above, and $d$ is high, but not too high to warrant the generalization of single-task BP. Heuristically, we anticipate that if the target design satisfies the RE condition (as in the references), TBP would benefit by staying close to $\hat\beta\_{BP}^{(q)}$ not captured by the target, provided that the source task itself is benign. This mirrors the algebraic decomposition obtained in the ridgeless (linear,RKHS) regimes.
>
> In contrast to $\ell_2$ and RKHS settings, the TBP lacks a closed-form expression; nevertheless, it is a convex LP problem solvable by standard methods.  Because the transfer behavior of BP remains virtually unexplored, deriving risk bounds for TBP is an open and highly promising research direction. More broadly, although our precise characterization focuses on the linear $\ell_2$ geometry, we believe the underlying “project–then–carry’’ framework provides a versatile baseline that naturally extends to both sparsity-aware ($\ell_1$) and RKHS regimes. We hope this work stimulates further investigation into when such structure-aware interpolators can deliver positive transfer under distribution shift.
>
> ***
> **References**
>
> [1] ISIT.2017.8006947.
>
> [2] arXiv:2003.10431.
>
> [3] arXiv:1011.3027.
>
> [4] 19-AOS1849.
>
> [5] arXiv:1812.11167.
>
> [6] arXiv:2409.03891.
>
> [7] arXiv:2404.07709.
>
> [8] arXiv:2110.09502.
>
> [9] arXiv:2111.05987.
>
> [10] arXiv:2012.00807.
>
> [11] arXiv:2110.02914.

---

> ### Author Response · Authors · 2025-08-05
> **We appreciate your re-evaluation and trust**
>
> Dear Reviewer E9gm,
>
> We sincerely appreciate your positive score re-evaluation and your trust in our commitment to improving the discussion of related works. As the suggested paragraphs in our previous response are marked as tentative, we will make every effort to polish the discussion of related works and add appropriate citations and cross-references in both the main text and the appendix.
>
> Once again, thank you for your thoughtful and constructive engagement throughout the review process.
>
> Best Regards,

---

### Official Review · Reviewer_KZWN · 2025-07-06

**Clarity:** 3
**Significance:** 2
**Originality:** 3
**Rating:** 5
**Confidence:** 3

**Summary:**

This investigates the bias and variance trade-off in transfer learning based on Minimum-l2-norm interpolator (MNI) in high dimensional linear regression models. A two-step transfer learning approach is proposed, and the non-asymptotic exccess risk is studied under some theoretical assumptions on model-shift and covariate-shift. Numerical performance of the proposed transfer learning procedure is evaluated through simulation studies. Overall, the paper is well written and clearly presented.

**Questions:**

1. The theoretical insights are based on rather restrictive assumptions. For example,

 (a) In point 3 of Assumption 1, the condition that all covariance matrices can be simultaneously diagonalized (as assumed in [19], [24], [29]) implies that $\prod_{j=1}^p\frac{\lambda_j^{(0)}}{\lambda_j^{(q)}}$ is bounded for all $q$. This assumption appears quite restrictive and somewhat unnatural in practice. Could the authors provide some meaningful real-world examples where this condition would realistically hold?

(b) In Section 3.1, the paper analyzes the “non-benign” case that violates Assumption 2. However, in this setting, the MNI estimator does not even provide any benefit. How do the theoretical insights from this analysis offer practical value for developing or understanding transfer learning methods?

(c) The results in Sections 3.2 and 3.3 are interesting from an intuitive perspective but seem to have limited theoretical implications. Notably, the actual estimator proposed in Section 4 is not derived from these theoretical results, but is instead based purely on using cross-validation errors. While this is intuitively reasonable, it is also fairly straightforward and does not appear to leverage the earlier theory in a meaningful way.

2. The result in Lemma 1 is somewhat counterintuitive. Typically, one would expect transfer learning to increase the effective sample size, thereby reducing the variance. However, Lemma 1 suggests that the variance is universally increased by transfer learning. Could the authors provide further clarification or intuition for why this phenomenon occurs?

**Ethical Concerns:**

["NO or VERY MINOR ethics concerns only"]

**Final Justification:**

The authors' rebuttal has clarified my earlier questions and I would like to raise my score to 5.

**Limitations:**

Yes

**Quality:**

3

**Strengths And Weaknesses:**

Strength: some insights are derived for transfer learning using MNI estimator.
Weakness: the derived results has little practical implications.

---

> ### Author Rebuttal · Authors · 2025-07-31
>
> Dear Reviewer KZWN,
>
> We deeply appreciate your valuable feedbacks and positive summary of our submission. We address your questions as follows.
> ***
> **Q1.(a) Clarifying point 3 of Assumption 1**
>
> Under simultaneous diagonalizability, we may write $\Sigma^{(0)}=VD^{(0)}V^\top$ and $\Sigma^{(q)}=VD^{(q)}V^\top$ for a shared orthogonal matrix $V$. In this case, point 3 of Assumption 1 implies
> $$
> ||\Sigma^{(0)}(\Sigma^{(q)})^{-1}||\_{op}=||VD^{(0)}(D^{(q)})^{-1}V^\top||\_{op}=\max\_{j\in [p]}\frac{\lambda_j^{(0)}}{\lambda_j^{(q)}} = O(1).
> $$
> That is, the **maximum** of eigenvalue ratios is bounded, rather than the **product** of ratios. This bounded eigenvalue ratio condition is commonly adopted by recent works on the generalization of linear model under covariate shift; please refer to, e.g., [1], [2], [3], and [4]. In addition, [5] assumes a uniform bound on the density-ratio $dP_0/dP_q$ in RKHS setting. This upper-bounds the operator norm of the kernel-integral operators, the infinite-dimensional analogue of the above $O(1)$ condition. Moreover, this condition is not confined to technical convenience. For example, in multi-site neuro-imaging, [6] found that scanner-specific variance multipliers are mostly contained in $[0.5,1.7]$, implying a bound of $\lesssim 2$. Hence our assumption aligns with current covariate shift analysis and is well grounded in real data.
>
> We hope this answer clarifies the implication and relevance of our assumption.
>
> ***
> **Q1.(b) Motive for analyzing transfer MNI under isotropic covariates**
>
> We especially thank the reviewer for raising this important question. First of all, we kindly note that Assumption 2 holds for the isotropic covariance $\Sigma=I_p$ as well, since taking $k^*=0$ gives $r_0(I_p)=p>n$. This underpins the pattern where the estimation variance of single-task MNI vanishes as $p\gg n$ even under isotropic sub-Gaussian designs. Consequently, the bias of MNI dominates its variance in high-dimension, and the excess risk is $O(||\beta^{(0)}||_2^2)$. In this sense, MNI under isotropic design is often named **harmless interpolation**-while not "benign", at least excess risk does not catastrophically explode.
>
> Although MNI under isotropic design is sub-optimal as noted, **most papers that built today’s understanding of single-task MNI start from an isotropic (and primarily Gaussian) design**, because it is the only setting where one can interpret the bias/variance terms in closed form and read off qualitative phenomena such as double descent. Many references cited in our manuscript fall into this case, including, but not limited to, [7], [8], and [9]. Isotropic design is even more predominant in analyzing the min-$\ell_1$-norm interpolant, with an additional sparsity assumption typically imposed; please see references [8]-[11] in our response to Reviewer E9gm.
>
> That said, we believe that our Theorem 1 and Corollary 1 under isotropic design provide an adequate baseline for quantifying how the measures of source quality such as signal-to-shift ratio **(SSR)** $||\delta^{(q)}||_2^2/||\beta^{(0)}||_2^2$ and signal-to-noise ratio **(SNR)** $||\beta^{(0)}||_2^2/\sigma_q^2$ interact. Our analysis explicitly accounts for the role of model shift $\delta^{(q)}$, which has not been addressed by recent works solely concerned with covariate shift (e.g., [1] and [3]). Moreover, our results draw a **favorable comparison** to [10], which proposes the **pooled MNI** approach and investigates transfer performance under isotropic Gaussian designs. As a pre-train and fine-tune approach, our TM estimate perfectly separates the contribution of target and source noise levels $\sigma_0^2$ and $\sigma_q^2$, whereas pooled MNI assumes a common noise level. In addition, Corollary 1 not only provides the necessary-and-sufficient condition for positive transfer, but also identifies the maximal excess risk improvement at the optimal transfer size $n_q^* \propto \mathrm{SNR}(1-\mathrm{SSR})$, while Proposition 3.3 in [10] only gives  a sufficient condition and fails to quantify such maximal improvement.
>
> In summary, analyzing Transfer MNI under the isotropic design furnishes the clearest lens through which one can quantify TM's bias-reduction mechanism. Moreover, in practice, **the the target-model signal magnitude $||\beta^{(0)}||^2$ is itself typically an increasing function of ambient diemsion $p$** (e.g., more informative pixels, genes, or sensor channels that additively contribute to the square signal norm). Thus, as $p$ grows, the bias component-already dominant under isotropy-swells accordingly, **creating even a larger room for TM to reduce excess risk** compared to the target-only baseline. Our simulation studies in Sections 5.2 and F.2 corroborate these insights, as TM achieves marked reduction in total excess risk under isotropic designs.
>
> ***
> **Q1.(c) Drawing a theoretical connection for WTM estimate**
>
> We also appreciate the reviewer for this question, as it closely relates to our future research agenda suggested in Section 6 of our submission. To address the question, we suggest some sketch of the theoretical properties of WTM as follows. First, we recall that $I$ is the true (oracle) set of informative sources as identified in Eq. (7) of our paper. Accordingly, we define the oracle WTM (oWTM) as
> $$
> \hat\beta\_{oWTM}:=\sum\_{i \in I}w_i \hat\beta\_{TM}^{(i)},
> $$
> i.e., oWTM ensembles truly informative TM estimates, while in practice our WTM is based on the set $\hat{I}$ estimated by our cross-validation algorithm. Then, the following would hold for the oWTM:
> $$
> \mathcal{R}(\hat\beta_{oWTM})=\mathbb{E}\left[\left(\sum_{i \in I}w_i\hat\beta\_{TM}^{(i)}-\beta^{(0)}\right)^\top\Sigma^{(0)}\left(\sum_{i \in I}w_i\hat\beta\_{TM}^{(i)}-\beta^{(0)}\right)\right] \leq \mathbb{E}\left[\sum_{i \in I}w_i \left(\hat\beta\_{TM}^{(i)}-\beta^{(0)}\right)^\top\Sigma^{(0)}\left(\hat\beta\_{TM}^{(i)}-\beta^{(0)}\right)\right] \leq \sum_{i \in I}\mathbb{E}\left[\left(\hat\beta\_{TM}^{(i)}- \beta^{(0)}\right)^\top\Sigma^{(0)}\left(\hat\beta\_{TM}^{(i)}-\beta^{(0)}\right)\right] \leq |I| \max_{i \in I} \mathcal{R}(\hat\beta\_{TM}^{(i)}),
> $$
> where the first inequality follows from Jensen's inequality, and $|I|$ is the cardinality of $I \neq \emptyset$.This implies that if $|I|$ is finite (which is true in general) and $\max_{i \in I}\mathcal{R}(\hat\beta\_{TM}^{(i)}) \ll \mathcal{R}(\hat\beta_{M}^{(0)})$, then oracle-WTM can achieve a robust positive transfer.
>
> Then, it remains to precisely characterize the consistency of source detection, i.e., to prove that the event $\hat{I} = I$ holds with high probability, which is outlined as our future research agenda. We anticipate that such theoretical guarantee will require a "separation condition" that distinguishes informative from non-informative sources. For example, for some "transferability threshold" $h >0$, write $\mathcal{I}_h := \\{q \in [Q]: ||\delta^{(q)}||_2 \leq h\\}$. Then, we may need the condition such as $||\delta^{(q)}||_2 \geq h\log(n_0)$ for $q \notin \mathcal{I}_h$ to infer detection consistency. Characterizing the minimal separation condition and matching it with bounds on excess risk when an uninformative source is transferred remains an open problem and is part of our planned research agenda.
>
> ***
> **Q2. Why is the variance of TM always higher than the target-only variance?**
>
> Before answering this question, we would like to first recall the decomposition structure of TM estimate, which is given by
> $$
> \hat\beta_{TM}^{(q)}=\hat\beta_{M}^{(0)}+(I_p-H^{(0)})\hat\beta_{M}^{(q)},
> $$
> where $I_p-H^{(0)}$ is the orthogonal projection onto the null space of target design, and $\hat\beta_{M}^{(0)}$ and $\hat\beta_{M}^{(q)}$ are independent MNIs exclusively trained on the target and source datasets respectively. Let $S:=row(X^{(0)})=Im(H^{(0)})$ and $S^\perp=Ker(H^{(0)})$. Then, any vector uniquely decomposes as $\beta = (\beta)\_S + (\beta)\_{S^\perp}$ and our TM estimate satisfies
> $$
> (\hat\beta_{TM}^{(q)})\_S=\hat\beta_M^{(0)},\quad(\hat\beta_{TM}^{(q)})\_{S^\perp}=(I_p-H^{(0)})\hat\beta_M^{(q)}.
> $$
> That is, TM "preserves" the signal learned by the target-only MNI on the target-supported subspace and "carries over" source information only in the null-space where the target offers no information. We kindly ask the reviewer to see how this “projection + carry-over’’ principle is further supported in our reply to **Reviewer bZxw (Q3)** and extends verbatim to the non-linear RKHS setting (**Reviewer E9gm, Q2**).
>
> While the mechanism explains bias reduction, it inevitably inflates variance. Since each interpolant is independent, the covariance (conditional on $X$) of TM satisfies
> $$
> Cov(\hat\beta_{TM}^{(q)}) = Cov(\hat\beta_M^{(0)})+(I_p-H^{(0)})Cov(\hat\beta_M^{(q)})(I_p-H^{(0)}) \\: \succeq \\: Cov(\hat\beta_M^{(0)}),
> $$
> so the estimation variance of TM measured on target domain inflates:
> $$
> \mathcal{V}(\hat\beta_{TM}^{(q)}) = Tr[\Sigma^{(0)}Cov(\hat\beta_{TM}^{(q)})] \\: \geq \\: \mathcal{V}(\hat\beta_M^{(0)}) = Tr[\Sigma^{(0)}Cov(\hat\beta_M^{(0)})].
> $$
>
> Nevertheless, the second term of TM lives entirely in the target null-space, so the variance inflation is confined to directions that already have "small" predictive power under the target distribution. Our main message shows that whenever the accompanying bias reduction dominates this controlled variance inflation, Transfer MNI achieves strictly lower excess risk than the target-only baseline.
>
> We hope our answers clarify your question.
>
> ***
> **References**
>
> [1] arXiv:2404.00522.
>
> [2] arXiv:2106.12108.
>
> [3] arXiv:2208.01857.
>
> [4] arXIv:2412.14474.
>
> [5] arXiv:2205.02986.
>
> [6] Neuroimage.2017.11.024.
>
> [7] arXiv:1903.07571.
>
> [8] arXiv:1903.09139.
>
> [9] 21-AOS2133.
>
> [10] arXiv:2406.13944

---

> > ### Comment · Reviewer_KZWN · 2025-08-05
> > **thanks for the clarification**
> >
> > I would like to thank the authors for the clarifications. They are very helpful. I would suggest move some of the explanations to the paper. For example, the clarification of Assumption 1 would be helpful for the readers to evaluate the content of Assumption 1.
> >
> > I have thereby raise my score accordingly.

---

> > > ### Author Response · Authors · 2025-08-05
> > >
> > > Dear Reviewer KZWN,
> > >
> > > We are delighted to hear that our clarifications were very helpful and that your concerns have been resolved. We also appreciate your suggestion as well as your positive re-evaluation of our work. Please don't hesitate to reach out if any further question arises.
> > >
> > > Best Regards,

---

### Author Response · Authors · 2025-08-09
**Summary of Author-Reviewer Discussion**

**Dear Reviewers and Area Chair,**

We deeply appreciate your insightful feedback and the opportunity to address the points raised during the discussion period. The exchange has been highly productive and has helped us strengthen both the clarity and scope of our work. As the discussion period approaches the end, we would like to provide a brief summary.

***
**Strengths noted in the initial reviews** (reviewer IDs in parentheses):
* Well-written, clearly presented, and logically developed (KZWN, bZxw)
* **Extremely comprehensive** analysis (E9gm)
* Potentially **high-impact baseline work** at the intersection of transfer learning and benign overfitting (E9gm)
* High-quality paper with **crisp results** and numerical results providing a **nice demonstration** of theoretical analyses (bZxw)
* **High-quality, significant, and original** contribution (zKSu)

***
**Main questions raised and addressed during the discussion:**
* How restrictive are our assumptions? (Q1.a, KZWN; Q2-Q3, zKSu)
* How should the bias of TM be interpreted? Why is the variance of TM always higher than that of the target-only MNI despite an increased total sample size? (Q2, bZxw; Q2, KZWN)
* Why analyze transfer performance under isotropic (Gaussian) covariates? Can the result be extended to isotropic sub-Gaussian covariates? (Q1.b, KZWN; Q1, E9gm)
* How does our theoretical analysis apply to the WTM estimate? (Q1.c, KZWN; Q4, zKSu)
* How does our finding of *free-lunch covariate shift* relates recent discussion on "feature overlap"? (Q3, bZxw)
* How does our source transferability detection algorithm perform in non-linear settings? (Q4, bZxw)
* Can our proposed transfer scheme be extended to min-$\ell_p$-norm interpolators ($p \neq 2$) as well as to non-linear (RKHS) settings? (Q2, E9gm)

***
**Key clarifications provided in rebuttals:**

We illustrated that the projection-based **“Retain+Transfer”** mechanism of our proposed TM estimate is well grounded in the principle of LP-FT (Q3, bZxw). We further showed that this structure extends naturally to non-linear RKHS settings (Kernel "Ridgeless" Regression) with a closed-form expression (Q2, E9gm), and we numerically demonstrated a RKHS setting where the RKHS version of TM estimate facilitates positive transfer under benign overfitting (Q4, bZxw).

***
**Looking forward:**

As also noted by Reviewer E9gm, we believe our method has potential as a high-impact baseline for transfer learning with interpolating methods. As suggested in our rebuttals, we believe that the "Retain+Transfer" mechanism offers a versatile baseline that can naturally extend to both $\ell_1$-norm and RKHS-norm-aware transfer under distribution shifts. Motivated by the valuable suggestions from all reviewers, we are committed to incorporating them and highlighting this promising future research direction in our revised version.

***

We thank all reviewers for their constructive engagement and thoughtful questions, which have substantially improved the quality and clarity of our work. We look forward to reflecting these enhancements in the revised version.

Sincerely,

---

### Decision · Program_Chairs · 2025-09-17

**Decision:**

Accept (spotlight)

**Comment:**

The paper focuses on bias and variance trade-off in high-dimensional transfer learning by minimum $\ell_2$-norm interpolator. The authors propose a transfer learning strategy and provide both a theoretical analysis and a numerical study of its performance. The paper has been appreciated for its quality, clarity, and potential impact, collecting unanimous high ratings by all reviewers. The discussion period between reviewers and authors lead to the clarification of some points related to the possible limitations of the setup, and a further increase in the scores. The theoretical analysis of transfer learning in modern machine learning and benign overfitting is naturally extremely important in the current machine learning research, and the submission has been recognised as a solid, comprehensive and high-impact contribution in this direction. I therefore recommend its acceptance as *spotlight*.